# A humanized NOVA1 splicing factor alters mouse vocal communications

Yoko Tajima [1] ✉, César D. M. Vargas[2], Keiichi Ito[3], Wei Wang [4], Ji-Dung Luo [4], Jiawei Xing [5], Nurdan Kuru[5], Luiz Carlos Machado[5], Adam Siepel [5], Thomas S. Carroll[4], Erich D. Jarvis [2,6] & Robert B. Darnell [1,6] ✉

NOVA1, a neuronal RNA-binding protein expressed in the central nervous system, is essential for survival in mice and normal development in humans. A single amino acid change (I197V) in NOVA1's second RNA binding domain is unique to modern humans. To study its physiological effects, we generated mice carrying the human-specific I197V variant (*Nova1^{hu/hu}*) and analyzed the molecular and behavioral consequences. While the I197V substitution had minimal impact on NOVA1's RNA binding capacity, it led to specific effects on alternative splicing, and CLIP revealed multiple binding peaks in mouse brain transcripts involved in vocalization. These molecular findings were associated with behavioral differences in vocalization patterns in *Nova1^{hu/hu}* mice as pups and adults. Our findings suggest that this human-specific NOVA1 substitution may have been part of an ancient evolutionary selective sweep in a common ancestral population of *Homo sapiens*, possibly contributing to the development of spoken language through differential RNA regulation during brain development.

Fossil records indicate that modern humans (*Homo sapiens*) emerged 200,000–300,000 years ago as the predominant species from a common ancestral population[1,2]. Humans differ significantly from their closest living relatives, the great apes, particularly in their ability to communicate through complex learned vocal communication, a necessary component of spoken language[3]. This complexity is driven by some anatomical adaptions of the vocal tract and intricate neural networks linking various brain regions[3–7]. However, the genetic basis underlying these specialized human traits remains to be fully identified.

The closest evolutionary relatives of modern humans are two extinct lineages: Neanderthals and Denisovans. Genome sequencing from fossilized remains of these archaic humans has identified distinct genetic differences between them and modern humans, which may be relevant to recent human evolution[8–11]. Additionally, the availability of extensive human genome data over the past few decades initially focused on European populations, has significantly expanded the scope of evolutionary studies[12–14].

The transcription factor forkhead box P2 (FOXP2) is of particular interest as a potential driver of human language function, as it harbors two amino acid substitutions present in humans but not in chimpanzees and many other mammalian genomes. Families with FOXP2 mutations exhibit severe speech defects[15,16] while FOXP2 disruption in mice leads to vocalization abnormalities[17,18] suggesting a role in spoken language function. Studies on mice with the two amino acids substituted to the human version have reported vocal changes both in the neonatal[19] and adult stages[20,21]. While Hammerschmidt et al. observed minimal vocal changes, von Merten et al. reported qualitative changes under a more natural vocalization paradigm[20,21], suggesting the involvement of these two amino acids in vocalization. However, these substitutions are also present in archaic humans, and comprehensive

[1]The Laboratory of Molecular Neuro-oncology, The Rockefeller University, New York, NY, USA. [2]The Laboratory of Neurogenetics of Language, The Rockefeller University, New York, NY, USA. [3]The Laboratory of Biochemistry and Molecular Biology, The Rockefeller University, New York, NY, USA. [4]Bioinformatics Resource Center, The Rockefeller University, New York, NY, USA. [5]Simons Center for Quantitative Biology, Cold Spring Harbor Laboratory, Cold Spring Harbor, New York, NY, USA. [6]Howard Hughes Medical Institute, The Rockefeller University, New York, NY, USA. ✉e-mail: ytajima@rockefeller.edu; darnelr@rockefeller.edu

analyses using diverse human genome datasets have found no evidence of recent selection. This suggests that the FOXP2 substitutions occurred earlier than initially thought[12,22]. Similarly, the TKTL1 gene contains a human-specific amino acid thought to influence greater neurogenesis in humans than Neanderthal frontal cortex, though this finding is based on European ancestry genome datasets[23]. Broader analyses of modern human genomes reveal that 0.03–0.2% of individuals possess the 'putative Neanderthal variant', indicating its presence in a significant portion of the population[14]. These findings underscore the importance of incorporating diverse human samples to identify and validate the genetic background of modern human traits through genomic comparisons.

Genomic comparisons between archaic humans, ape genomes, and the broader human population have identified 61 human-specific non-synonymous coding variants that are fixed or nearly fixed in modern humans[13]. Notably, one of the genes includes an isoleucine to valine substitution at position 197 (I197V) in the RNA binding protein neuro-oncological ventral antigen1 (NOVA1)[8,13]. NOVA1 is highly expressed in neurons of the central nervous system (CNS) in both mice and humans[24], with its expression also noted in cultured human and rat cells[25–27]. NOVA1 was first identified as an autoantigen targeted in the paraneoplastic neurologic disorder (PND) opsoclonus-myoclonus ataxia (POMA)[24]. PNDs develop when tumor cells ectopically express proteins normally restricted to the nervous system, triggering an anti-tumor immune response that breaches the blood-brain barrier, leading to autoimmune neurologic disease[28,29]. In POMA, a robust immune response is mounted against NOVA1 and its paralog, NOVA2[30]. The autoimmune disorder is characterized by motor dysfunction due to the failure midbrain neuron inhibition, resulting in the hyperactivity associated with opsoclonus-myoclonus ataxia[24]. In mice, homozygous deletion of the Nova1 gene results in an early postnatal lethal phenotype due to abnormal motor function[31]. Therefore, NOVA1 plays a crucial role in neural development and neuromuscular control in mammals. On a molecular level, NOVA proteins directly bind RNA in the mouse brain[32,33] to regulate pre-mRNA processing[31,34,35], translation[36] and neurophysiology[37]. Genetic studies mapping NOVA target RNAs in mice and humans have also linked it to autism[38]. Interestingly, a human patient with a heterozygous deletion of NOVA1 presented with delay of language development, learning disabilities, motor hyperactivity and behavioral dysregulation[36].

Studies have explored the role of the NOVA1 I197V variant by reverting the ancestral isoleucine 197 variant back into human iPSC-derived organoids reveal morphological and electrophysiological changes in vitro[13,39]. While these effects were not observed in a study that reintroduced the same substitution in different iPSCs[40], technical concerns continue to make definitive conclusions about the nature of the NOVA1 I197V variant in brain challenging. Therefore, we generated humanized mice harboring this variant to study its consequences for RNA regulation and behavior in vivo.

In this work, we used gene-editing to substitute the NOVA1 isoleucine (I) isoform present in most mammals and archaic hominids (Neanderthals and Denisovans) with the human-specific valine (V) variant at position 197 in mice. Comparison of these humanized NOVA1 mice ($Nova1^{hu/hu}$) with wild-type mice carrying the ancestral Nova1 gene ($Nova1^{wt/wt}$) revealed specific transcriptomic and behavior differences related to vocalization. Taken together, the unique role of NOVA1 in neurons, its association with human disease, and evidence that the human-specific amino acid 197 variant confers vocalization changes in humanized mice suggest a role for NOVA1 in the evolution of human-specific language.

## Results

### NOVA1 I197V has characteristics of a variant that underwent a strong evolutionary selective sweep in modern humans

NOVA proteins have three K homology (KH)-type RNA binding domains and bind YCAY repeat sequences on target transcripts[36,41–45].

A comparison of the amino acid sequences of NOVA1 and NOVA2 across various organisms showed that NOVA1 is highly conserved throughout the entire protein sequence, whereas NOVA2 exhibits greater variability between species (Fig. 1a). We compared eight modern human genomes with three high-coverage Neanderthal genomes and one high-coverage Denisovan genome. The only change between modern and ancient humans was a nonsynonymous nucleotide substitution encoding the 197th amino acid of NOVA1, resulting in a valine in modern humans replacing an isoleucine in ancient humans (Fig. 1b). An expanded genomic analysis of the dbSNP database revealed that this substitution was present in all but six of 650,058 human sequences, five of which were from individuals of Asian descent (Supplementary Fig. 1a). As the samples are deidentified, it is not possible to assess additional details about these individuals.

We analyzed the mean allele frequencies (MAF) of NOVA1 in 121,412 human genomes from the ExAC database (Fig. 1c, Supplementary Fig. 1b). This analysis confirmed the low frequency of variants encoding 197 V in humans, consistent with strong selective pressure across the entire NOVA1 coding sequence. Specifically, the upper bound of 95% confidence interval for the MAF of NOVA1 was 0.00071, significantly lower than that of NOVA2 (0.0099) or the average across all genes on chromosome 14, where NOVA1 is located (0.0042; Fig. 1c). Moreover, evolutionary analysis on the NOVA1 gene yielded a Tajima's D statistic of -2.48. Normalizing Tajima's D values between genes by the theoretical minimum[46], we found that NOVA1's normalized Tajima's D statistic was exceptionally low (Supplementary Data 1). This suggests that NOVA1 has undergone strong purifying selection, particularly in comparison to NOVA2, neighboring genes (FOXG1, STXBP6), and a set of genes associated with 61 human-specific nonsynonymous coding variants[13] (Fig. 1d, Supplementary Fig. 2, Supplementary Data 1).

To further investigate whether a selective sweep occurred at the NOVA1 gene locus, we performed the DH test, a method for detecting ancient selective sweeps that focuses on high-frequency variants. This method is relatively robust to confounding effects caused by background selection and demographic changes[47]. Using human genetic data from the 1000 Genomes Project, we calculated the DH value for the NOVA1 locus, which was nominally significant at $p = 0.046$. However, this result is only suggestive, given the multiple hypothesis testing involved in our exploration of several selection tests.

To increase statistical power, we analyzed the ancestral recombination graph (ARG) for modern and archaic hominins in the surrounding region, using ARGweaver-D[48], and applied the CLUES2 software[49] to estimate a selection coefficient that best explains the observed changes in allele frequency over time around the NOVA1 SNP. We inferred that selection at the NOVA1 SNP is $s = 0.00082$ ($p = 0.019$ for the null hypothesis of no selection). This corresponds to a population-scaled coefficient of $S = 2N_e s \cong 19$, indicating strong selection relative to nearly neutral evolution (where $|S| <= 1$) (Supplementary Fig. 3). For comparison, we performed the same CLUES2 analysis to 38 other SNPs with informative ARGs, previously identified as potentially selected in the human genome[13]. The results indicated that selection at the NOVA1 SNP was relatively strong compared to these other genes, with 33 of the 38 showing either non-significant results or smaller selection coefficients (Supplementary Data 2). The human-specific variant in NOVA1 resides on the third-largest human-specific haplotype among the fixed human-specific sites—featuring two high-frequency haplotypes[13]—further supporting the hypothesis that the haplotype on which this variant resides was subject to a selective sweep.

Taken together, the observation that the NOVA1 197 V allele became nearly fixed and is shared across human population groups suggests it arose and increased to high frequency before their divergence. Our analyses support the idea that the NOVA1 197 V variant was part of an ancient selective sweep in modern humans, predating many other known sweeps in the human genome.

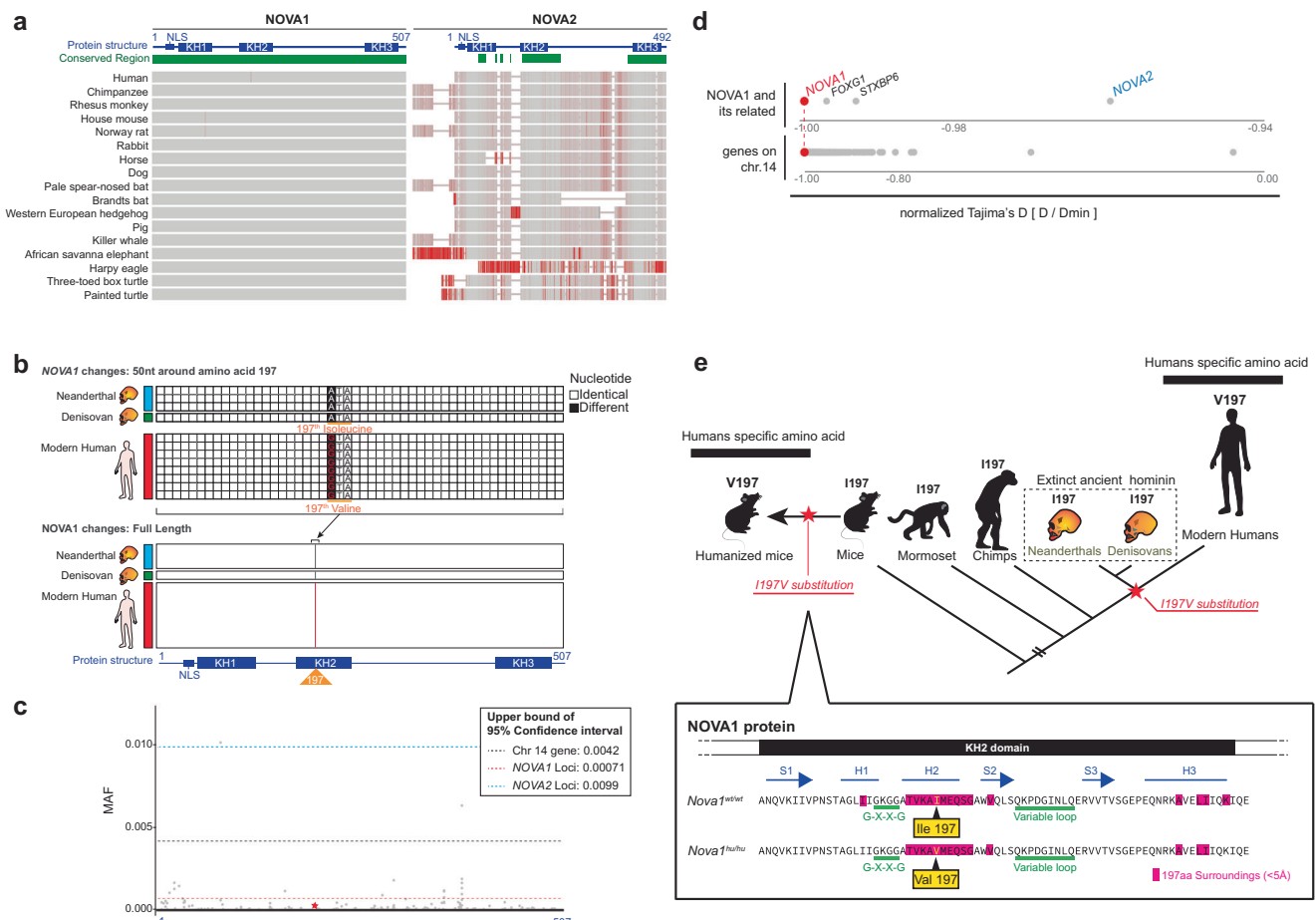

**Fig. 1 | Evolutionarily conserved NOVA1 harbors a modern human-specific amino acid at position 197. a** Conservation analysis of NOVA1 and NOVA2 protein conservation across species using high-quality genome assemblies from NCBI. The analysis was performed using NCBI's Constraint-based Multiple Alignment Tool (Cobalt). "Conserved Region" is defined by the relative entropy threshold of the amino acid residue. Green bars indicate highly conserved positions. Gray-red bars indicate the Column Quality score: scores for amino acid residues based on agreement within the column/position. Rare residues are highlighted in darker red, while positions with any mismatch are anchored. **b** Sequence comparison of *NOVA1* coding sequences (CDS) from four ancient humans (three Neanderthals; blue bar and one Denisovan; green bar) and eight modern humans (red bar). The upper panel highlights the 50 bases surrounding the 197th amino acid; the lower panel shows the full *NOVA1* CDS. **c** SNP frequency analysis of *NOVA1* gene in modern humans. Mean allele frequencies (MAFs) across the *NOVA1* gene from 121,412

modern humans are shown as gray dots. The upper bound of the 95% confidence interval for *NOVA1* SNPs is indicated by red dotted lines, while blue- and black-dotted lines indicate those for *NOVA2* and all genes on chromosome 14 (where *NOVA1* gene is located), respectively. The red star indicates the V197I variant: rs762662114, chr14:26448894/ GRCh38.p14. Source data are provided as a Source Data file. **d** Comparison of normalized Tajima's D values. The first gene set includes *NOVA1* and *NOVA2*, and *NOVA1*-neighboring genes (*FOXG1* and *STXBP6*) on chromosome 14. The second gene set includes all genes on chromosome 14. **e** Model of the evolutionary timing for the 197th amino acid change in the *NOVA1* gene, noting the *Nova1^{hu/hu}* mice generated in this study. *Nova1^{hu/hu}* mice express the modern human-specific amino acid in the NOVA1 protein. The bottom panel shows the corresponding position within the KH2 domain of the NOVA1 protein. Amino acids structurally proximal (<5 Å) to the 197th amino acid, as predicted by AlphaFold2, are highlighted in pink.

## Humanized NOVA1 mice are comparable to wild-type mice in development and gene expression in the brain

To explore the physiologic and biological significance of the I197V amino acid substitution in NOVA1, we used CRISPR/Cas9-based gene editing to introduce nucleotide changes, generating human-type NOVA1 knock-in mice (*Nova1^{hu/hu}* mice; Fig. 1e, Supplementary Fig. 4). *Nova1^{hu/hu}* mice developed normally (Fig. 2a) and exhibited fertility similar to that of littermate controls (*Nova1^{wt/wt}*). The brain to body weight ratio was comparable between *Nova1^{hu/hu}* and *Nova1^{wt/wt}* mice (Fig. 2b). Comprehensive gene expression analysis of the midbrain at embryonic day 18.5 (E18.5) and postnatal day 21 (P21), when fundamental neural circuits and behaviors have been established, revealed that the transcript levels, including *Nova1* itself, were nearly identical between genotypes throughout development (Fig. 2c) and across different brain regions (Supplementary Fig 5a-d). Similarly, the expression pattern and levels of NOVA1

protein were equivalent between genotypes in the brain (Supplementary Fig. 6).

The only gene that showed a significant steady-state difference in *Nova1^{hu/hu}* mice was *Gkn3* at P21, a secreted protein involved in endothelial cell proliferation (Fig. 2c; *p*-value < 0.05, FDR < 0.1). *Gkn3* is thought to be involved in adaptive gene loss during recent human evolution[50], and showed down-regulation in the P21 midbrain of *Nova1^{hu/hu}* mice (average TPM 18.6 in *Nova1^{wt/wt}*, 11.3 in *Nova1^{hu/hu}*, $\log_2 FC = -0.71$, *p*-value = $2.3 \times 10^{-6}$, FDR = 0.037).

## Modern human-specific amino acid substitution does not affect sequence-specific RNA-binding capacity of NOVA1

NOVA proteins harbor three KH domains that are responsible for sequence-specific RNA-binding[41,44,45]. The KH domain, found in many RNA binding proteins, includes common motifs: an invariant Gly-X-X-Gly motif, a hydrophobic core, and a variable loop. The 197th amino acid in

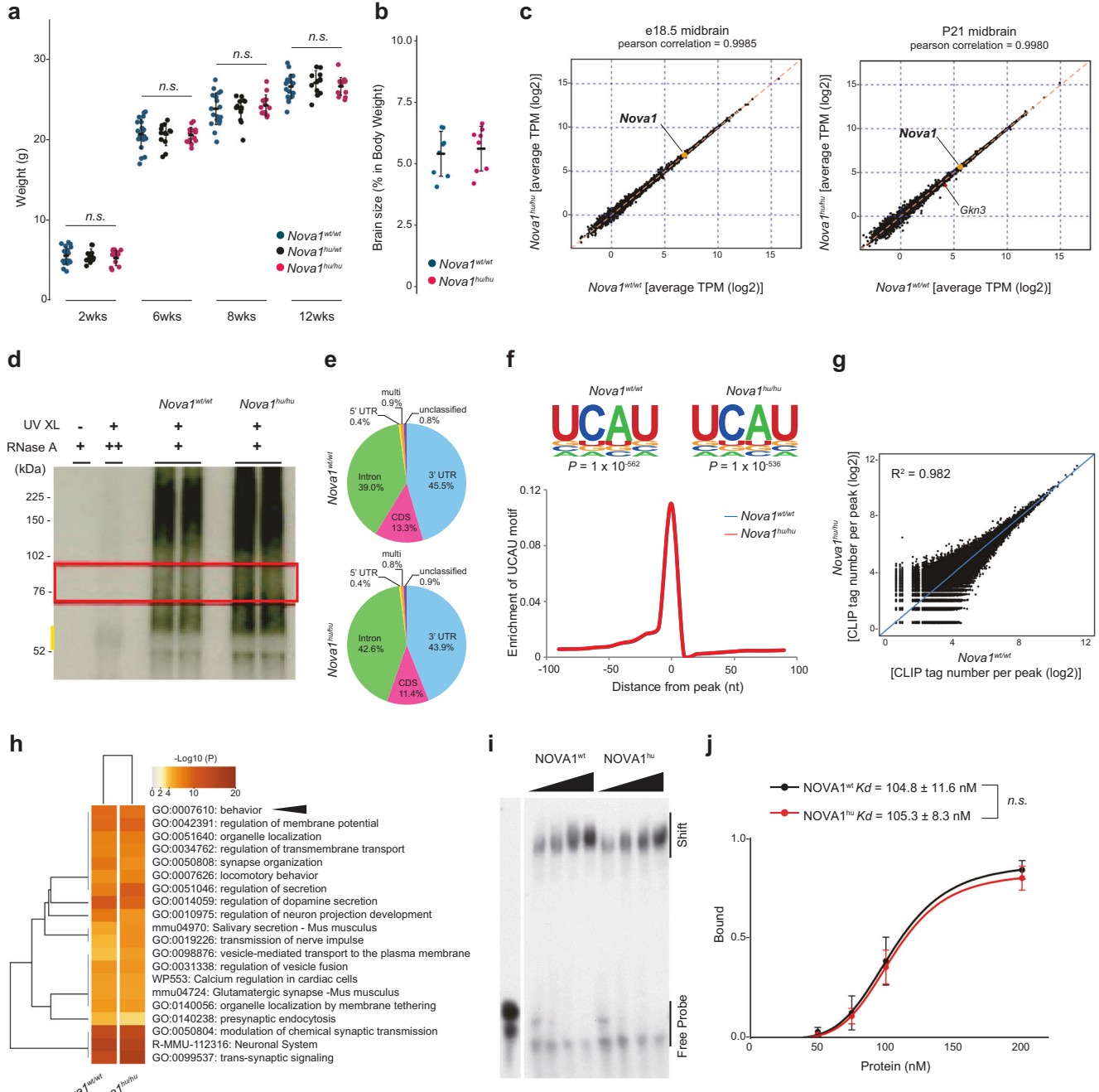

**Fig. 2 | Human specific amino acid substitution does not affect mouse development, gene expression and RNA binding characteristics in NOVA1.**
**a** Body weight comparison measured from 2-12 weeks postnatal. *Nova1^wt/wt*, $N = 19$, *Nova1^hu/wt*, $N = 12$, *Nova1^hu/hu*, N = 14. **b** Brain-to-body weight ratio comparison in 3-week-old mice ($N = 8$ per group). **a**, **b** Data are presented as mean values +/- SEM. Wilcoxon rank sum test, two-sided. **c** Gene expression correlations between *Nova1^wt/wt* and *Nova1^hu/hu* in midbrain at E18.5 and P21. Scatterplots show gene expression in average TPMs ($log_2$). The red dot marks one differentially expressed gene (*Gkn3*; $p < 0.05$, FDR < 0.1), and the yellow dot indicates *Nova1*. E18.5 midbrain: $N = 6$ per group; P21 midbrain: $N = 4$ per group. Statistics are performed using edgeR. **d** Representative autoradiography image from three independent NOVA1-CLIP experiments of 3-week-old midbrain in *Nova1^wt/wt* and *Nova1^hu/hu* mice. The yellow line marks the NOVA1 protein size, and the red outline highlights RNA extraction by sectioning. **e** Distribution of NOVA1 CLIP peaks on the mouse genome. **f** The most enriched binding sequence from NOVA1 CLIP peaks (top) and frequency of the UCAU binding sequence around the CLIP binding sites (bottom). **g** Scatterplot of CLIP tag number per peak between *Nova1^wt/wt* and *Nova1^hu/hu* mice at P21 midbrain ($log_2$). $N = 3$ per group. **h** Gene annotation analysis by Metascape[127] of NOVA1 bound transcripts. Transcripts with the top 1% peak heights (read count) for each genotype were analyzed. Genes expressed in the midbrain of P21 mice served as the background set. The term 'behavior' is highlighted with a black arrowhead. **i** Gel shift assay using purified NOVA1 protein and $^{32}$P-labeled UCAU RNA oligo probe. The representative image from five independent experiments. The bottom band represents the free probe, and the top-shifted band indicates protein-bound RNA. **j** Quantified bands from the gel shift assay showing binding per protein concentration. Dissociation constants (Kd values) for each purified NOVA1 protein are plotted. Data are presented as mean values +/- SE. Two-tailed Student's *t*-test. $N = 5$ for each point. Source data for (**a**, **b**, **d**, **j**) are provided as a Source Data file.

NOVA1 is located within this hydrophobic core. Previous studies have demonstrated that single amino acid substitutions within the hydrophobic core can cause loss of function in RNA binding proteins[51–53]. In Nova1, in vitro binding assays have demonstrated that substitutions such as Ile→Thr in KH1 and Leu→Asn in KH3 in the hydrophobic cores result in loss of RNA binding capability[41,54] (Supplementary Fig. 7).

To determine whether the I197V substitution affects NOVA1's RNA binding ability, we performed CLIP (cross-linking and immunoprecipitation) to compare NOVA1 genome-wide binding maps in *Nova1^{hu/hu}* and *Nova1^{wt/wt}* midbrains at P21 (Fig. 2d). Across three biological replicates for each genotype, we identified 26,155 binding peaks in *Nova1^{hu/hu}* and 27,720 in *Nova1^{wt/wt}* mice, respectively. NOVA1 binding peaks were predominantly located on introns and 3'UTRs (Fig. 2e). The binding motifs were highly enriched for the known NOVA1 binding sequence (UCAU repeats), and the genomic distribution of NOVA1 binding to this motif was highly similar between genotypes (Fig. 2f). The number of tags in the detected CLIP peaks were highly comparable ($R^2 = 0.982$), with only minor differences in low-count peaks (a total of 250 peaks: average CLIP tags/ peak 16.3 in *Nova1^{hu/hu}*, 13.1 in *Nova1^{wt/wt}*, $p$-value < 0.01, $|\log_2 FC| > 1$) (Fig. 2g, Supplementary Data 3). In both genotypes, NOVA1-bound transcripts were strongly enriched for those encoding proteins involved in the synaptic signaling, transmission and secretion (Fig. 2h). These nearly identical characteristics were also observed in the genomic distribution of NOVA1 binding peaks, enriched motifs, and peak correlations in cortex and cerebellum at P21 (Supplementary Fig. 8). Notably, NOVA1-bound transcripts identified by CLIP analysis were enriched for behavior- and synapse-related categories across midbrain, cortex and cerebellum (Fig. 2h, Supplementary Fig. 8e, 8j).

We also performed in vitro RNA binding assays to compare the RNA binding characteristics of the NOVA1 proteins. Full-length NOVA1 proteins were purified and RNA oligonucleotides containing the NOVA1 binding site (UCAU repeat sequence with stem-loop structure[41]) were used for gel shift assays. Both humanized and wild-type NOVA1 protein bound $^{32}$P-labeled UCAU-RNA and caused a dose-dependent shift in migration (Fig. 2i), with indistinguishable binding dissociation constants (Kd for NOVA1^{hu} protein was 105.83 ± 8.3, and for NOVA1^{wt} protein was 104.8 ± 11.6; Fig. 2j). Taken together, these in vivo and in vitro studies reveal not only the resilience of the I197V variant in maintaining the biophysical properties of RNA binding with minimal global disruption but also its remarkable conservation of overall function. However, this variant exerts specific effects on alternative splicing (AS), as explored in the following section.

## Humanized NOVA1 mice exhibit alternative splicing changes of specific genes

The role of RNA binding proteins in mRNA processing is influenced by various factors beyond their inherent RNA binding capability. Several studies suggest that interactions with competing or cooperating proteins, as well as non-protein factors like metals or ATP, contribute to determining the downstream effects on target mRNAs[55–60]. NOVA proteins, in particular, regulate the splicing or stabilization of specific mRNA targets through their interactions with other proteins[36,43,61]. While the I197V substitution has minimal impact on RNA affinity or sequence specificity, X-ray crystallography and protein structure predictions suggest subtle changes in amino acid interactions within the KH domain (Supplementary Fig. 9, see Discussion). These changes may influence protein-protein interactions or NOVA dimerization[62], prompting us to investigate the I197V variant's impact on RNA regulation, particularly on AS.

We first determined the brain regions for AS analysis based on the expression patterns of NOVA1.

Immunostaining of mice brains at E18.5 and P21 showed that NOVA1 is expressed throughout brain development across various regions (Fig. 3a, Supplementary Fig. 10). NOVA1 is most highly expressed in midbrain, with low-levels across cortical layers, sparse expression in the striatum and hippocampus, and intermediate levels in the granular layer of the cerebellum. Analysis of NOVA protein expression in the P21 mouse brain using NOVA1 and panNOVA antibodies revealed that among the various isoforms of NOVA1 and NOVA2, a specific NOVA1 isoform (lacking Exon4[63]) is highly expressed in the midbrain, particularly in the periaqueductal gray (PAG) region (Fig. 3b, Supplementary Fig. 11). The PAG is involved in a broad range of physiological and behavioral functions, including defense reaction, pain and anxiety, fear, micturition, and vocalization[64–72], and is thought to integrate sensory signals from the periphery, acting as a control center for behavioral regulation[73–75]. AS analysis was thus performed in the mouse midbrain, where NOVA1 is most highly expressed.

AS analysis in the P21 midbrain revealed that several AS events that were specifically altered in *Nova1^{hu/hu}* mice compared to *Nova1^{wt/wt}* controls (Fig. 3c). Specifically, 720 events showed significant changes ($|dI| > 0.05$, $p < 0.05$; Supplementary Data 4). Among the most intriguing changes were AS of tandem cassette exons in *Fnbp1l* (formin binding protein 1-like, exons 10 and 11), a gene implicated in human intelligence[76,77], and cassette exons in *Itpr1* (inositol 1,4,5-trisphosphate receptor type 1, exon 41), a receptor that mediates calcium release from the endoplasmic reticulum (Fig. 3d). The effect of the I197V substitution on AS was smaller than the effects observed in previous NOVA knockout studies[31,43,61].

We further explored the NOVA1 I197V variant's influence on AS by cross-referencing the CLIP and gene annotation datasets and assessing the statistical significance of the results using random resampling methods, as detailed below. This analysis specifically aimed to assess NOVA1 binding on AS transcripts. Of the 720 differential AS events, 258 (41%) had NOVA1 binding peaks on their transcripts (Fig. 3c). The AS splicing changes were not specific to NOVA CLIP targets. While NOVA1 CLIP-peaks link some AS events directly to the NOVA1 I197V variant, NOVA1-CLIP were not detected on 59% of transcripts with AS changes. This may relate to technical issues, including the lack of resolution of NOVA1-CLIP on large areas of brain studied here that are unmasked in cTag CLIP studies undertaken in individual cell types[61,78]. Gene annotation analysis revealed that the 630 transcripts with differential AS events were enriched in processes related to cell projection organization or chromatin remodeling, while the transcripts with NOVA1 binding peaks showed further enrichment in processes involving cell projection, morphogenesis, and synaptic function (Fig. 3e).

To explore the potential effects on behavioral control in *NOVA1^{hu/hu}* mice, we examined behavior-related genes among those transcripts showing differential AS events. Of the 630 transcripts with differential AS events, 27 were associated with the behavior category (Supplementary Data 5). Remarkably, the vocal behavior term showed the highest ratio of gene coverage, with four of the 22 genes annotated for vocal behavior (*Auts2, Myh14, Nrxn2, Srpx2*)[79–82] being differentially spliced in *Nova1^{hu/hu}* mice relative to *Nova1^{wt/wt}* mice (Fig. 3f, Supplementary Fig. 12). This number of vocalization-related genes was significantly higher than expected by chance, exceeding the 5% significance threshold in random re-samplings tests (1000 resamplings, mean 0.71, median 1; Supplementary Fig. 13).

Interestingly, many genes involved in vocalization, including *Foxp2*, *Celf6*, *Auts2*, *Nrxn1-3*, and *Shank1-3*, showed reproducible NOVA1 binding peaks across multiple brain regions (Supplementary Fig. 14). Given that Pasilla, the fly ortholog of mammalian NOVA1/2, regulates splicing of target transcript in an experience-dependent manner[83], it is plausible that these vocalization related transcripts are similarly affected in a context-dependent manner, such as in response to sensory cues from surrounding environment. Together, these findings indicate that *Nova1^{hu/hu}* mice with I197V substitution exhibit subtle but specific impact on RNAs in the brain, particularly in genes involved in animal behavior and vocalization.

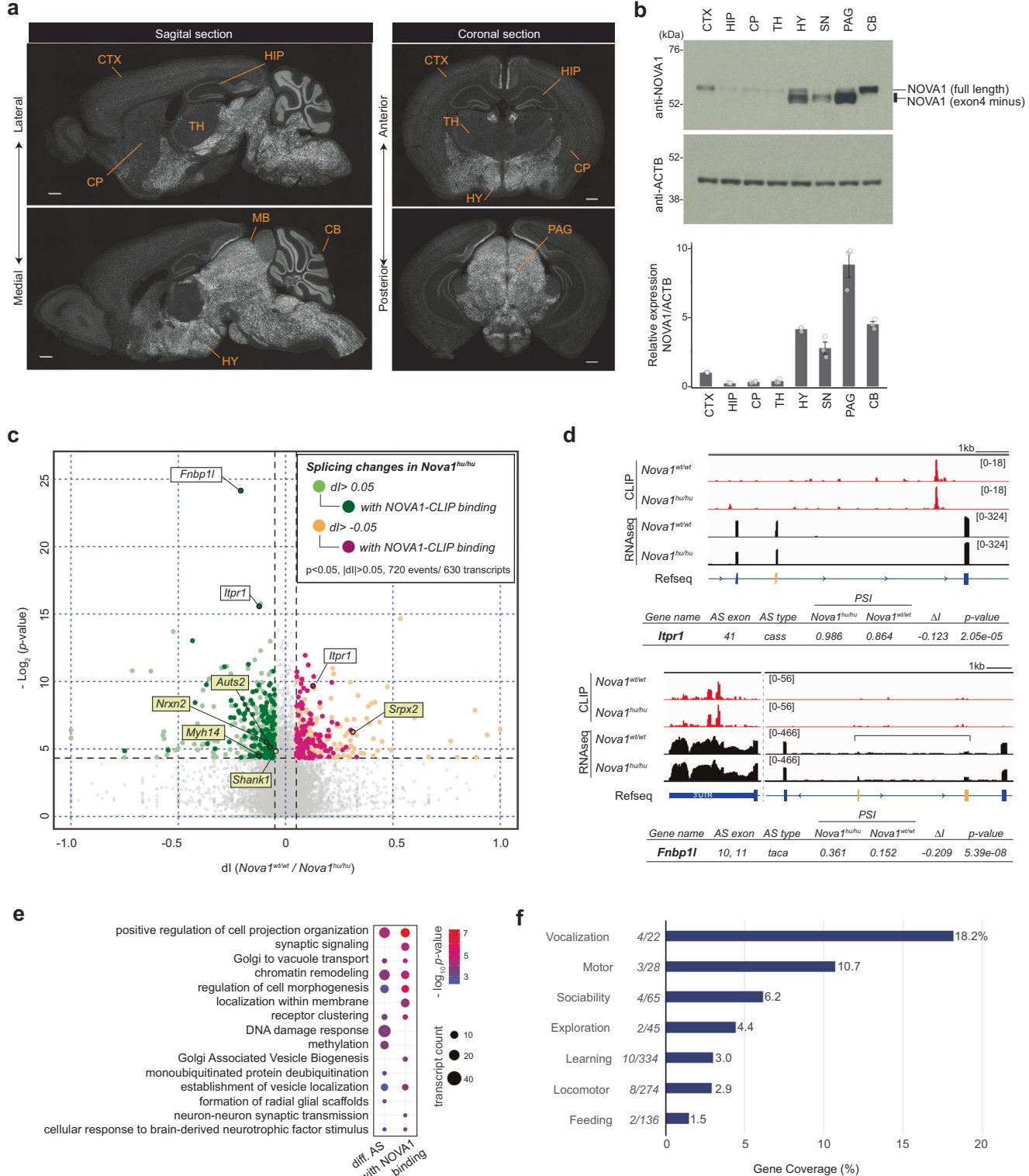

**Humanized NOVA1 mice pups exhibit altered vocalization**

We next examined whether *Nova1^{hu/hu}* mice with I197V substitution display changes in vocalization behavior. Prior studies in birds, fish, and mammals have shown that all vocal species have a conserved midbrain/brainstem vocal motor pathway[84–86]. Of particular importance in the vocal circuit in the mammalian brain is the periaqueductal gray (PAG), which plays a central role in the neural basis of primate vocal production[87]. The PAG projects to brainstem respiratory pre-motor and vocal motor nuclei, including the nucleus ambiguous

(Amb), which directly innervates the larynx[85,88]. Inhibition of PAG or Amb function leads to a loss of innate vocalization[89–91], while PAG stimulation induces vocalizations in both primates and mice[90,92], suggesting that the PAG and downstream brainstem circuits are essential for vocalization. We found that NOVA1 is highly expressed in the midbrain, particularly in the PAG and Amb (Fig. 3a-b, Supplementary Figs. 10–11). In addition to *Nova1^{hu/hu}*-specific actions on RNA splicing, we also found that many of the genes reported to be involved in vocal behavior (14 out of 22 transcripts) were also NOVA1 binding targets

**Fig. 3 | _Nova1[hu/hu]_ mice exhibit alternative splicing changes in specific neuronal genes. a** NOVA1 immunostaining in P21 mouse brain. Scale bars represent 500 μm. Abbreviations: CTX cortex, HIP hippocampus, CP Caudate putamen, TH thalamus, HY hypothalamus, MB midbrain, CB cerebellum, SN substantia nigra, PAG periaqueductal gray. **b** Western blotting analysis of NOVA1 in dissected mouse brain regions. The corresponding isoforms for each band is listed on the right. Experiments were repeated three times independently (Supplemental Fig. 11). Each lane's NOVA1 band is quantified and normalized to the ACTB band signal. The cortex value is set at 1. Data are presented as mean values +/- SEM. Source data are provided as a Source Data file. **c** Alternative splicing (AS) changes in the midbrain of 3-week-old _Nova1[hu/hu]_ mice. Differential AS events (_p_-value and delta PSI (dI) > 0.05 for _Nova1[hu/hu]_ vs. _Nova1[wt/wt]_ are shown in light green, and those having NOVA1-CLIP peaks are shown in dark green. Events with dI < −0.05 (in _Nova1[hu/hu]_ vs. _Nova1[wt/wt]_) events are shown in light orange, with NOVA1-CLIP peaks in magenta.

Representative AS events are labeled with gene names. Differential AS events in vocal behavior related genes are highlighted with a yellow box (Supplementary Fig. 12). **d** Examples of transcripts showing significant AS changes, with NOVA1-CLIP peaks on each transcript. Differential AS exons are colored in yellow. Each IGV snapshot is annotated with gene name, AS exon number, AS type, percent splice-in value (PSI), percent change (ΔI, ΔPSI) and _p_-value. AS events are classified as cassette exon (cass), alternative 5' splice site (alt5), alternative 3' splice site (alt3), tandem cassette (taca), mutually exclusive exons (mutx). **c, d** Statistical tests are performed using Quantas pipeline[125]. _N_ = 4 per group. **e** Gene annotation analysis by Metascape[127] of transcripts with differential AS events. Expressed genes in the P21 midbrain were used as background for this analysis. **f** Percentage of genes with differential AS changes in each behavior-related gene ontology category. The number of transcripts with differential AS events in _Nova1[hu/hu]_ is shown relative to the total number of genes in each category.

(Supplementary Fig. 14). These data strengthen the potential relationship between _Nova1[hu/hu]_ and vocalization, suggesting that vocalization studies in these mice would be valuable.

We first compared vocalizations from pups of _Nova1[hu/hu]_, _Nova1[hu/wt]_ (heterozygous) and _Nova1[wt/wt]_ mice. When pups are isolated from their mothers, they produce isolation-induced ultrasonic vocalizations (USVs), which are distress calls that attract their mother[93–95]. USVs were recorded from 7-day-old mice pups for 5 min in a dark sound isolation chamber (Fig. 4a, Supplementary Data 6, Supplementary Data 7). Following a previous protocol[96], we classified the syllables into four types (simple [s], upward [u], downward [d], multiple [m]), based on the direction and number of pitch jumps that separate notes within a syllable. Each syllable was analyzed for peak frequency measures (frequency value of start: $Fq_{start}$, minimum: $Fq_{min}$, maximum: $Fq_{max}$, end: $Fq_{end}$, mean: $Fq_{mean}$), variance ($Fq_{variance}$), bandwidth, duration, amplitude and purity (Fig. 4b).

The average number of USVs was $77.9 \pm 6.0$ per minute in _Nova1[wt/wt]_ and $62.7 \pm 6.4$ per minute in _Nova1[hu/hu]_ (mean ± standard error), with no significant difference between genotypes (Supplementary Data 8). However, several changes in USV features, which are thought to be important for mouse vocal communication[84], were observed. The "s" syllable (without pitch jumps), the most abundant syllable type, showed a trend of increased percentage within the total syllables in _Nova1[hu/hu]_ pups, while its amplitude was significantly lower in _Nova1[hu/hu]_ compared to _Nova1[wt/wt]_ pups. The amplitude of syllables containing pitch jumps ("u", "d" and "m") did not differ between genotypes, but the percentages of these syllables were decreased in _Nova1[hu/hu]_ pups, particularly for the "u" and "m" syllables (Supplementary Fig. 15a, Supplementary Data 8).

Pup isolation-induced USVs in mice show a bimodal distribution in their peak frequency (Fq), especially at 5-9 days of age, before consolidating to a single peak as they grow[97]. In our study, 7-day-old pups exhibited a clear bimodal distribution in the Fq parameter regardless of the genotype (Supplementary Fig. 15b). We assessed bimodality in $Fq_{max}$, showing clear separation using an Ashman's D score[98]. This analysis indicated distinct separation into two peaks for the "s", "d" and "m" syllables ($D > 2.0$), but not for the "u" syllable ($D = 1.603$) (Fig. 4c). To separate these peaks, we fitted two Gaussians[99] for each syllable type showing bimodal distribution, comparing USV characteristics for each peak (Fig. 4c, Supplementary Data 9). Individual syllables were classified into those with 'low' or 'high' $Fq_{max}$ according to the cutoff value (the intersection point of each distribution). In the jump syllables of _Nova1[hu/hu]_ pups, the proportion of low $Fq_{max}$ decreased, while the proportion of high $Fq_{max}$ increased (Fig. 4d, Supplementary Data 10). We also tested the bimodality and syllable ratios with $Fq_{min}$ and confirmed the same trend (increased ratio in high Fq in _Nova1[hu/hu]_ pups). Heterozygous _Nova1[wt/hu]_ pups showed intermediate values between _Nova1[hu/hu]_ and _Nova1[wt/wt]_ pups for these parameters, suggesting that the effect of the I197V substitution in NOVA1 protein on pup USVs is dosage-

dependent. These observations demonstrate distinct changes in the vocalizations of _Nova1[hu/hu]_ pups.

The vocalizations of pups isolated from the nest are known to influence maternal behavior in mice[95,100,101]. To explore whether the vocal motor changes in _Nova1[hu/hu]_ pups on their mothers, we tested if mother mice were more attracted to recorded vocalizations from each genotype. We set up an experiment using a three-room box connected by passageways, with speakers placed in each of the two end rooms (Supplementary Fig. 16a). Neonatal vocal recordings (from _Nova1[hu/hu]_ mice or _Nova1[wt/wt]_ control mice) were played from each speaker, and either a _Nova1[hu/hu]_ or a _Nova1[wt/wt]_ mother mouse was placed in the center room (Supplementary Fig. 16a–c). Regardless of the mother's genotype or the neonatal vocal recordings, no significant differences were observed in the mother's orientation toward the vocalizations (Supplementary Fig. 16d, e). Thus, changes in vocal quality in _Nova1[hu/hu]_ pups had no impact on the behavior of the mother mice in this assay.

## Humanized NOVA1 adult mice have altered vocalization

Adult mouse USVs are often produced in long, continuous sequences of the same four major syllable types (Fig. 4b), particularly during courtship[97,102,103]. We explored whether changes in courtship vocalization behavior would occur in _Nova1[hu/hu]_ mice, using a previously established paradigm[96,103,104]. Courtship USVs were elicited from adult male mice by exposing them to adult female mice in estrus (Fig. 4e, Supplementary Data 11, Supplementary Data 12). In this context, more than 90% of the vocalizations came from the males[105]. The average number of courtship-induced USVs did not differ between genotypes: $144.8 \pm 24.0$ per minute in _Nova1[wt/wt]_ and $165.5 \pm 27.0$ per minute in _Nova1[hu/hu]_ (mean ± standard error) (Supplementary Data 13).

We next examined short-duration and long-duration syllables[18,106], which have been previously shown to exhibit a bimodal distribution, particularly for "s" syllables[19]. Two Gaussian distributions were fitted to characterize each class of USVs (Fig. 4f). The cutoff duration (the intersection point of each distribution) between short and long USVs was determined to be 44 ms, with 29% of "s" syllables categorized as long duration. The average duration of short- and long-"s" syllables was $22.6 \pm 9.1$ ms and $69.0 \pm 40.1$ ms, respectively. _Nova1[hu/hu]_ mice tended to have a lower frequency characteristic of long-duration "s" syllables, where the differences for $Fq_{start}$, $Fq_{end}$ and $Fq_{mean}$ were significantly lower compared to controls (Fig. 4g, Supplementary Data 14). These changes were not observed in pup isolation-induced USVs, indicating that this effect is developmentally specific and/or context-dependent.

We next examined peak frequencies. In adult USVs, the $Fq_{max}$ parameter exhibited a unique distribution. High $Fq_{max}$ syllables were those that exceeded 100 kHz, and constituted a distinct population from the main population (Supplementary Fig. 15c, d). Syllables with High $Fq_{max}$ of wild-type animals are often observed in harmonic syllables, containing pitch jumps and some simple syllables[97,107] (Supplementary Fig. 15e). Fitting two Gaussian distributions to the data, the

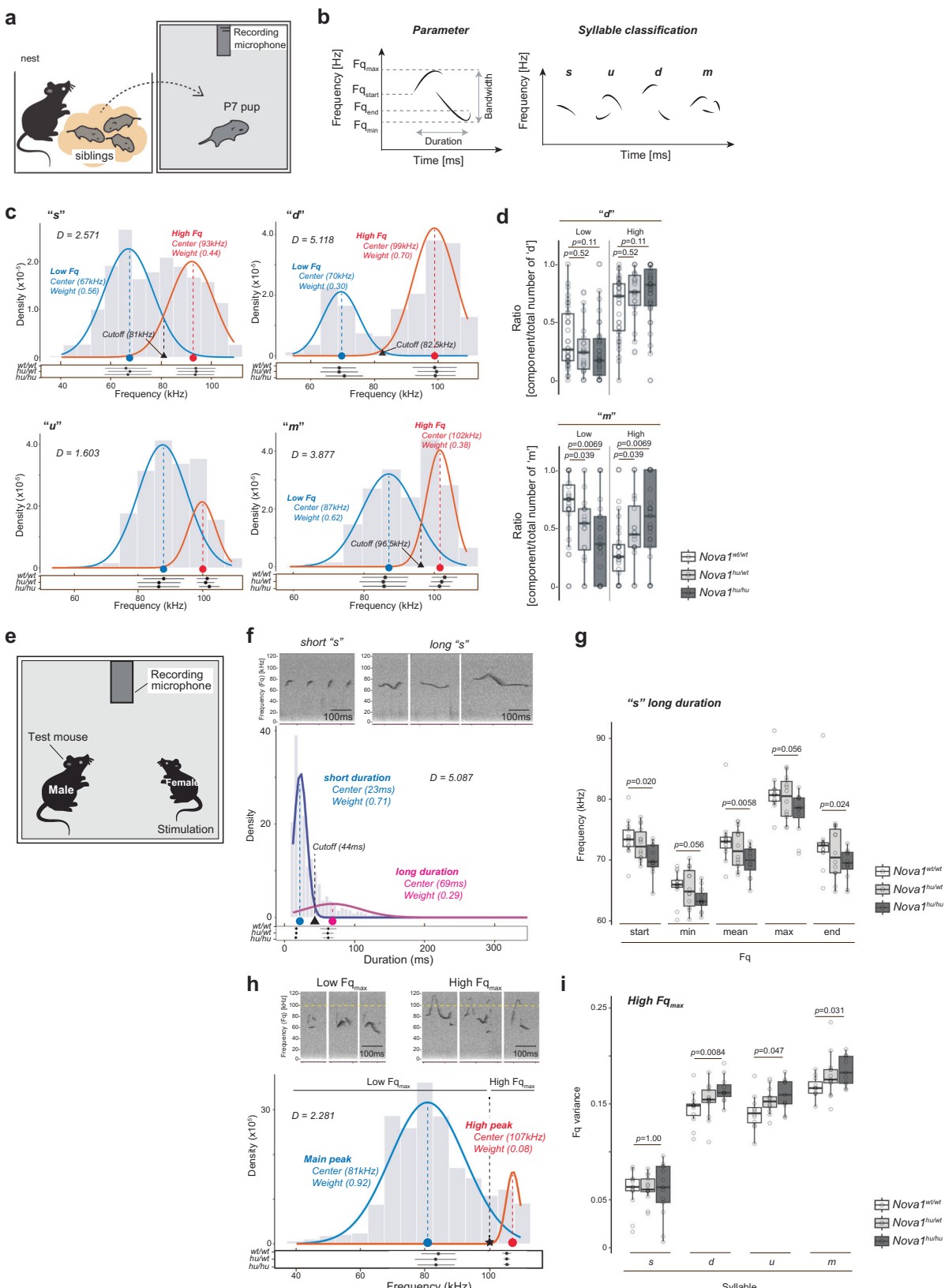

High Fq$_{max}$ was found to be 107.1 ± 2.1 kHz, whereas the main peak was 80.8 ± 11.5 kHz (mean ± standard deviation) (Fig. 4h). High Fq$_{max}$ syllables, as defined, accounted for roughly 8.5 % of total USVs observed. There were no significant differences in syllable composition with High Fq$_{max}$ among genotypes (Supplementary Data 15). However, the Fq$_{variance}$ of High Fq$_{max}$ syllables containing pitch jumps ("d", "u" and

"m") was significantly greater in *Nova1*$^{hu/hu}$ mice (Fig. 4i, Supplementary Data 16). Fq$_{variance}$ tends to increase with syllable complexity[108], with simple syllables like "s" having lower values and more complex syllables like "m" with multiple jumps having higher values. This suggests that *Nova1*$^{hu/hu}$ mice produce more complex high-frequency USVs than *Nova1*$^{wt/wt}$ mice. These findings demonstrate that vocal behavior is

**Fig. 4 | *Nova1*[hu/hu] mice exhibit altered vocal patterns. a** Isolation-induced ultrasonic vocalization (USV) test in pups. **b** USV parameters and syllable classification. **c** $Fq_{max}$ distribution and two-Gaussian fit for pup USVs. Ashman's D score (a measure of separation of two distributions, where a score above 2 indicates good separation) is shown. Each Gaussian center and weight are labeled, with the intercept of the two Gaussian distributions (black triangle) used as the cutoff between high and low $Fq_{max}$ USVs. **d** Ratio of high or low $Fq_{max}$ in syllables "d" and "m". The ratio of syllables belonging to each distribution (high or low) is calculated for the total number of each syllable type. **e** Courtship-induced USV test for adult mice. **f** Duration distribution and two-Gaussian fit for syllable "s" in adult USVs. Ashman's D score, Gaussian center, and weight are shown. The intercept of the two Gaussian distributions (black triangle) was used as the cutoff between long and short duration. Short and long "s" syllable examples are shown at the top of the plot. **g** Peak frequency parameters in long duration "s". **h** $Fq_{max}$ distribution and two-Gaussian

fit in adult USVs. Ashman's D score, Gaussian center and weight are labeled. The black star marks the 100 kHz cutoff between high and low $Fq_{max}$. Examples of low and high $Fq_{max}$ syllables are shown at the top of the plot. **i** Frequency variance (Fq variance) in high $Fq_{max}$ in adult USVs. The dot plots for (**c**, **f**, **h**) at the bottom of the density plots show the mean (black dots) and standard deviation (whiskers) by peak for each genotype. No significant differences were observed between genotypes. Boxplots for (**d**, **g**, **i**) represent the minimum, first (lower) quartile, median, third (upper) quartile, and maximum values. For pup data (**c**, **d**), each circle represents data from a single pup: *Nova1*[hu/hu] $N = 41$, *Nova1*[hu/wt] $N = 23$, *Nova1*[wt/wt] $N = 40$ pups. For adult data (**f–i**), three experiments were conducted over consecutive weeks, and the average value for each mouse is plotted (white circles): *Nova1*[hu/hu] $N = 13$, *Nova1*[hu/wt] $N = 14$, *Nova1*[wt/wt] $N = 13$ adults. Statistical analysis was performed by Wilcoxon rank sum tests (two-sided, Bonferroni correction).

altered in both pups and adults in *Nova1*[hu/hu] mice, resulting in unique vocal characteristics.

## Discussion

In line with studies of genetic variants that have played a role in the evolution of modern humans[19,109,110], we investigated the biological effect of a single amino acid substitution, I197V in NOVA1, which is unique to modern humans. By analyzing *Nova1*[hu/hu] mice carrying this allele, we identified molecular changes in alternative splicing in the brain, including brain regions associated with vocal behavior, and identified changes in vocalization patterns in pups and adult mice. These findings suggest that during human evolution, the I197V substitution in NOVA1 protein may have contributed to the development of neural systems involved in more complex vocal communication.

The importance of NOVA1 in mammals is evident from the lethal phenotype of *Nova1* knockout mice[31] and the neurological symptoms caused by *NOVA1* haploinsufficiency in humans[36]. This significance is further highlighted by the high conservation of the NOVA1 protein in mammals. Interestingly, the *NOVA1* gene harbors an Ultra Conserved Element (UCE; uc.359) at the end of the 3' UTR[111,112] with additional high conservation extending upstream from the *NOVA1* UCE to most of the 3' UTR and terminal exon encoding NOVA1 KH2 and KH3 domains. This underscores the unique nature of the I197V variant, which occurred within a region of the genome resistant to change.

Previous studies have confirmed the evolutionary restriction of NOVA1 variants, including the I197V variant (termed I200V in one study[13]). We support this analysis and have expanded upon it with larger human sequence datasets across diverse ethnic groups, and from methods that infer selection coefficients from ancient samples[48,49]. These results confirm that NOVA1 has undergone strong positive selection and that the I197V variant is part of an evolutionary selective sweep in the emergence of *Homo sapiens*.

The observation that the I197V NOVA1 allele is nearly fixed across human populations (Supplementary Fig. 1a) suggests that it emerged and increased in frequency well before the divergence of ancient human lineages. The earliest split among modern human groups—that of the San - is currently estimated to have mostly occurred by roughly 200 kya, well before migration of modern humans out of Africa and the Near East to Eurasia around 50kya[1,9,113]. Unlike more recent selective sweeps, such as the LCT locus, which is dated around 10 kya, and is population-specific, the NOVA1 variant is part of an older, more widespread sweep. These older sweeps, shared across modern human populations, may leave subtler genetic signatures that require novel detection methods. This suggests that the ancient NOVA1 selective sweep may represent part of a broader set of undiscovered ancient sweeps.

One possible explanation for the changes in vocal behavior observed in *Nova1*[hu/hu] mice could be molecular changes in midbrain and brainstem vocal pathways, which express high levels of NOVA1 and are involved in regulating innate vocalizations (USVs), including breath

coordination, timing, and amplitude[114–116]. An alternative possibility is that changes occurred in more recently evolved cortical vocal regions, which control pitch, frequency modulation, and duration (the Kuypers/ Jürgens hypothesis[84,117] and the volitional articulatory motor network[87,118]). Given that NOVA1 is expressed in the mouse cortex, predominantly in inhibitory neurons[61], it is plausible that the I197V substitution affects cortical regulation of vocalization.

Notably, *Nova1*[hu/hu] mice exhibit qualitative changes in vocal characteristics compared to control mice both in pups and adults, despite producing a similar number of calls. These findings suggest that the vocalization changes in *Nova1*[hu/hu] mice are not simply the result of alterations in general motor performance. This idea is supported by other observations showing that *Nova1*[hu/hu] mice perform similarly to control mice in motor function tests, such as the rotarod, and display comparable locomotion activity levels in the Y-maze test (Supplementary Fig. 17). Additionally, the Y-maze test results indicated that *Nova1*[hu/hu] mice had spatial working memory comparable to that of control mice.

The changes in vocalization in *Nova1*[hu/hu] mice varied in a development- or context-dependent manner. It has been reported that high-frequency (Fq) USVs are emitted more frequently by male adult mice during social interactions[119] and that female mice are attracted to male mice that emit more complex USVs[96]. Given these reports, the increased the proportion of higher Fq USVs in pups, and the increased complexity of these USVs in adults may potentially offer social advantages in mice. However, since auditory frequency resolution in mice has been reported to be limited[120] and our preference experiment using humanized NOVA1 pup USVs showed no significant preferences in the mother's responses, it remains unclear to what extent other mice can recognize these vocalization changes. It should be noted that we were unable to address the effect of the I197V substitution on the vocalizations of adult female mice, as our study focused on a courtship-induced vocalization paradigm that predominantly elicits USVs from male mice[121]. However, recent studies suggest that female mice also vocalize under certain experimental conditions or social contexts[21,122,123]. Future studies will be necessary to investigate the effects of the I197V substitution on USVs in female mice, as well as adult female preferences to USVs in adult *Nova1*[hu/hu] male mice.

Interestingly, the vocalization changes observed in *Nova1*[hu/hu] mice share some similarities with those observed in humanized *Foxp2* mice (with two human-specific substitutions). In both cases, the changes were developmental or context-dependent and included a decrease in peak frequency in simple syllables and modulation of high-frequency regions in complex syllables[19–21] (Supplementary Data 17). Conversely, male mice with a humanized *Foxp2* mutation produced simpler song bouts with more "s" syllables[17,18]. These observations may indicate a common or related molecular alteration in the neural circuits involved in the USV production between humanized *Nova1* mice and humanized *Foxp2* mice. Future studies should aim to identify the molecular and neural basis of these alterations, as well as the physiological

significance of these vocalization changes in the context of social behavior.

Our molecular analysis showed that the sequence-specific RNA binding of NOVA1 was unaffected by the human substitution and that steady-state gene expression levels in the brains of *Nova1*[hu/hu] mice were nearly identical to those of wild type mice. However, we detected alternative splicing changes in several transcripts associated with vocalization. The expression pattern of NOVA1 in the brain and the enrichment of its target transcripts to specific biological pathways support a link between NOVA1 function and vocal behavior. Uncovering the precise molecular mechanisms underlying the phenotypes in *Nova1*[hu/hu] mice will require further study of the neural circuits for vocalization, as well as on regulatory factors influencing NOVA protein function. This study sets the groundwork for understanding molecular mechanisms driving the evolution of human vocal communication.

Biochemically, NOVA proteins harbor three KH domains responsible for sequence-specific RNA-binding[41,44,45], with amino acid 197 located in the KH2 domain. Although the I197V substitution in NOVA1 alters the hydrophobic core of the KH domain, it does not lead to a loss of RNA-binding capacity or functional attenuation, in contrast to other KH domain point mutations[51–53]. This is supported by the unchanged global gene expression levels observed in the brains of *Nova1*[hu/hu] mice, while NOVA1 knockout mice (which exhibit postnatal lethality[31]) show significant expression changes of key neuronal genes in the midbrain at E18.5 (Supplementary Fig. 5e, f, Supplementary Data 18). KH domains harbor three alpha-helices (H) and three beta-sheets (S) (S1-H1-H2-S2-S3-H3; Supplementary Fig. 9a, b); the first and second alpha helices (H1-H2) determine single-stranded RNA binding specificity[44,45], and may also be involved in protein-protein dimerization[62] (Supplementary Fig. 9c, d). Protein structure predictions suggest that the addition of a single carbon atom in valine 197 extends its ability to interact with several nearby amino acids (185Ile in H1; 232Ala, 235Leu, 236Ile, 239Lys in H3), allowing the KH2 domain to gain contact with H1 and 239Lys in H3. Thus, while the I197V substitution does not change sequence specificity or binding affinity of NOVA1 with RNA (Fig. 2i, j), it may affect KH domain dimerization or may have undiscovered effects on protein-protein interactions[62]. Interestingly, the amino acid corresponding to NOVA1 amino acid 197 is also an isoleucine in the related proteins FMR1 and hnRNP E1/E2/K, but is a valine in both human and mouse NOVA2[44] (Supplementary Fig. 7). Functional differences between NOVA1 and NOVA2 in mice[43,61] may reflect structural and functional differences in their respective KH domains.

In summary, we analyzed a single amino acid unique to modern humans in the RNA binding protein NOVA1 and examined its biological effects in vivo by introducing this amino acid in mice. NOVA1 is highly intolerant to changes in amino acid sequences during evolution with the exception of this single amino acid change in humans. We propose that this change was part of an evolutionary sweep associated with specific changes in the neuronal transcriptome and vocal communication.

## Methods
### Ethical approval
All procedures were performed according to the guidelines of the Institutional Animal Care and Use Committee (IACUC) under the IACUC protocol # 23014 at the Rockefeller University.

### Animal experiments
C57BL/6J (stock no. 000664) mice were obtained from the Jackson Lab. *Nova1*[hu/hu] mice generated in this study were backcrossed to C57BL/6J strain at least 8 times. The mice were housed in individually ventilated cages (five per cage) under conditions of a 12 h light/dark cycle and ambient temperature of $21 \pm 4\,°C$ with 40–70% humidity. Male or female mice aged 7 days (for isolation induced pup USV test) and 8–20 weeks (for playback behavioral experiment and courtship-

induced adult USV test) were used for animal experiments, as described. Littermates of the same sex were randomly assigned to experimental groups.

### Generation of *Nova1*[hu/hu] mice
*Nova1*[hu/hu] mice were generated by directly injecting the sgRNA/Cas9 RNP with a single-stranded repair template DNA (ssDNA) into C57BL6 zygote to substitute isoleucine to valine at amino acid 197 of mouse NOVA1. gRNA and the ssDNA were designed as follows. gRNA (5'-TGCTACTGTGAAGGCTATAA-3'): overlapping the DNA sequence (mm10/ chr12: 46,700,902–46,700,904) of the mouse *Nova1* genomic locus encoding the 197th amino acid of NOVA1. ssDNA: 140 nt length DNA homologous to the NOVA1 locus with a nucleotide substitution (A to G) to cause an amino acid change from isoleucine to valine at the 197th position. Two silent mutations were also designed to create *BtsaI* restriction enzyme recognition site for genotyping.

Genomic DNA was extracted from the tail of the F0 animals, and the DNA corresponding to the area around the 197th amino acid was amplified by PCR and subsequently cloned into a plasmid for determining the sequence of the modified allele. Genomic sequence analysis revealed that among 13 F0 animals, 8 animals harbored the designed allele (with three nucleotide substitutions: one causing I197V amino acid substitution, two for restriction enzyme recognition site for genotyping (not causing amino acid changes)). Animals carrying the designed humanized Nova1 allele were crossed to the wild-type C57/BL6 mice, and this process was continuously repeated for subsequent generations to eliminate possible off-target mutations.

Genome sequencing analyses were performed on the possible off-target sites of the gRNA used (10 potential off-target loci with mismatches outside of the PAM+12mer core sequences; predicted by CRISPR direct: https://crispr.dbcls.jp/) to compare sequences of control (wild type) and humanized NOVA1 mouse. For all the potential off-target sites, the sequence in the humanized NOVA1 mice was identical to control mice and the reference genome, with I197V substitutions being the only detectable edits (Supplementary Data 4d, e).

For routine genotyping, sequences around the genomic DNA encoding the 197th amino acid were amplified by PCR subsequently digested with the *BtsaI* restriction enzyme. Each Mouse genotype; wild type (*Nova1*[wt/wt]), humanized *Nova1* homozygous (*Nova1*[hu/hu]), heterozygous (*Nova1*[hu/wt]) was determined by band size obtained by electrophoresis. Siblings obtained by crossing heterozygous parents were used in the experiment. Primers used for typing are shown below.

NOVA1hu-Fwd: 5'- ccctctttttgacatgctggt -3'
NOVA1hu-Rvs: 5'- cataaggagatccggttgga -3'

DNA band size after restriction enzyme treatment: wild type (613 bp), homozygous (389 bp and 224 bp), heterozygous (613 bp/ 389 bp + 224 bp) (see Supplementary Fig. 3c).

### Antibodies
Primary antibodies used for immunohistochemistry and western blotting were as follows; rabbit anti-NOVA1 (1/1000 dilution) [EPR13847] (ab183024, abcam), rabbit anti-NOVA1 C-terminal (1/1000 dilution) [EPR13848] (ab183723, abcam), human anti-pan NOVA (1/10,000 dilution) (anti-Nova paraneoplastic human serum) and rabbit anti-ATCB (1/10,000 dilution) (ab8227, abcam).

### Immunohistochemistry
Postnatal day 0, 3 or 12-week-old mice were perfused with PBS and 4% paraformaldehyde (PFA), and the brain was dissected. Dissected brain was further fixed. Overnight by 4% PFA at 4 °C. The solution was sequentially replaced with 15% sucrose/ PBS and 30% sucrose/ PBS, embedded with OCT compound, and stored at −80 °C until use. Frozen brains were sliced into 30–50 μm thick sections in a cryostat (CM3050S, LEICA). Slices were washed three times with PBS at room temperature (RT), incubated in 0.2% Triton X-100/PBS for 15 min at RT,

blocked in 1.5% normal donkey serum (NDS)/PBS for 1 h at RT, incubated overnight at 4 °C with primary antibody in 1.5% NDS/PBS, then incubated in Alexa Incubated with 488, 555 or 647 conjugated donkey secondary antibody. The nuclei were stained using 4',6-diamidino-2-phenylindole (DAPI) solution (1 μg/ml). Images of specimens were collected with a BZ-X700 (KEYENCE) microscope.

## Western blotting

Each dissected brain region (cortex, midbrain and cerebellum) of P21 mouse brains were lysed in RIPA buffer (50 mM Tris-HCl; 150 mM NaCl; 0.1% SDS; 0.5% sodium deoxycholate; 1% NP-40). Extracts were separated by SDS-PAGE, and subjected to immunoblotting using the antibodies described above. Quantification of western blots was done with ImageJ (v1.53). Each band signal was quantified and normalized with ACTB signal to control for differences in loading.

## Electrophoretic Mobility Shift Assay (EMSA)

**Protein purification.** The genes encoding each NOVA1 protein (NOVA1$^{wt}$ and NOVA1$^{hu}$) were cloned into the pGEX6p1 vector and expressed in E. coli BL21 strain. N-terminally GST (Glutathione S-Transferase) fused NOVA1 was induced by the addition of IPTG (final conc. 0.1 mM) for 4 hr. Pelleted cells were sonicated, and then incubated in the presence of Triton-X (final conc. 1%, 30 min). Cleared supernatant was collected after centrifugation (12,000 x $g$ 10 min 4 °C). After incubating with Glutathione Sepharose beads (GE Healthcare Biosciences, 17075601) for 30 minutes, the mixture was washed three times with PBS. The GST tag was cleaved from the NOVA1 protein by PreScission Protease treatment (GE Healthcare, 27-0843-01; 50 mM Tris-HCl, 150 mM NaCl, 1 mM EDTA, 1 mM DTT, pH 7.5, 4 °C for 4 h) to obtain purified NOVA1 protein. The concentration of each purified NOVA1 protein was determined by SDS-PAGE followed by GelCode Blue staining (Thermo Fisher Scientific, PI24590) using BSA (Sigma-Aldrich, B8667) as standards.

**Single strand RNA probe preparation.** The single-stranded RNA probe was designed as previously[41]. The following single strand RNA were synthesized by IDT:

CCTTATCATGCTGACTCACGTCATTTCATCTCATCAAGGGAGT CAGTGGGATA

Synthesized RNA was first incubated at 80 °C for 10 min, then rapidly incubated on ice, mixed with [gamma-32P] ATP (3000 Ci/mmol, 10 mCi/ml) (Revvity, BLU502A), and labeled at the 5' end by T4 polynucleotide kinase treatment (New England BioLabs, M0201S). The labeled probes were purified by G-25 column (VWR, 95017-621) and diluted to the appropriate concentration with water.

**Binding assay.** Purified NOVA1 protein (0.5–2 pmol/reaction) and labeled RNA probe (0.08 pmol/reaction) were mixed under the following buffer conditions: 10 mM HEPES, 3 mM MgCl2, 150 mM NaCl, 5% Glycerol, 0.1 mM DTT, 0.1U/μl RNase OUT (Thermo Fisher Scientific, 10777019)., 10 mg/ml Yeast tRNA (Thermo Fisher Scientific, AM7119), 1 mg/ml poly dI-dC (Thermo Fisher Scientific, 20148E). The binding reaction was performed at 22 °C for 60 min, and the reaction solution was mixed with loading dye and run on an 8% acrylamide native gel at 150 V for 3 h to separate RNA-protein binding. After the gel electrophoresis, the gels were dried by a gel dryer and autoradiographs were detected. Quantification of each signal was performed by ImageJ, and the Kd value of each NOVA1 protein for the RNA probe was calculated using Prism software (https://www.graphpad.com/features).

## RNA-seq library preparation and analysis

For RNA-seq, samples included dissected cortex, midbrain and cerebellum at P21, or dissected midbrain at E18.5 of Nova1$^{hu/hu}$ and Nova1$^{wt/wt}$ mice. The mRNA-seq library was prepared from RNA extracted with Trizol following the Illumina TruSeq protocol of polyA selection,

fragmentation, and adapter ligation. Multiplex libraries were sequenced as 125 nt paired-end runs on the HiSeq-2500 platform at Rockefeller University Genomic Core. These raw datasets and processed data files have been deposited with Gene Expression Omnibus (GSE253297).

## Cross-linking Immunoprecipitation (CLIP)

NOVA1-CLIP was performed in P21 dissected cortex, midbrain and cerebellum of Nova1$^{hu/hu}$ and Nova1$^{wt/wt}$ using each three biological replicates. Tissues were dissected in PBS, triturated using 20 G needle and crosslinked three times on ice for 400 mJ/cm2 using Stratalinker. Crosslinked material was collected by centrifugation, resuspended in wash buffer (1X PBS, 0.5% NP-40, 0.5% deoxycholate and 0.1% SDS with protease inhibitor), and subjected to DNase (RQ1 DNase: Promega) and RNase (RNase A: Affymetrix) treatment at a final dilution of 1:20,000 for 5 min. The lysate was clarified by centrifugation at 20,000 × $g$ for 20 min. The supernatant was used for immunoprecipitation with 200 μL of Protein A Dynabeads (Invitrogen) loaded with 18 μg anti-Nova1 antibody (abcam) for 2 h at 4 °C. The samples were washed as follows: twice with wash buffer, twice with Nelson stringent wash buffer (15 mM Tris pH 7.4, 5 mM EDTA, 2.5 mM EGTA, 1% Triton X-100, 1% Sodium deoxycholate, 0.1% SDS, 120 mM NaCl, 25 mM KCl), twice with Nelson high salt buffer (15 mM Tris pH 7.4, 5 mM EDTA, 2.5 mM EGTA, 1% Triton X-100, 1% Sodium deoxycholate, 0.1% SDS, 1 M NaCl), twice with Nelson low salt buffer (15 mM Tris pH 7.4, 5 mM EDTA), and twice with PNK wash buffer (50 mM Tris pH 7.4, 10 mM MgCl2, 0.5% NP-40). RNA fragments were dephosphorylated using FastAP Alkaline phosphatase (Thermo Fisher Scientific) and subjected to 3' ligation overnight at 16 °C with a pre-adenylated linker (preA-L32) using truncated KQ T4 RNA Ligase2 (NEB). The RNA-protein complexes were labeled with $^{32}$P-γ-ATP using T4 PNK (NEB) and subjected to SDS-PAGE and transfer to nitrocellulose membrane. Appropriate regions of the membrane were cut out and RNA was extracted according to the following conditions: 100 mM Tris PH7.5, 50 mM NaCl, 10 mM EDTA, 7 M Urea with proteinase K. RNA was purified by phenol-chloroform extraction method. Cloning was performed using the BrdU-CLIP protocol. Briefly, the reverse transcription reaction was performed using Superscript III (Thermo Fisher Scientific), and the cDNA was BrdU-labeled by including BrdU in the reaction solution. Immunoprecipitation was performed with 5 μg anti-BrdU antibody (abcam) and 25 μg protein G Dynabeads per reaction (45 min at room temperature), followed by washing with the following solutions (including Denhardt's solution): once with IP buffer (0.3x SSPE, 1 mM EDTA, 0.05% Tween 20), twice with Nelson low salt buffer, twice with Nelson stringent wash buffer, twice with IP buffer. After eluting the cDNA, BrdU-immunoprecipitation was performed again under the same conditions. cDNA was circularized on beads using CircLigase II (Epicentre) and PCR was performed using Accuprime Pfx supermix (Thermo Fisher Scientific) and Syber Green until RFU 250–500. PCR products were purified using Agencourt AMPure XP (Beckman Coulter) and concentrations were measured by TapeStation. High-throughput sequencing was performed at the Rockefeller University Genome Resource Center. These raw datasets and processed data files have been deposited with Gene Expression Omnibus (GSE253296).

## Bioinformatics

Paired-end reads from RNA sequencing were aligned to the mouse genome (mm10) builds of the mouse genome using OLego (v1.1.7) (https://zhanglab.c2b2.columbia.edu/index.php/OLego)[124]. Mapped reads were counted using gapless (for inference of transcript structure) and countit (for quantification of gene expression and alternative splicing) in Quantas (v1.0.9) (https://zhanglab.c2b2.columbia.edu/index.php/Quantas)[125]. All reads mapping to transcripts were included in the differential expression analysis using edgeR[126]. The data set for Nova1 knockout mouse (E18.5 midbrain) was kindly provided by Dr.

Yuhki Saito. The data are available from GEO submission GSE69711. Data visualizations were done using R (v4.2.0). Correlation matrix was visualized using corrplot package. PCA analysis was performed using FactoMineR and factoextra packages and visualized using ggplot2 package. Sequencing tracks were visualized using Integrative Genomic Viewer (IGV, v2.13.0).

Quantification of splicing for annotated cassette exons was performed using the Quantas pipeline (v1.0.9)[125]. In brief, the inclusion level of each cassette exon (percent-spliced-in: PSI) was calculated from the number of supporting exon junction reads for the inclusion and skipping isoforms. Only quantifications with ≥20 supporting junction reads were used for downstream analysis. Gene annotation analysis was performed using Metascape (v3.5.20240101)[127]. The expressed genes in the P21 midbrain (filtering lowly expressed genes by edgeR) were set as background for the analysis. Attribution for genes was performed using Gene Ontology (GO) resource in MGI (6.24) (https://www.informatics.jax.org/).

CLIP reads were processed using the CLIP Tool Kit (CTK, v1.1.3) as described previously[128]. Briefly, raw reads were filtered for quality and demultiplexed using indexes introduced during the reverse transcription reaction. PCR duplicates were collapsed and adapter sequences removed. Reads were mapped to the mm10 build of mouse genome using novoalign (v3.09.02) (www.novocraft.com). Mapped reads were further collapsed for potential PCR duplicates by coordinates and taking into consideration the degenerate barcodes introduced during the reverse transcription. Only unique CLIP tags were used for subsequent analyses. We performed three biological replicates per sample. All scripts used in the analysis including the peak finding algorithm and more information can be publicly obtained at (https://zhanglab.c2b2.columbia.edu/index.php/Standard/BrdU-CLIP_data_analysis_using_CTK). De novo motif analysis and motif density analysis were done using findMotifsGenome.pl and annotatePeaks.pl commands in HOMER (v4.11).

## Ultrasonic vocalization (USV) tests

**Isolation induced pup USV test.** To elicit isolation-induced USV, 7-day-old pups were isolated from their mother and littermates. Each pup was placed quietly on a small open-faced plastic plate in the sound attenuating chamber (15″ × 24″ × 12″ Igloo® beach cooler with a tube for pumped air circulation input, no light). An ultrasonic microphone was suspended a small distance from the pup, and the USVs were recorded for 5 min. Between trials, the recording box was cleaned with 70% alcohol and distilled water, and allowed to fully dry before the next experiments. Vocalizations were recorded with UltraSoundGateCM16/CMPA ultrasonic microphones connected to an Ultrasound Gate USGH amplifier. Recordings were saved using the AvisoftRecorderUSG software (Sampling frequency: 250 kHz; FFT-length: 1024 points; 16-bits). All acoustic hardware was obtained from Avisoft Bioacoustics® (Berlin, Germany).

**Playback behavioral experiment.** Mothers rearing 7-day-old offspring were used in the playback experiment. We used a three-chamber box (12″ × 23.5″ × 15.5″) connected by a passageway through which a mouse could pass for the test. Each chamber at both ends was equipped with a speaker (Vifa ultrasonic speaker, Avisoft Bioacoustics), and a camera (Firefly S, FFY-U3-16S2C-S, FLIR) was placed on the ceiling of the chamber to record the behavior of the mouse. The speakers were connected to an UltraSoundGate Player 216H (Avisoft Bioacoustics), using Avisoft Recorder USGH and had a frequency range (±12 dB as the maximum deviation from the average sound volume) of 25–125 kHz. We adjusted the loudness between the channels by controlling the level of the peak power before the experiment. Using two microphones, we made sure that both songs were audible at the entrance of both rooms so that the mother can respond to the songs but not loud enough that the microphones could detect the song

being played in the other room. After the 10 min habituation period, playbacks were triggered when the mouse broke an infrared sensor located in the center of the three-chamber box. One speaker on one side played one pup-USV recording and the other speaker simultaneously played another pup-USV recording both of which were previously recorded during the pup isolation induced USV test for 5 min.

Pup-USV recording was prepared in Audacity® by stitching vocalizations from 4-5 pups for each genotype. These recording files contained an equivalent number of pup-USVs (*Nova1*[wt/wt] 1689 USVs, *Nova1*[hu/hu] 1561 USVs) and were confirmed to reflect the vocal characteristics of each genotype. After the first 5 min playback session, a second 5 min playback session was conducted after 1 min quiet period. In the second playback experiment, the two recordings playing from the speakers were switched to eliminate the possible preference by the location. During each 5 min period, the mother was allowed to explore freely in the box and the time she spent in each room was counted. The box was cleaned between experiments with 70% alcohol and distilled water and allowed to fully dry before the next experiments.

**Courtship induced adult USV test.** The protocol for the courtship-induced vocalization test in adult male mice has been performed according to previous studies[96] with minor modifications. Briefly, adult male mice (8–12 weeks old) were sexually socialized by spending one night with a sexually mature female to enhance the male's motivational state to exhibit courtship USV behavior[108]. On the next day, female mouse was removed from the cage, and the male mice housed in the same cage until the test day. On the day of the recording, the males were placed in a new cage and then singly habituated in the sound recording environment (as described for pup USV test) for 15 min. The males were then exposed to adult female mice for 5 min. We used the females (8-12 weeks old) in estrus (selected visually for wide vaginal opening and pink surrounding). The test was conducted three times per mouse, one week apart, and a different female mouse was used as the stimulus each time to avoid familiarity effects. The order of mice tested each time was shuffled to avoid the possible order effects. Between trials, the mouse cage was cleaned with 70% alcohol and distilled water, and allowed to fully dry before the next experiments.

**USV analysis.** Acoustic waveforms were processed using a custom Python program called "Mouse Song Analyzer 2" (MSA2): available on the website (https://github.com/Neurogenetics-Jarvis/MSA2 [96,103,108,129], and analysis as previously described[96,103,108]. Briefly, the software computed the sonograms from each waveform, threshold to eliminate the white noise component of the signal, and truncated for frequencies outside the USV song range (35–125 kHz). We used a criterion of 10 ms minimum to separate two syllables and 3 ms as the minimum duration of a syllable. The identified syllables were then classified by presence or absence of instantaneous "pitch jumps" separating notes within a syllable into four categories: (1) simple syllable without any pitch jumps ("s"); (2) complex syllables containing two notes separated by a single upward ("u") or (3) downward ("d") pitch jump; and (4) more complex syllables containing a series of multiple pitch jumps (type "m"). Any sounds the software could not classify were put into "not identified: notIDd" category. The following spectral features were calculated automatically by MSA2 from the sonograms of each of the classified syllable types: Syllable duration, inter-syllable interval, standard deviation of pitch distribution, pitch (mean frequency), frequency modulation, spectral purity, and bandwidth. The fitting of Gaussian in the duration and peak frequency maximum ($Fq_{max}$) distributions in pup isolation-induced USVs and in adult courtship USVs were performed using R package mixtools: tools for analyzing finite mixture models[99]. The bimodality of the USV duration distributions was assessed using Ashman's D Score[98], where μ and σ are the center and standard deviation of each Gaussian, respectively. Cutoff $Fq_{max}$ in pup USVs between low and high USVs were defined as the intercept of the

two Gaussian fits to the distribution to the nearest frequency (kHz). Cutoff durations between short and long USVs were defined as the intercept of the two Gaussian fits to the USV duration distribution to the nearest millisecond. Statistical analysis was performed by pairwise Wilcoxon rank sum tests, with correction (Bonferroni) for multiple comparisons between genotypes. Correction was not applied for the parameters in call structure. This is because individual properties are assumed to be related to each other, which increases type 2 errors caused by overcorrection.

### Rotarod test
The tests were performed using the elevated revolving rod (Stoelting, Cat#. 57624). Mice were placed on the apparatus and habituated for few minutes. The rod accelerated at a constant rate (4 to 40 rpm in 300 s) and the time it took the animals to fall was recorded. Tests were performed three times and the average value was calculated. Statistical analysis was performed by Wilcoxon rank sum test.

### Y-maze test
The Y-maze tests were conducted according to the described procedure[130]. The tests were performed in a Y-maze with three arms of equal length at 120° angles to each other (Stoelting, Cat#. 60180). Mice were placed in the center of the maze and has free access to all three arms. If the animal chooses an arm different from the arm it arrived in, this choice is called an alteration. This is considered a correct response; conversely, returning to the previous arm is considered an error. The number of times and the order in which the animals entered the arms are recorded and used to calculate the alternation rate. The behavior of the mice was recorded for 8 min. Statistical analysis was performed by Wilcoxon rank sum test.

### Variant identification and annotations of Neanderthal and Denisovan
The sequencing data of three Neanderthal genomes were obtained from The Draft Neanderthal Genome Project (https://www.ebi.ac.uk/ena/browser/view/PRJEB2065). The sequencing data of Denisovan genome accompanied with nine modern human genomes were obtained from Denisovan Genome Project (http://cdna.eva.mpg.de/denisova/). The fastq files of each sample were aligned to the human genome (hg19) by Burrows-Wheeler Aligner (BWA, v0.7.17-r1188). The aligned SAM files were processed into BAM files by Picard (v2.18.7) and Genome Analysis Toolkit (GATK, v4.4.0.0). Variant calling for each sample was processed with Mutect2 of GATK4. The variants were annotated by using ANNOVAR (v2).

### Variants in NOVA1 loci
The variants in modern human populations were obtained from ExAC database (v0.3.1) (https://gnomad.broadinstitute.org/downloads), which contains 60,706 exomes mapped to hg19. Then, the variants in NOVA1 loci (chromosome 14: 26912296 ~ 27067239) of Neanderthal, Denisovan, and modern human populations were subsetted by using bcftools (v1.19). The frequency of minor alleles for each position in NOVA1 loci was calculated.

### Tajima's D statistical analysis
Corrected allelic frequencies of each SNP site from ExAC (v0.3.1) were extracted from the dbSNP database (version 2022-11-16). The sites with very low total allele count were filtered out (cutoff for total_count: 200). pi, theta and Tajima's D values were calculated for all genes examined[131]. The normalized Tajima's D value was calculated as the ratio of Tajima's D to its theoretical minimum value (D(min))[46].

### DH test
Allelic frequency for each mutation site was extracted from refsnp files (version b156) downloaded from NCBI (version of November 2022). In-house program was used to calculate the evolutionary statistic, Tajima's D is calculated by according[131], Fay and Wu's $H$-statistic is defined as $\theta_\pi - \theta_H$[132], we calculate normalized H (formula 11, 12), E-test (formula 13, 14), and DH-test (formula 15)[47]. Our in-house program is shared to public: https://github.com/cafeblue/popgen_dbsnp [133]

### Selection analysis on NOVA1 197 V
The ancestral recombination graph (ARG) analysis was performed using *ARGweaver-D*[48]. These ARGs explicitly describe gene trees and accompanying recombination events throughout the region. They were sampled from an approximate posterior distribution using Markov chain Monte Carlo methods to capture uncertainty in the ARG, given the sequence data and evolutionary parameters. The sampled ARGs included two Yoruba (HGDP00927, SS6004475), two Mbuti (SS6004471, HGDP0456), and two San (HGDP01029, SS6004473) individuals, all sequenced to high coverage, as well as the Altai Neanderthal and Denisovan sequences and a chimpanzee outgroup (panTro4). Because the NOVA1 197 V variant is nearly fixed in modern humans and likely predates the separation of major continental population groups, we do not anticipate significant additional insights from including more modern human samples, as most would coalesce well after the allele reached high frequency. Based on information from the ARG, the sampled ARGs were analyzed using *CLUES2*[49].

### Statistical information
Information of statistical methods and the number of biological replicates in the analysis are in the figure legends and methods section of each analysis as appropriate.

### Reporting summary
Further information on research design is available in the Nature Portfolio Reporting Summary linked to this article.

## Data availability
The data supporting the findings of this study are available from the corresponding authors upon request. Source data are provided with this paper. The sequencing data generated in this study have been deposited in the GEO database under the SuperSeries GSE253298 comprising Subseries GSE253296 and GSE253297 [https://www.ncbi.nlm.nih.gov/geo/query/acc.cgi?acc=GSE253298]. The vocalization data generated in this study are provided in the Supplementary Data file. The sequencing data for *Nova1* knockout mouse used in this study are available in the GEO database under accession code GSE69711. Source data are provided with this paper.

## Code availability
The code used in this study is available in the Zenodo database under accession code https://doi.org/10.5281/zenodo.14367749 [https://zenodo.org/records/14367749][133].

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

## Acknowledgements

The authors are deeply grateful to Dr. Yuhki Saito for providing guidance on the method and analysis of the transcriptome experiments. We wish to thank Dr. J Lomax Boyd for assistance in the design of the playback experiments. We are deeply grateful to the Rockefeller University Resource Centers: the CRISPR and genome editing center, the Transgenic and Reproductive Technology Center and the Genomics Resource Center. We wish to thank Dr. Molly Przeworski and Dr. David Reich for critical review and constructive comments, as well as members of the Darnell lab for discussions of the manuscript. Japan Society for the Promotion of Science postdoctoral fellowship for research abroad (J.S.P.S.) (YT). NIH Awards NINDS Outstanding Investigator Award R35NS097404 (R.B.D.). Keck Foundation Award and NIH Transformative Research Award R01DC018691 (E.D.J.). R.B.D. and E.D.J. are Howard

Hughes Medical Institute Investigators. This research was supported by US National Institutes of Health grant R35-GM127070 (to A.S.) and the Simons Center for Quantitative Biology at Cold Spring Harbor Laboratory. The content is solely the responsibility of the authors and does not necessarily represent the official views of the US National Institutes of Health.

## Author contributions

Conceptualization: R.B.D.; Methodology: Y.T.; Investigation: Y.T.; Statistics on population genetics: J.D.L., W.W., T.S.C.; Evolutionally analysis: J.X., N.K., L.C.M., A.S.; Visualization: Y.T.; Funding acquisition: Y.T., R.B.D., E.D.J.; Project administration: Y.T., R.B.D.; Supervision: K.I., R.B.D.; Writing—original draft: Y.T., R.B.D.;—review & editing: Y.T., K.I., C.D.M.V., J.D.L., W.W., E.D.J., A.S., R.B.D.

## Competing interests

The authors declare no competing interests.
