## [Transparent Peer Review file · Nature Communications]

A humanized NOVA1 splicing factor alters mouse vocal communications

Corresponding Author: Dr Yoko Tajima

Figures on pages 12, 49, 63, and 64 in this Peer Review File has been amended to remove third-party material where no permission to publish could be obtained.

Version 0:

Reviewer comments:

Reviewer #1

(Remarks to the Author)

This paper describes interesting effects on gene expression, splicing and vocalization in mice that carry an amino acid substitution in the splice factor Nova1. However, there are some issues with the work and the manuscript that need to be addressed before it can be published.

General, major comments/suggestions:

1: The review of the literature and previous work is incomplete and incorrect. Some major examples are:

a: The authors claim that no vocalization effects have been seen in mice that carry the two human-specific amino acid substitutions in the Foxp2 protein (page 2; line 10-17). This is incorrect.

Ref. 6 (Enard et al., Cell 2009) describes effects in pup isolation calls much like the authors do in the current paper.

The authors also claim that no effects have been seen in adult mice carrying these substitutions and cite ref 8

(Hammerschmidt et al., Genes Brain Behavior, 2015). It is correct that that paper do not find any effects. However, a later paper (van Meurens et al., Genes Brain Behavior, 2021) describes effects both in adult males and in adult females.

b: It is true that these two changes are seen also in Neanderthals and Denisovans as the authors say. However, this does not lead to the conclusion that the changes in FOXP2 do not underlie any aspects of human-specific language development as the authors claim.

Firstly, contrary to what the authors claim, vocalization have been described in mice humanized for Foxp2 (see above).

Second, the fact that the changes occur also in Neanderthals and Denisovans does not contradict that they have to do with human-specific language as these extinct forms of humans surely had language or vocal communication in some form. In fact, it is not known when fully modern human language evolved. Thirdly, the fact that no selective sweep can be seen in FOXP2 in present-day people (ref. 12) has no bearing on if the two amino acid substitutions were positively selected as they occurred so long ago (prior to the divergence to Neanderthals) that a sweep signal would not be detectable today.

c: The TKTL1 change (ref. 15) has nothing to do with chromosomal segregation (page 2, line 25).

d: The study where organoids where NOVA1 was humanized (ref. 30) has been criticized on technical grounds (Maricic et al., Science 2021) and effects like those described have not been seen in other organoids where this substitution has been introduced (Riesenberg et al., Nature Methods 2023). This should at least be mentioned.

e: "NOVA1 is neuron-specific RNA binding protein..." is the first sentence of the abstract and neuron-specificity of NOVA1 was mentioned also later in the paper. However, Villate et al. (PMID: 25249621, "Nova1 is a master regulator of alternative splicing (AS) in pancreatic beta cells") has shown that Nova1 has a major role in pancreatic beta cells of rats and humans. Additionally, human protein and mRNA atlases show that NOVA1 and NOVA1 are expressed in many tissues in addition to brain. This discrepancy could be addressed in the paper or the text modified.

2: A concern is that the vocalization effects may reflect general motoric effects of NOVA1^{hu/hu}. It would be good to test the mice in a battery of other standard tests of motor control and motor activity (e.g., rotarod, open field) to exclude that they have more generalized motoric issues. This is important especially as effects of NOVA1 mutations in humans have more generalized effects that are not restricted to speech and language.

3: They should compare effects they observe on USVs with the ones described for the mice carrying the two substitutions in *Foxp2*, both for the pups and the adults (for which effects have been described contrary to what the authors claim). It would be valuable to not restrict the analysis of the adult vocalization to males but also include females (van Meurens et al., *Genes Brain Behavior* 2021).

4: Are the 27 of 630 transcripts in the category "Behavior" more than expected by chance? This can be tested by resampling many sets of 630 transcripts to see how often 27 or more are associated with the category "Behavior".

5: The evolutionary analyses they present (pages 3; lines 12-42) are extremely similar to the ones presented in reference 12. To claim that a "strong evolutionary selective sweep" affected the NOVA 1 region, we would like to see further analyses. Negative Tajima's D and low allele frequencies of minor SNPs could be explained by purifying selection as well as selective sweeps. If a strong selective sweep affected the region, we would expect other measures such as extended haplotype lengths and reduced intra-population diversity relative to divergence to indicate this.

6: Differential expression analysis has been done in mouse embryo at day 18.5. CLIP and AS have been done on 3-week-old mice. Why use different mouse ages for different assays? This is unfortunate, as it is not possible to say if the genes that come up in CLIP and AS would come up as differentially expressed genes. If the authors would like to perform these analyses at the same time, it would be valuable (but not required in our opinion).

Minor comments

Line 316: "with two substitutions enriched in humans" should be "two substitutions specific for humans".

Line 376: a comma is not needed.

line 387: "which they term I200V accounting for an AS variant". Here, a transcript ID should be given if there is one.

Line 474: "8 animals harbored the designed allele (one missense substitution and two nonsense substitution)" is unclear. Did one allele have three substitutions? Probably not.

Reviewer #2

(Remarks to the Author)

This is an exciting manuscript describing the "humanization" of the NOVA1 gene in mice. NOVA1 is an RNA binding factor with a critical role during neurodevelopment. Interestingly, NOVA1 is one of the genes that contain genetic variants unique to modern humans (not even present in Neanderthals, our closest relatives). The group used genome editing technologies to change a single amino acid (I197V) in mice and observed changes in gene expression and alternative splicing. Remarkably, the humanized animal showed significant alterations in vocalization, both in pups and adults. The work supports the idea that this genetic variant was positively selected in *Homo sapiens*, likely due to language development. The observation that a humanized splice factor can affect vocalization in mice is quite an amazing finding. Overall, I find the work quite impressive, both in terms of its technical aspects and its message and contributions to human evolution. While I am enthusiastic and supportive of the publication of these findings, several aspects need clarification. I have listed below a series of suggestions to improve the manuscript, from the choice of NOVA1 as a candidate gene for these experiments to some more technical questions.

-The introduction has a detailed summary of human-specific genes, which I appreciated. However, the reason why they chose NOVA1 is not immediately clear; this information will emerge later. Below, I have some suggestions on how this could appear early in the manuscript.

-They use reference #2 to illustrate how humans are the only primates capable of vocal learning. However, this reference seems related to mouse vocalization and does not support that idea.

-When discussing FOXP2, previously called a "human language gene", the authors should point out that reference #6 has actually found alterations in mouse vocalization. Later, reference #8 goes against it. The difference between these two works should be highlighted as this work also uses mice vocalization as a readout, and readers need to understand why the previous work has observed a significant difference that gives rise to the misconception that FOXP2 was uniquely human.

-Similarly, reference #2 has found evidence for positive selection of FOXP2, but that was later debunked by reference #12. The reasons these two groups have divergent conclusions about FOXP2 should also be stated in the introduction.

-More recently, another gene (TKTL1) was shown to have human-specific consequences, but again, that was also debunked, as the variant is also present in modern human populations. The sentence "this substitution is present in a small percentage (0.03-0.2%) of modern humans" refers to Pinson et al (16), but the correct reference is from Herai et al., 2023 ("Comment on "Human TKTL1 implies greater neurogenesis in frontal neocortex of modern humans than Neanderthals". *Science*. 2023").

-Both FOXP2 and TKTL1 have failed to be validated as modern human-specific genes, which seems to be because a uniquely European-biased dataset was used as representative of all modern humans (on references #12 and #30). This also needs to be stated in the introduction to avoid further misinterpretations.

-Thus, the paragraph: "Comparison of Neanderthal and Denisovan genomes to modern humans identified 260 human-

specific single nucleotide changes (SNCs) that result in amino acid substitutions¹⁰. Of the 23 changes that were most conserved, eight were involved in brain function and neural development. One of the eight was human-specific isoleucine to valine substitution at position 197 (I197V) in the neuron-specific RNA binding protein neuro-oncological ventral antigen1 (NOVA1)", needs to be updated. The more recent work on reference #30 actually uses several modern human datasets to narrow the number of human-specific genes to 61.

-The NOVA1 variant is, indeed, one of the 61 genes uniquely present in modern humans. Moreover, reference #30 showed that the archaic variant alters neurodevelopment in humans. Thus, this information should appear up front as a rationale for why the authors focused on this gene.

-The sentence "An expanded genomic analysis of the dbSNP database revealed that this substitution was present in all but six of 650,058 human sequences, five of which were from individuals of Asian descent" is related to homozygote or heterozygote individuals? What are these frequencies?

-The group has generated NOVA1wt/wt and NOVA1hu/hu. NOVA1ko/ko is lethal, but what is the behavior of NOVA1wt/hu or NOVA1ko/hu? Is a single human allele enough to drive the phenotypes described here?

-The vocalization phenotypes are pretty amazing. When discussing the data, the authors should revisit the differences observed in references #6 vs. #8 and situate their own methods. This seems quite important to avoid another scientific fiasco, as others might want to replicate the author's mice observations.

-Some information is missing from the methods, including:

*detailed information on the bioinformatics analysis, including software parameters for all multi-omics analysis;

*accession code for all sequenced multi-omics datasets;

* version of ExAC database and from all others used in the manuscript.

-Figure 2B shows related gene expressions indicating they are similar. However, what is the statistical significance variation, and is there a detailed calculation on which reference gene was used to perform the related expression (this information is unclear)? Moreover, according to the figure, NeuroD1, an important neuronal marker, looks to present a significant alteration between Nova1 (hu/hu) and Nova1 (wt/wt), which is contrary to what is claimed by the manuscript.

-according to the authors, Figure 2C indicates that Nova1 is most highly expressed in the mouse midbrain. However, additional protein quantification is required to confirm this observation.

-Figure 2d shows the expression variation between Nova1 (ko/ko) and Nova1 (wt/wt). A table listing all genes, including expression, fold change, and statistical significance would be desirable. Moreover, please include a clear description of the x/y axis in the figure's legend.

-non-significant (p -value <0.05) gene expression alterations could be removed from Figure 2, and it should not be mentioned as altered in the manuscript since there's no statistical support.

-Figure 3C is explained as "Among the transcripts with significant AS changes, 47 of them had significantly different NOVA1 binding peaks on their transcripts between NOVA(hu/hu) and Nova1(wt/wt) (Figure 3C)." However, RNA-seq and CLIP data show similar and comparable read coverage and peaks, respectively. The only observed variation is between the number of reads, but the values do not look to be normalized for a correct comparison between them. I suggest to clarify this description.

-The gene ontology analysis did not correctly describe the origin of the genes considered in each pathway (Figure 3D). Did the authors use a background dataset from a normal brain, or was the analysis based on the background provided by the software?

-How did the authors confirm that the CRISPR experiment did not generate any relevant off-target effects? How many founders were used in this work?

Reviewer #3

(Remarks to the Author)

Reviewer #4

(Remarks to the Author)

In this manuscript, Tajima and co-workers look to address a fascinating question about the molecular and genetic underpinnings that accompanied human evolution, in particular surrounding vocal learning. The work described in this manuscript builds on the observation of a highly conserved, 'human-specific' mutation within the splicing factor NOVA1. To better understand the role of this mutation, authors generate a mouse containing a 'humanized' version of NOVA1 and

examined the impact of this mutation on several different molecular and phenotypic readouts. The bulk of the experimental data included in the manuscript are derived from RNAseq analysis, CHIPseq analysis, and analysis of vocal patterning. While I find the problem and the approach quite interesting and compelling, I am not convinced that the data presented support a strong conclusion about any role for this particular mutation in the pathway of interest. Certainly there are hints about the potential for a role, but for reasons expanded upon below it doesn't strike me that there is strong evidence in support and so I don't see this as a good candidate for publication in a journal with broad readership like Nature Communications.

Regarding the RNAseq dataset, I have several questions/concerns about methodology surrounding data analysis, and more broadly about the conclusions drawn from this. From a methodological perspective, it seems that data shown in figure 2 panels B, D, E, F, and G are all derived from analysis of RNAseq done on mice with three different genotypes (wt, ko, and 'humanized' NOVA1). Although the manuscript notes that the data from these experiments are available via GEO, no accession number is provided and no such dataset appears available that is linked to the two corresponding authors (specific to this study). Although it isn't entirely clear from the text, it appears that figure 2B is derived from one RNAseq study (P21 cerebral cortex), while figures 2D, E, F, and G are derived from a separate RNAseq study (e18.5 midbrain). For the former, authors have selected (how?) a handful of cell-type specific markers and have represented the data from the sequencing data using box plots and (apparently?) statistical analysis using a Wilcoxin test to compare their differences (based upon the figure legend). From a statistical perspective, I would argue that it is inappropriate to extract RNAseq data in this way and use individual tests on the underlying counts. Authors describe using edgeR for analysis of differential expression, but it isn't clear how or why they changed the approach in this figure. Certainly there is wide agreement in the field that the statistical approaches used by a program like edgeR are superior (in spite of their limitations) to the approach that authors appear to have taken for these extracted genes. This technical point aside, it is surprising/disappointing that authors have provided no further insights into any of the other information derived from this initial RNAseq experiment. Are there other changes observed? What do those changes suggest? How do they compare with the other RNAseq experiments described in the figure (from a different developmental time point)? Currently, it reads as if authors are only presenting the data that support a single conclusion and but they don't provide access for the readers to contemplate alternative explanations/interpretations.

Regarding panels D, E, F and G of figure 2, the approach to analysis here seems flawed intellectually rather than statistically. If I understand authors' approach, they have used edgeR to identify a subset of genes with differential expression in the ko strain (vs. wt), and then asked about the behavior of that subset of transcripts in the background of the humanized strain. But it seems clear from the lower panel of 2D that many of these transcripts are not affected by the V197I variant and so their inclusion in this analysis seems irrelevant (certainly one could imagine other alleles of NOVA which altered its activity in ways that would be totally unrelated to the V197I variant, why should the ko be considered differently?). Much more important would be to see a robust statistical analysis of all of the transcripts whose expression is differentially impacted in a comparison of the V197I variant to the wt strain. Perhaps a reciprocal analysis could be interesting (that is, highlight in 2D the edgeR identified transcripts from the hu vs. wt experiment and then show those in the context of the ko), but the current approach does not make sense for trying to understand the role of the V197I variant.

Regarding the CLIPseq, I would again note that it was difficult to easily evaluate the raw data from authors' experiments for a number of reasons including trivial ones (the supplemental tables are mis-numbered between the text and the supplemental tables) to meaningful ones (only author-selected data are available). Nevertheless, my main concern with these data are that the apparent changes in CLIP between wt and V197I are modest in impact, modest in statistical significance, and of unclear relationship to any of the changes previously described from the RNAseq experiment. In the text, authors describe 215 differential peaks between wt and hu, but then say that 78 were higher in human and 67 were higher in V197I ... what does that mean about the other 70 peaks? In looking at Supplemental Table 2 (not 3 as described in the text), it appears that authors have 'trimmed' the list of peaks included on the "ST2.1_diff.peaks" tab to include only those with a log2FC value above a certain threshold (apparently 1.47?) on the subsequent tabs, presumably resulting in the 78 and 67 peaks noted above. But it is unclear how this threshold was selected. Moreover, if authors don't have confidence in those with lower FC then why were they called significant in the first place? Likewise, if there is a threshold for FC in considering significance, shouldn't there also be for count depth? I note that many of the called peaks are covered by an exceedingly small number of reads (again, it is difficult to compare these to the remaining ~29,000 peaks because authors have only provided a small amount of the underlying data.).

Importantly, there appear to be several inconsistencies between the data shown in figure 3 and the data included in the supplemental tables. Figure 3C includes three transcripts which authors describe as having statistically significant changes in AS and having differential CLIP peaks. But two of these transcripts, *Gria2* and *Ube2q2*, don't appear in the supplemental data files that authors include. What are the data that are included for this figure representing? For me, this is the weakest portion of the paper. To really make the argument that the V197I variant is important there needs to be some molecular defect that can be plausibly traced to the mutation. Figure 3B is quite uninteresting to me: it shows many apparent AS events which seem to be unrelated to any detectable change in NOVA binding. Are these simply noise in the AS calling? If not, how to understand them if NOVA binding (CLIP) isn't impacted? If the V197I is altering something other than binding (CLIP), then are the CLIP differences all noise? Critically, I am again worried about cherry-picking of data given the apparent inconsistency between what is shown in the figure and what is presented in the supplemental table.

Regarding the AS analysis, I again note the challenge of critically analyzing authors' underlying data. Authors note that ~40% of the detected AS events also had NOVA peaks on them "implicating direct effects of changes in NOVA1 amino acid substitution" but it is entirely unclear how the former connects to the later. As noted above, there are many apparent AS events for which there is no apparent change in NOVA binding. And the counter to authors' note is that ~60% of the detected AS events contain no NOVA binding. Together, it is entirely unclear if/how the V197I variant can be mechanistically linked to a critical role in the observed changes in AS (or gene expression, or CLIP signal).

Regarding the vocalization data included in the manuscript, I would note that I am not expert in this field and so cannot fully weigh in on these data. Nevertheless, from a statistical perspective, it strikes me that these are modest changes at best (in some cases relying on differences not in means but in variance between samples). Perhaps this is considered acceptable

within this field, but given my concerns with authors' treatment of statistical data from the molecular aspects of this work described above, I am concerned about the significance of these results as well.

Reviewer #5

(Remarks to the Author)

The exploration of the genetic basis for human language and elaborate vocal communication is both fascinating and critical. This topic has been previously addressed but often without providing satisfactory answers. The study in question investigates the role of the NOVA1 gene, particularly a human-specific mutation, I197V, and its implications for vocal behavior. The findings present a compelling narrative, though some aspects warrant further scrutiny.

As a non-expert in genetics, my ability to evaluate the detailed genetic analysis is limited. However, the study's approach appears thorough and convincingly novel, with deep evidence supporting its conclusions. The introduction of the human-specific I197V variant in NOVA1 into mice (Nova1^{hu/hu}) and the subsequent analysis of molecular and behavioral changes provide a significant contribution to our understanding of human evolution.

The relationship between the described NOVA1 mutation and vocalization is particularly intriguing. The study suggests that the I197V variant may have played a role in the evolution of human spoken language. The analysis indicates that this variant affects RNA regulation and alternative splicing in the brain, which in turn influences vocal behavior. However, while the evidence for changes in brain gene expression are compelling, the link to vocalization behavior appears somewhat limited and inconclusive. While it provides evidence suggesting a connection, it does not definitively prove it. The evidence mainly supports the hypothesis but lacks conclusive power.

In order to strengthen this link the authors generate a mouse mutant where the humanized mutation is introduced and compared well characterized ultrasonic vocalizations (USVs) behavior in mice, both in pups and adults.

- a. It is unfortunate that raw vocalization data, particularly for adult courtship, is not presented in the main text, limiting the ability to fully assess the findings.
- b. The performed analysis is reasonable but not highly sophisticated, potentially impacting the robustness of the conclusions.
- c. The statistical results are weak, with all the significant tests showing significance levels around 0.05 except for Fig 4E and 4G. There is no mention of corrections for multiple comparisons (at least I couldn't find it), raising concerns about the reliability of the significant findings. If these tests were not corrected it is highly likely that they would not pass a significance test and will make the conclusions obsolete. It is also suspected that the high levels of significance in Fig 4E,G are due to the huge number of samples and may not represent a real change. For example the huge green bars on the histograms on fig4g at about 120KHz are clearly an artifact and also are likely beyond the hearing range of the mice. It is surprising that a research team with a strong genetic analysis would apply less stringent criteria to non-genetic data.

In conclusion, the topic of genetic mutations contributing to human language is of extreme interest and potential. The study makes a significant effort to link the NOVA1 mutation to changes in vocalization behavior, but the connection remains tentative. The evidence for a real change in vocal behavior due to the genetic mutation needs to be stronger to make definitive conclusions. I would expect more rigorous analysis and if required additional data to establish the link. The authors have laid important groundwork, but further efforts are necessary to solidify the relationship between the genetic findings and the phenotypic expressions in vocalization.

Version 1:

Reviewer comments:

Reviewer #1

(Remarks to the Author)

The authors have done an impressive amount of work to address the concerns of the reviewers. All our concerns are adequately addressed.

Reviewer #2

(Remarks to the Author)

In this revised version, the authors comprehensively answered many of the reviewers' questions, including demonstrating the specificity of NOVA1 to the CNS, excluding motor alterations in their mouse model, improving their vocalization data and statistics with comparative analyses with other publications, confirming the strong selection pressure at the NOVA1 197V, include more details about their bioinformatic pipelines, performed a GO analysis and found genes related with speech in certain brain regions, included off-target analyses, showed that the 197V substitution is not causing loss of function, among several other clarifications.

The manuscript is a much-improved version, and several novel data back up their findings. The observation that the archaic NOVA1 variant is linked to language is fascinating and corroborates previous observations made in human cells. Nonetheless, certain aspects of the introduction need to be scientifically accurate.

1) Maricic et al.'s technical criticism has been previously responded to by Herai et al. Science 2021. It nicely demonstrated that even though one alteration occurred in one clone of one of the cell lines, this alteration changed the phenotype from Arc/Arc to Arc/KO. Thus, the specific clone in question has only the Arc version of NOVA1, confirming the original results from Trujillo et al. 2021. Therefore, the authors should not use Maricic et al. 2021 to highlight a technical inconsistency in Trujillo's original publication.

2) I was surprised to learn about the work of Riesenbergs et al., 2023 and decided to check it out more carefully. In fact, the authors claimed somewhere hidden in the manuscript that they could not reproduce the organoid morphological changes upon replacing the NOVA1 version with the archaic variant in a single cell line. However, upon close inspection, the work does not challenge the original Trujillo et al. 2021 publication. Technically, the experiment was done with a single allele being substituted, using a single cell line and a completely different brain organoid protocol to analyze organoid morphology. Moreover, the authors only analyze organoid morphology as a phenotype. Organoid morphology is variable and dependent on the genetic background of the cell line (Trujillo et al. used different iPSC lines, not just one). Based on this, one cannot conclude that the experiments done by Riesenbergs fall short of trying to reproduce the effects of NOVA1 archaic version in the human model. Thus, as of now, no publication experimentally challenged the data of Trujillo et al. 2021 in a rigorous fashion. This sentence in the introduction needs to be altered for scientific accuracy and to avoid propagating wrong information:

"Studies have explored the NOVA1 I197V variant by reverting the ancestral isoleucine 197 variant 93 back into human iPSC-derived organoids, revealing morphological and electrophysiological 94 changes in vitro¹³. However, these effects were not observed in another study that reintroduced the 95 same substitution in different iPSCs³⁹. This discrepancy underscores challenges of obtaining 96 consistent results with varying experimental methods and materials in vitro^{40,41}."

Other than these adjustments, the manuscript compiles a series of experimental evidence that links, for the first time, a modern-human-specific point mutation to vocalization. This observation opens novel perspectives to study this alteration from evolutionary, linguistic, and neuroscience perspectives.

Reviewer #4

(Remarks to the Author)

In this revised manuscript, Tajima and colleagues have made extensive changes in response to feedback from original reviewers. I commend authors on their approach to addressing the concerns raised by this (and other) reviewers, it is a remarkable amount of work that has been clearly dedicated to this revised manuscript. Nevertheless, and in spite of my continued general interest in the work presented here, for reasons elaborated upon below I admit to being underwhelmed by the experimental support that is presented for the claimed conclusions, and so can't provide a strong argument in favor of its publication.

One important change in the revised manuscript is a change in the conclusions about the RNAseq and CLIPseq data generated by authors. Whereas the original manuscripts had sections with the following titles: "Humanized NOVA1 mice show normal development, but have specific gene expression changes in brain" and "Human and ancestral NOVA1 have the same RNA-binding activity but distinct RNA targets", the revised approach to data analysis in the current manuscript led authors to remove any such claims, so that the equivalent sections are now titled: "Humanized NOVA1 mice are comparable to wild-type mice in development and gene expression in the brain" and "Modern human specific amino acid substitution does not affect sequence specific RNA-binding capacity of NOVA1". In my view, this is certainly a positive improvement in the quality of the science, and it relieves some of my prior criticism regarding the disconnect between binding targets, AS targets, and overall changes in gene expression. On the other hand, it does reduce the experimental support for a biologically meaningful impact of the I197V variant. Indeed, at a molecular level authors have now reduced the molecular argument to the idea that there are changes in AS in the humanized variant. Certainly this is a plausible outcome of an allele of a protein like NOVA1, but the results shown here are again modest in impact, and modest in statistical significance.

When considering the changes in AS associated with the I197V variant as described in the current manuscript, I would argue that by themselves they aren't sufficient to make a strong claim. Certainly if this were a manuscript exclusively focusing on the molecular changes associated with this variant, one would expect considerably more work in support of these claims. Authors correctly note that interactions of RBPs with spliced RNAs can be complex, and indeed complicated in how they impact AS, but alleles with single amino acid substitutions are likely to be much less pleiotropic in their impact – here we are left with no hints about how this variant is driving the changes in AS. Again, AS events are about equally likely to have a NOVA1 binding signature as not. Based on homology, authors (plausibly) suggest that the I197V allele could disrupt the protein-RNA interface, but how would this work for those events without CLIP signals? Again, authors argue that the absence of a peak may reflect a technical 'artifact' (that detection requires the interaction to satisfy the chemical requirements for UV-induced crosslinking), but at ~60% of the detected AS events lacking a CLIP signal, this is a tough argument to swallow. Alternatively, authors suggest that those AS events without a CLIP signal might reflect 'secondary' effects of changes in other RBP whose splicing was directly impacted by I197V – again, if this manuscript were strictly focused on the molecular defects associated with I197V then one would expect some additional analysis of the 'true' CLIP-positive transcripts to determine the cis-features that enabled their differential regulation in the I197V relative to wildtype. Perhaps the differential AS events described here by authors are indeed real, but I'm not sufficiently convinced that they are anything more than noise that might be seen in a complex experiment such as this.

Regarding the vocalization studies that authors include, I would once again note that I am not expert in this field and so

cannot strongly weigh in on the conclusions drawn. I would, however, reiterate my hesitation given the statistical significance associated with many of these studies – I am concerned that these merely reflect noise. Perhaps equally importantly, I don't consider these studies to be a substitute for a more considered analysis of the molecular defects of the I197V variant.

Reviewer #5

(Remarks to the Author)

I appreciate the efforts made by the authors to address my concerns, particularly with regard to the reanalysis of the vocalization data. The new analyses demonstrate a more careful approach in addressing sample size issues and corrections for multiple comparisons, which I had previously flagged as a potential concern. Indeed, some of the previously reported findings have failed to pass these criteria, but some findings were also found to be robust with stricter rules. The mixed findings presented in the revised manuscript provide a more nuanced view of the NOVA1 mutation's impact on vocalization.

The updated data of ultrasonic vocalizations (USVs) show important differences across developmental stages. The observation that the humanized genotype alters some features of pup USVs but that those changes do not persist into adulthood, while other differences emerge in adult USVs, adds depth to the analysis and conclusions.

At this stage, I think that it is mainly a judgment call. I don't have strong suggestions for further analysis and I think that overall the findings are of high interest and importance. Nevertheless, I still find that the link between these vocalization changes and the NOVA1 mutation remains suggestive rather than conclusive. On the other hand, the yardstick here may be previous studies on humanized mutation in the FOXP2 genes that have seen similar effects on vocalizations (even similar in nature as the authors now highlight in the text of the new revision). These FOXP2 findings have also been published in high-profile journals, and thus, I personally feel that this manuscript has passed the threshold.

I recommend that it be published in Nature Communications.

Response to Reviewers

We sincerely appreciate the reviewers' insightful and constructive comments on our manuscript 'A humanized NOVA1 splicing factor alters mouse vocal communications'. In response to each of the reviewers' suggestions, we have made significant improvements to the manuscript.

Specifically, we have undertaken extensive revisions and additional analyses to address the concerns raised. We have updated the manuscript, incorporating the new data and findings highlighted in blue font, alongside the additions and revisions described below. We are grateful for the reviewers' feedback, which has significantly enhanced the quality and rigor of our study.

The following is a summary of the data that has been updated or incorporated into the revised manuscript:

- **Fig. 2c**, Gene expression correlations between *Nova1^{wt/wt}* and *Nova1^{hu/hu}* in midbrain at e18.5 and P21.
- **Fig. 2e**, Distribution of NOVA1 CLIP peaks on the genome.
- **Fig. 2g**, Scatterplot of CLIP tag number per peak between *Nova1^{wt/wt}* and *Nova1^{hu/hu}*.
- **Fig. 2h**, Gene annotation analysis of NOVA1 bound transcripts.
- **Fig. 3a**, NOVA1 immunostaining in P21 mouse brain.
- **Fig. 3b**, Western blotting for NOVA1 in dissected mouse brain.
- **Fig. 3e**, Gene annotation analysis for the transcripts with differential AS events.
- **Fig. 4c**, $F_{q_{max}}$ distribution and two Gaussian fit in pup USVs.
- **Fig. 4d**, Ratio of high or low $F_{q_{max}}$ in syllables "d" and "m".
- **Fig. 4f**, Duration distribution and two Gaussian fit for syllable "s" in adult USVs.
- **Fig. 4g**, Peak frequency parameters in long duration "s".
- **Fig. 4h**, $F_{q_{max}}$ distribution and two Gaussian fit in adult USVs.
- **Fig. 4i**, Frequency variance in high $F_{q_{max}}$ in adult USVs.
- **Supplementary Fig. 1a**, The SNP report from dbSNP database.
- **Supplementary Fig. 3**, Recent burst of coalescence of a sweep for NOVA1 197V.
- **Supplementary Fig. 4d**, The gRNA sequence for the generation of *Nova1^{hu/hu}* mice and predicted off-target site information.
- **Supplementary Fig. 4e**, The genomic sequencing of the potential off target loci.
- **Supplementary Fig. 5a**, Global correlation matrix of gene expression levels between brain samples.
- **Supplementary Fig. 5b**, Principal component analysis of gene expression levels between samples.
- **Supplementary Fig. 5c**, Principal component analysis of gene expression levels in each corresponding sample.
- **Supplementary Fig. 5d**, Gene expression correlations between *Nova1^{wt/wt}* and *Nova1^{hu/hu}* in corresponding brain regions and age.
- **Supplementary Fig. 5e**, Principal component analysis of gene expression levels between *Nova1^{ko/ko}* and *Nova1^{wt/wt}* midbrain at e18.5.
- **Supplementary Fig. 7**, KH domain sequence alignment.

- **Supplementary Fig. 8**, NOVA1-CLIP analysis in 3-week-old mouse cortex and cerebellum.
- **Supplementary Fig. 10**, NOVA1 immunostaining in the mouse brain.
- **Supplementary Fig. 11**, NOVA1 protein expression in dissected mouse brain.
- **Supplementary Fig. 13**, The resampling analysis for differential AS events in *Nova1^{hu/hu}* mice.
- **Supplementary Fig. 15a**, Syllable composition and amplitude for pup USVs.
- **Supplementary Fig. 15d**, Density plots of Fq_{\min} and Fq_{\max} in adult USVs.
- **Supplementary Fig. 17a**, Rotarod test.
- **Supplementary Fig. 17b**, Y-maze test.
- **Supplementary Table 2**, Selection analysis for human-specific SNPs using CLUES2.
- **Supplementary Table 3**, Differential CLIP peaks between *Nova1^{wt/wt}* and *Nova1^{hu/hu}*.
- **Supplementary Table 4**, Differential alternative splicing events in *Nova1^{hu/hu}* mice.
- **Supplementary Table 6**, Syllables detected in isolation induced USV test in pups.
- **Supplementary Table 7**, USV features in each pup.
- **Supplementary Table 8**, USV features in each genotype of pup.
- **Supplementary Table 9**, Bimodal distribution parameters at Fq_{\max} (maximum frequency) in pup USVs
- **Supplementary Table 10**, Proportion of high/low Fq_{\max} in pup-USVs in each genotype.
- **Supplementary Table 11**, Syllables detected in courtship induced USV test in adults.
- **Supplementary Table 12**, USV features in each adult mouse.
- **Supplementary Table 13**, USV features in each genotype of adult mouse.
- **Supplementary Table 14**, USV characteristics in long/ short duration “s” in adult-USVs in each genotype.
- **Supplementary Table 15**, Proportion of high/low Fq_{\max} in adult-USVs in each genotype.
- **Supplementary Table 16**, USV characteristics in low/ high Fq_{\max} in adult-USVs in each genotype.
- **Supplementary Table 17**, Comparison of vocalization tests between humanized mouse models.
- **Supplementary Table 18**, Expression changes in *Nova1^{ko/ko}* midbrain at E18.5.

Reviewer comments and Response (Response are in blue)

Reviewer #1 (Remarks to the Author)

This paper describes interesting effects on gene expression, splicing and vocalization in mice that carry an amino acid substitution in the splice factor Nova1. However, there are some issues with the work and the manuscript that need to be addressed before it can be published.

We sincerely thank the reviewer for their thorough evaluation of our manuscript and are pleased that they found our observations interesting. We also appreciate the constructive comments provided. We hope that our revised manuscript fully addresses the concerns raised by the reviewer.

General, major comments/suggestions:

1: The review of the literature and previous work is incomplete and incorrect. Some major examples are:

a: The authors claim that no vocalization effects have been seen in mice that carry the two human-specific amino acid substitutions in the Foxp2 protein (page2; line 10-17). This is incorrect. Ref. 6 (Enard et al., Cell 2009) describes effects in pup isolations calls much like the authors do in the current paper. The authors also claim that no effects have been seen in adult mice carrying these substitutions and cite ref 8 (Hammerschmidt et al., Genes Brain Behavior, 2015). It is correct that that paper do not find any effects. However, a later paper (van Meurens et al., Genes Brain Behavior, 2021) describes effects both in adult males and in adult females.

We thank the reviewer for pointing this out. We now have correctly cited these three studies.

Introduction:

“Studies on mice with the two amino acids substituted to the human version have reported vocal changes both in the neonatal²⁰(Enard et al., 2009) and adult stages^{21,22}(Hammerschmidt et al., 2015, von Merten et al., 2021). While Hammerschmidt et al. observed minimal vocal changes, von Merten et al. reported qualitative changes under a more natural vocalization paradigm^{21,22}, suggesting the involvement of these two amino acids in vocalization.”

We have also responded to comment #3 from this reviewer and made a table comparing these studies to our own (Page 11-13 of this response letter).

b: It is true that these two changes are seen also in Neanderthals and Denisovans as the authors say. However, this does not lead to the conclusion that the changes in FOXP2 do not underlie any aspects of human-specific language development as the authors claim.

Firstly, contrary to what the authors claim, vocalization have been described in mice humanized for Foxp2 (see above). Second, the fact that the changes occur also in Neanderthals and Denisovans does not contradict that they have to do with human-specific language as these extinct forms of humans surely had language or vocal communication in some form. In fact, it is not known when fully modern human language evolved. Thirdly, the fact that no selective sweep can be seen in FOXP2 in present-day people (ref. 12) has no bearing on if the two amino acid substitutions were positively selected as they occurred so long ago (prior to the divergence to Neanderthals) that a sweep signal would not be detectable today.

We agree with the reviewer on all three points. We have re-cited previous works on humanized Foxp2 mice as noted above. We have also documented the relationship between FOXP2 and vocal communication, and added a revised interpretation of the changes during evolution.

Introduction:

“The transcription factor forkhead box P2 (FOXP2) is of particular interest as a potential driver of human language function, as it harbors two amino acid substitutions present in human but not in chimpanzee and many other mammal genomes. Families with FOXP2 mutations exhibit severe speech defects^{15,16} while FOXP2 disruption in mice leads to vocalization abnormalities^{17,18} suggesting a role in spoken language function.

... these substitutions are also present in archaic humans, and comprehensive analyses using diverse human genome datasets have found no evidence of recent selection. This suggests that the FOXP2 substitutions occurred earlier than initially thought^{12,22}.”

c: The TKTL1 change (ref. 15) has nothing to do with chromosomal segregation (page 2, line 25).

We appreciate the reviewer for pointing this out. The main text has been corrected.

d: The study where organoids where NOVA1 was humanized (ref. 30) has been criticized on technical grounds (Maricic et al., Science 2021) and effects like those described have not been seen in other organoids where this substitution has been introduced (Riesenberg et al., Nature Methods 2023). This should at least be mentioned.

We appreciate the reviewer pointing this out. We have cited these references and included a discussion of the NOVA1 organoid study as follows.

Introduction:

“Studies have explored the NOVA1 I197V variant by reverting the ancestral isoleucine 197 variant back into human iPSC-derived organoids, revealing morphological and electrophysiological changes *in vitro*¹⁴(Trujillo et al., 2021). However, these effects were not observed in another study that reintroduced the same substitution in different iPSCs⁴⁰(Reisenberg et al., 2023). This discrepancy underscores challenges of obtaining consistent results with varying experimental methods and materials *in vitro*^{41,42}(Maricic et al., 2021, Herai et al., 2021).”

e: “NOVA1 is neuron-specific RNA binding protein...” is the first sentence of the abstract and neuron-specificity of NOVA1 was mentioned also later in the paper. However, Villate et al. (PMID: 25249621 , “Nova1 is a master regulator of alternative splicing (AS) in pancreatic beta cells”) has shown that Nova1 has a major role in pancreatic beta cells of rats and humans. Additionally, human protein and mRNA atlases show that NOVA1 and NOVA1 are expressed in many tissues in addition to brain. This discrepancy could be addressed in the paper or the text modified.

We thank the reviewer for the comment regarding NOVA1 expression.

Although we did not detect NOVA1 protein expression in mouse pancreatic islets (see below), we recognize that these results cannot entirely exclude the possibility of NOVA1 expression in some rare cell types outside of the CNS. In light of the reviewer’s comment and the consideration, the following modification was made to the manuscript:

Abstract:

“NOVA1, a neuronal RNA-binding protein expressed in the central nervous system, is essential for survival in mice and normal development in humans.”

Introduction:

“NOVA1 is highly expressed in neurons of the central nervous system (CNS) in both mice and humans²⁵ (Buckanovich et al., 1993), and its expression has also been observed in cultured human and rat cells²⁶⁻²⁸ (Eizirik et al., 2012, Villate et al 2014, Yang et al., 2023).”

We are aware of reports on the expression and function of NOVA1 in peripheral tissues. We have also noted that in POMA patients with large amounts of autoimmune antibodies against NOVA protein in the serum, symptoms are restricted to the nervous system (e.g. oculomotor disorder, ataxia), and no abnormalities of the pancreatic, gastrointestinal tract, or reproductive system has been reported (Luque et al., 1991, Simard et al., 2020, Velazquez et al., 2014, Musunuru et al., 2008, Fadare et al., 2004). Moreover, previous NOVA1 expression analysis in human and mouse tissues (include pancreas) showed that there is no detectable NOVA expression other than central nervous system (Buckanovich et al., 1993) (figures below).

REDACTED

Figure from Buckanovich et al., 1993. (left) Human Northern Blot analysis of NOVA expression. (right) Mouse IHC of NOVA in the e18.5 mouse. Fig 6A from Buckanovich et al., 1993. Reactivity is restricted to the ventral nervous system. Vmb, ventral midbrain; dmb, dorsal midbrain; vsc, ventral spinal cord; li, liver; si, small intestine.

While these observations do not exclude the possibility that NOVA protein is expressed outside of the CNS, particularly in a rare population, it is strongly suggested that it is predominantly the nervous system where NOVA1 is expressed and functions in human.

The study of NOVA1 in pancreatic islets was conducted in rat tissues (Villate et al., 2014), which was a follow up of their previous study of human pancreatic islets (Eizirik et al., 2012). In their report, NOVA1 expression as well as its function for splicing was analyzed either in primary culture or in a cell line (INS-1E), but not intact tissues. Increased expression of the NOVA protein has been reported in some cancer cells or related cell lines (Luque et al., 1991, Ludlow et al., 2018, Yu et al., 2018, Saito et al., 2024), and we are not fully convinced that NOVA1 expression in pancreas is conclusive as cells cultured *in vitro* may not reflect the expression *in vivo*. They

showed one immunostaining image of NOVA1 in human pancreatic islet in tissue, but the specificity of the antibody does not appear to have been validated.

Regardless of these concerns about their *in vivo* relevance, we have carefully considered the possibility and performed NOVA1 immunostaining in mouse pancreas cryosections.

NOVA1 immunostaining in mouse tissues (pancreas and brain). Twelve-week-old male mice were used for the analysis, corresponding to the age of adult rat cells studied by Villate et al. To identify the pancreatic islets, co-staining was performed with antibodies against Glucagon and Insulin. Brain sections stained in a similar manner (sample preparation, staining protocol, NOVA1 antibody) were also presented as a reference for staining methods, although expression levels cannot be compared.

*Anti-NOVA1 antibody [EPR13847] (abcam). The reactivity and specificity were validated previously (Saito et al., 2016, Tajima et al., 2023).

In these experiments in our hands, NOVA1 signals in the adult mouse pancreatic islets (Glucagon⁺ or Insulin⁺ cells) were undetectable, whereas a strong NOVA1 signals were detected in cryosection of adult mouse brain tissue by the same immunostaining method. Based on these results, we believe that NOVA1 is not expressed at significant levels in the mouse pancreas. Possible reasons for the discrepancy between our NOVA1 expression results and the Villate et al. report include (1) the use of different antibody (we have validated the strength and the specificity of the antibody using KO tissues for various applications such as WB and IF/IHCs), or (2) species differences (mouse vs. human vs. rat). Indeed, based on the analysis of several RNA sequencing datasets, we have found the possibility of changes in expression pattern of NOVA1 between humans and mice. Interestingly, Yang et al. (2023) reported differences in NOVA1 expression during adipogenesis, noting that NOVA1 is upregulated during adipocyte differentiation in humans, leading to changes in downstream splicing events, whereas neither upregulation nor splicing effects were observed in mice.

As the reviewer pointed out, NOVA1 expression has been reported in various protein/RNA atlases, such as glandular cells in gastrointestinal tract and follicle cells in ovary. These might relate to cross reacting reagents, trace amounts of expression in rare cells in these tissues, or changes in NOVA1 regulation.

2: A concern is that the vocalization effects may reflect general motoric effects of NOVA1^{hu/hu}. It would be good to test the mice in a battery of other standard tests of motor control and motor activity (e.g., rotarod, open field) to exclude that they have more generalized motoric issues. This is important especially as effects of NOVA1 mutations in humans have more generalized effects that are not restricted to speech and language.

We appreciate the reviewer's point. Our previous work has demonstrated that NOVA1 plays physiological roles in motor, memory and learning functions (both through patient and knockout mouse studies), so it is important to examine if the USV alteration is part of the motor, memory and learning phenotypes in our humanized NOVA1 mice. Below are the analyses we have performed to address the reviewer's concern.

We performed the Rotarod test to evaluate motor function and coordination in mice. To first examine the effect of NOVA1 itself in the rotarod test, we assessed the performance in inhibitory neuron-specific NOVA1 knockout (cKO; *Gad2^{Cre} NOVA1^{fl/fl}*) mice, a mouse model that mimics the phenotype of patients with NOVA1 haploinsufficiency (Tajima et al., 2023). After a habituation period, mice were measured for the time it takes to fall off the rod that accelerates at a constant rate (4 to 40rpm in 300sec). Tests were performed three times and the average value are calculated. These data demonstrated that cKO and control mice performed at comparable levels, indicating that mouse NOVA1 itself has no discernable effect on motor function in the rotarod test. We then assessed the performance in *Nova1^{hu/hu}* mice. *Nova1^{hu/hu}* mice and wild-type control mice performed at comparable levels in the rotarod test, indicating that there appears to be no effect of I197V substitution on motor performance. The results of the experiment show no statistically significant differences between genotypes, and we recognize results could likely be made even more robust by increasing the sample sizes (*Gad2^{Cre} NOVA1^{fl/wt}* N=4, *Gad2^{Cre} NOVA1^{fl/fl}* N=5, *Nova1^{wt/wt}* N=3, *Nova1^{hu/hu}* N=3).

Rotarod performance test. To measure the locomotor performance (motor coordination), mice are placed on an elevated revolving rod and the time it takes them to fall is recorded. The bar graph represent mean \pm standard deviation. The dot indicates average time for one mouse.
Gad2^{Cre}NOVA1^{fl/wt} N=4, *Gad2^{Cre}NOVA1^{fl/fl}* N=5, *NOVA1^{wt/wt}* N=3, *NOVA1^{hu/hu}* N=3.

We also performed Y-maze test, an assay commonly used to assess spatial working memory and reference memory, but which is also a measure of exploratory activity and locomotion, measured by the number of arm entries in the maze. The test is performed in a Y-maze with three arms of equal length at 120° angles to each other, which allows mice to explore freely. Mice naturally tend to explore the arm that was not visited immediately before, resulting in sequential visits to all three arms (alternation). A high percentage of alternation is considered to be indicative of high spatial memory.

The effect of NOVA1 itself in Y-maze test has been previously reported with inhibitory neuron-specific NOVA1 knockout (cKO) mice (Tajima et al., 2023). cKO mice show lower alteration rates with higher number of entries compared to control mice, indicating that loss of NOVA1 in inhibitory neurons negatively affects spatial memory, even as it results in an increase in activity /locomotion. This elevated activity in cKO mice is interpreted to be due to increased level of anxiety. *Nova1^{hu/hu}* mice have comparable alteration rates (a measure of spatial learning ability) and number of entries into arms (a measure of activity level) as wild-type mice, indicating no differences in learning ability or activity level in this test.

Y-maze test. (left) illustration of the test and calculation for the alternation rate. Mice were allowed to freely explore a Y-shaped maze for 8 minutes. The number of entries into the arms and the number of triads were recorded to calculate the percentage of alternation. Alternations are consecutive entries into each arm of the Y-maze without any repeats (e.g., arm1 -> 2 ->3). (middle) results in NOVA1 deficiency mouse model (cKO mice). (right) results in humanized NOVA1 mice. The alteration rates and total number of entries into arms during the tests are represented as boxplots with the minimum score, first quartile, median, third quartile, and maximum score. Each dot indicates data from a single mouse. *Gad2^{Cre}NOVA1^{fl/wt}* (cHet) N=7, *Gad2^{Cre}NOVA1^{fl/fl}* (cKO) N=8, *Nova1^{wt/wt}* N=18, *Nova1^{hu/wt}* N=23, *Nova1^{hu/hu}* N=18.

Notably, *Nova1^{hu/hu}* mice exhibit qualitative changes in vocal characteristics compared to control mice both in pup USVs and adult USVs, despite having comparable level of number of calls, which likely reflects motor function. These findings suggest that the vocalization changes in *Nova1^{hu/hu}* mice appear not to be the result of alterations in general motor performance. This does not exclude the possibility that the changes in USVs result from local motor function alterations. For instance, subtle changes in the lung function or the larynx, via neural projections to the surrounding muscles, could underlie the observed vocal changes. Given the current limitations in our understanding of the neurological and anatomical basis of mouse vocalization, further studies are necessary to identify the specific neural mechanisms responsible for the vocal changes observed in *Nova1^{hu/hu}* mice.

The results of Rotarod tests and Y-maze tests are included in the revised manuscript (Supplementary Fig. 17a-b), and the discussion is included as below.

Discussion

“*Nova1^{hu/hu}* mice exhibit qualitative changes in vocal characteristics compared to control mice both in pups and adults, despite producing a similar number of calls. These findings suggest that the vocalization changes in *Nova1^{hu/hu}* mice are not simply the result of alterations in general motor performance. This idea is supported by other observations showing that *Nova1^{hu/hu}* mice perform similarly to control mice in motor function tests, such as the rotarod, and display comparable locomotion activity levels in the Y-maze test (Supplementary Fig. 17). Additionally, the Y-maze test results indicated that *Nova1^{hu/hu}* mice had spatial working memory comparable to that of control mice.”

3: They should compare effects they observe on USVs with the ones described for the mice carrying the two substitutions in *Foxp2*, both for the pups and the adults (for which effects have been described contrary to what the authors claim). It would be valuable to not restrict the analysis of the adult vocalization to males but also include females (van Meurens et al., *Genes Brain Behavior* 2021).

We thank the reviewer for the suggestion. Following the reviewer's remarks, we have compared our vocalization study and findings with previous studies on humanized *Foxp2* mice (Enard et al., 2009, Hammerschmidt et al., 2015, von Merten et al., 2021). Before mentioning them, we would like to explain the revision and improvements we made in vocal analyses and findings (the comparison between studies is two pages later).

In this revision, we have conducted a comprehensive update of our vocalization analyses, by enhancing the statistical methods and conducted additional analyses to compare each vocal property in more detail, for each syllable type separately. Specifically, for each vocal property analysis, we noticed that the analysis in the first manuscript was insufficient in some parameters, which shows bimodal distributions; mice change their vocal characteristics depending on their stage of development, social context, and stimuli (Grimsley et al., 2011, Chabout et al., 2015, Lefebvre et al., 2020, Hammerschmidt et al., 2015).

For example, in terms of peak frequency (Fq), neonatal mice show a clear bimodal distribution between P5 to P9 (i.e., high and low Fq), which later converges to a single peak (Grimsley et al., 2011). Similarly, syllable duration can be divided into two distributions depending on its length, and each characteristic is influenced by genotype, vocal induction stimuli, or experience (Castellucci et al., 2016, Castellucci et al., 2018, Enard et al., 2009, Hammerschmidt et al., 2015).

(left) Bimodality in USV peak frequency during mouse development. Figure from Grimsley et al., 2011. There are two clear peaks in the distribution. Pup isolation induced calls in mice show a bimodal distribution in their frequency (Fq), especially at 5-9 days of age, and to consolidate to one peak as they grow (Grimsley et al., 2011).

(right) Example spectrogram showing short and long duration USVs in adult male ultrasonic courtship vocalizations. Figure from Castellucci et al., 2018.

In the initial manuscript, we had simply averaged the values in the individual mouse syllables without taking into account their properties. However, if the data are compared by taking the overall average for parameters that are indeed bimodal, the results strongly reflect the more weighted distribution, and will mask the real characteristics in each distribution. To analyze these parameters more accurately, we assessed each bimodality, measuring the degree of separation between two distributions, and determined mean, weight, and cutoff value of each distribution to classify syllables accordingly.

Example of analysis for bimodal distribution. (left) Density plot of the maximum frequency (Fq_{max}) of syllable "d" observed in pup USVs. (right) The same data with left. D (Ashman's D) score indicates the degree of separation between two distributions. $D > 2$ is an indicator of marked separation. To separate distributions, two Gaussians (blue and orange lines) fitted are overlaid. The Gaussian centers (blue and orange circles) and weights, the intercept of the two Gaussian distributions (black triangle) used as the cutoff between two distributions are labelled.

This method allowed for a more careful breakdown of mouse vocal characteristics and rigorous comparisons between genotypes. The major findings from these revisions are listed below.

*Major points in the revised USV analyses.

[Pup USV]

- *Nova1^{hu/hu}* produces significantly lower amplitude for 's' syllable.
- *Nova1^{hu/hu}* produces less syllables containing pitch jumps.
- *Nova1^{hu/hu}* had increased proportion of complex syllables with high maximum frequency (Fq_{max}).

[Adult USV]

- *Nova1^{hu/hu}* had significantly lower start, mean and end peak frequencies (Fq) in long 's' syllable.
- *Nova1^{hu/hu}* had increased frequency variance ($Fq_{variance}$) in complex syllables with high Fq_{max} .

- *Nova1^{hu/hu}* mice emitted a comparable number of vocalizations, having qualitative changes in some specific parameters both in pups and adults.
- Vocalization changes are specific to each developmental stage (or specific to social context).
- Heterozygous (*Nova1^{hu/wt}*) mice show intermediate values between *Nova1^{wt/wt}* and *Nova1^{hu/hu}* in both pup and adult USVs, suggesting that the effect of the I197V substitution in NOVA1 protein is dosage dependent.

We have updated the manuscript with these revised analyses and results (see Fig. 4, Supplementary Fig. 15, and main text for detailed description). To provide the reviewers and all readers access to the full vocalization data, we disclosed all raw data and associated tables in supplementary materials (Supplementary Table 6-16).

Following the reviewer's suggestion, we compared three reports that analyzed the vocalization in humanized *Foxp2* mice (Enard et al., 2009, Hammerschmidts et al., 2015, von Merten et al., 2021) to our results. Below is a table comparing the experimental methods, analytical methods, and findings of these studies.

[Comparison table of vocalization tests in humanized mouse models]

Comparison on USV tests between this study and three studies using humanized *Foxp2* mice (Enard et al., 2009, Hammerschmidts et al., 2015, von Merten et al., 2021). The table contains experimental conditions (methods and analysis) and findings in each study. To avoid changing the nuance of the words in each report, the terms in each paper were quoted verbatim in the table (e.g., calls, elements, vocalizations).

Study	Title (DOI)		
Enard et al., 2009	A Humanized Version of Foxp2 Affects Cortico-Basal Ganglia Circuits in Mice (https://doi.org/10.1016/j.cell.2009.03.041)		
Experiment			
Mouse	modification	Age	sex
humanized FOXP2	2 amino acid substitution in mouse FOXP2	pup (P4, P7, P10, P13)	male and female
Method	USV induction	recording time	
isolation-induced	isolate pup from their mother and nest	2 min (3 min for P13)	
Recording setting			
microphone	preamplifier, recording software, whistle tracking algorithm	Call count / structure analysis	extraction for acoustic parameters
UltraSoundGate CM16	UltraSoundGate 116, Avisoft SASLab Pro v4.33c	AVISOFT Recorder 2.97	custom software program LMA 2005
Representative parameter			
Number of calls, peak frequency (mean, start, end, max and min), location of the max frequency, greatest difference [Hz] in peak frequency, Slope of the call from start to the end peak frequency, slope of a linear trend through the peak frequencies of a call, modulation of calls (global/local)			
Syllable classification			
call type1: all calls that contained no or only minor frequency jumps, short call (<50ms); call type2: all calls that contained no or only minor frequency jumps, long call (>50ms); call type3: all calls with frequency jumps; call type4: remaining sounds (not analyzed)			
Representative findings			
No significant differences in the number of calls Foxp2hum/hum had a significantly lower start, mean, minimum and maximum peak frequencies in short 's' For short 's', the slope of the calls declined less in frequency in Foxp2hum/hum Calls with frequency jumps lasted longer, had longer gaps, and started and ended with higher peak frequencies in Foxp2hum/hum mice			

Study	Title (DOI)		
Hammerschmidt et al., 2015	A humanized version of Foxp2 does not affect ultrasonic vocalization in adult mice (10.1111/gbb.12237)		
Experiment			
Mouse	modification	Age	sex
humanized FOXP2	2 amino acid substitution in mouse FOXP2	adult (9-16 wks)	male (with female)
Method	USV induction	recording time	
courtship	place male mouse with female in the same box	120sec for habituation, 30sec pre-recording, 2min recording	
Recording setting			
microphone	preamplifier, recording software, whistle tracking algorithm	Call count / structure analysis	extraction for acoustic parameters
UltraSoundGate CM16	UltraSoundGate 116, Avisoft SASLab Pro v4.33/5.2	Avisoft Recorder 4.2	custom software program lma 2012
Representative parameter			
number of call elements, inter-element interval (ICI), element duration, amplitude gap, PF start, P1max, P1max location, P1jume, element slope			
Syllable classification			
category1: elements<60ms; category2: elements >60ms; category3: elements >=200ms. Also classified 19 element type for finer grained level of analysis, using a visual classification.			
Representative findings			
No significant differences in the number of elements Males homozygous for humanized Foxp2 produced call elements with slightly more pronounced frequency jumps and a slightly earlier frequency maximum			

Study	Title (DOI)		
von Merten et al., 2021	A humanized version of Foxp2 affects ultrasonic vocalization in adult female and male mice (10.1111/gbb.12764)		
Experiment			
Mouse	modification	Age	sex
humanized FOXP2	2 amino acid substitution in mouse FOXP2	adult (10-16wks)	male and female
Method	USV induction	recording time	
courtship (natural condition)	Natural vocalizations of males and females without physical contact with each other (but with visual, olfactory, and acoustic contact)	12h, during night (19pm-7am)	
Recording setting			
microphone	preamplifier, recording software, whistle tracking algorithm	Call count / structure analysis	extraction for acoustic parameters
CM16/CMPA	Avisoft UltraSoundGate 416Hm, Avisoft USGH recorder	Selena (University of Tübingen)	custom-written MATLAB script (SVM, SH)
Representative parameter			
number of songs: number of syllables, duration, syllables per second syllable: average, start, and minimum frequency, frequency bandwidth, duration, frequency slope, relative amplitude			
Syllable classification			
Three syllable groups: simple syllables (S), turn syllables (T) and jump syllables (J). Also classified 12 syllable types; depending on their frequency slope and existence of turning points and jumps			
Representative findings			
No significant differences in the number of calls Foxp2hum/hum mice showed a significantly larger variance in average number of songs Foxp2hum/hum mice emitted USV with significantly higher average and start frequencies, and larger frequency bandwidths (a consequence of the larger number of jump syllables) Foxp2hum/hum mice used more complex syllables containing frequency jumps and less simple or turn syllables Foxp2hum/hum mice used many transitions containing jumps (Female) Females and males contributed almost equally on number of songs and syllables during recording (Female) Females emitted USV syllables with smaller frequency bandwidths than males (Female) Females used less jump syllables than males, which fitting to the smaller average frequency bandwidth of female calls.			

Study	Title		
This study	A humanized NOVA1 splicing factor alters mouse vocal communications		
Experiment			
Mouse	modification	Age	sex
humanized NOVA1 (Nova1hu/hu)	1 amino acid substitution in mouse NOVA1	pup (P7)	male and female
humanized NOVA1 (Nova1hu/hu)	1 amino acid substitution in mouse NOVA1	adult (8-12wks)	male (with female)
Method	USV induction	recording time	
(pup) isolation-induced	isolate pup from their mother and nest	5min	
(adult) courtship	place male mouse with female in the same box	15min for habituation, 5min recording	
Recording setting			
microphone	preamplifier, recording software, whistle tracking algorithm	Call count / structure analysis	extraction for acoustic parameters
UltraSoundGateCM16/CMPA	Ultrasound Gate USGH amplifier, AvisoftRecorderUSG	Avisoft Bioacoustics	Mouse Song Analyzer 2' (MSA2)
Representative parameter			
number of syllable, each duration, bandwidth, amplitude, purity, frequency variable, start- minimum- maximum end-frequency			
Syllable classification			
Four syllable type: simple syllable (s), one downward pitch jump (d), one upward pitch jump (u), multiple pitch jumps (M). Also classified with each duration or peak frequency (parameter with bimodal distribution)			
Representative findings			
pup	No significant differences in the number of syllables Nova1hu/hu produced lower amplitude for 's' syllable Nova1hu/hu had less syllables containing pitch jumps NOVA1hu/hu had increased proportion of complex syllables with high maximum frequency (Fqmax)		
adult	No significant differences in the number of syllables Nova1hu/hu had lower start, mean and end peak frequencies (Fq) in long 's' syllable Nova1hu/hu had increased frequency variance (Fqvariance) in complex syllables with high Fqmax.		
Nova1hu/hu mice had comparable number of vocalizations, having qualitative changes in some specific parameters both in pups and adults. Vocal changes are specific to each developmental stage (or specific to social context). Heterozygous (Nova1hu/wt) mice show intermediate values between Nova1wt/wt and Nova1hu/hu in both pup and adult USVs.			

Across the four studies, we consistently found no significant differences in the number of vocalizations, while qualitative changes in vocalizations were observed in either group of mice. Each qualitative change seems to be specific to each experiment (although there seem to be similarity as described below), suggesting these effects on vocal characteristics are age, sex or social context dependent.

Similarities:

- A decrease in the peak frequency in 's' syllable (context-dependent)
- Modulation in peak frequency in complex (jump containing) syllables (especially in the high frequency region).

Comparing observations across studies is challenging due to variations in experimental conditions: genes, age, sex, stimulation, and syllable classification. Nonetheless, it is noteworthy that similar changes—a decrease in peak frequency in certain syllables and alterations in the high-frequency region in jump syllables—were observed in both pup and adult ‘humanized’ mice. These may indicate that there may be a common or related molecular alteration in the neural circuits involved in the USV production between these mice. The physiological significance of these changes in USVs should be investigated in the context of social behavior tests in future studies.

In our adult vocalization test, we used a method very similar to that of Hammerschmidts et al., which does not account for female vocalizations during recording. This approach is based on earlier reports that USVs produced during brief recordings in courtship-induced vocalization tests are predominantly produced by male mice (Whitney et al., 1973). Since submission of our manuscript, a study from the Jarvis lab in our group submitted a preprint of another study showing that with individual microphones attached to each mouse, in the first 5 min of introduction, the male produces over 90% of the USVs (Waidmann et al 2024, bioRxiv). We now cite this study for such an estimate of the percentage of vocalizations produced by the males and females.

However, several studies have shown that female mice also vocalize under certain experimental conditions or social contexts (von Merten et al., 2021, Neunuebel et al., 2015, Moles et al., 2007). In longer recordings, such as 12 hours in a more natural co-housing environment, female mice produce similar number of vocalizations, though with different characteristics compared to males (e.g. less jump syllables) (von Merten et al., 2021). In our study, we used wild-type female mice for courtship induction and were unable to distinguish the USVs produced from different sexes. Although the contribution from females is considered less than that from males, we cannot exclude the possibility that mixed vocalizations may obscure some of the specific characteristics in male vocalizations across genotypes. Furthermore, it should be noted that we have not addressed the effect of the I197V substitution on the vocalizations of adult female mice. Given that our experimental setup could not distinguish USVs from males and females, future studies with the capacity for separate analysis are necessary to investigate the sex-specific effects of the I197V substitutions during courtship-induced vocalization tests.

We have added the comparison table (Supplementary Table 17) and above consideration to the Discussion section in the revised manuscript.

4: Are the 27 of 630 transcripts in the category “Behavior” more than expected by chance? This can be tested by resampling many sets of 630 transcripts to see how often 27 or more are associated with the category “Behavior”.

According to the reviewer’s suggestion, we performed the analysis as follows:

We conducted 1,000 resamplings of 630 transcripts detected in our RNAseq dataset to determine how many are annotated under the “behavior” category in gene ontology.

The results showed a mean of 21.6 transcripts, with a median of 21 and a third quartile was 24. In these trials, the number of 27 transcripts was a probability of 9.9%. Thus, the number of transcripts detected in this study is higher than the average number of transcripts detected by chance.

We conducted the same analysis for vocalization related transcripts. In our study, four of 27 transcripts are associated with the vocalization behavior category. We performed 1,000 resampling of 27 transcripts from the 843 transcripts annotated under the behavior category in the gene ontology database and calculated how many were annotated in the vocalization category.

The mean number of transcripts detected by resampling was 0.711, with a median of 1. The probability of detecting the four vocalization-related transcripts is less than 4%. Therefore, the number of vocalization-related transcripts detected in this study is significantly higher than what would be expected by chance.

These analyses are included in the Supplementary Figure 13 in the revised manuscript.

5: The evolutionary analyses they present (pages 3; lines 12-42) are extremely similar to the ones presented in reference 12. To claim that a “strong evolutionary selective sweep” affected the NOVA 1 region, we would like to see further analyses. Negative Tajima’s D and low allele frequencies of minor SNPs could be explained by purifying selection as well as selective sweeps. If a strong selective sweep

affected the region, we would expect other measures such as extended haplotype lengths and reduced intra-population diversity relative to divergence to indicate this.

We thank the reviewer for the suggestion. To examine the selective sweep on the *NOVA1* genomic region, we first conducted an additional statistical analysis, DH test (Zeng et al., 2016). Both Tajima's D and Way and Fu's H are powerful for detecting selection, but as the reviewer points out, Tajima's D (and Way and Fu's H) detects various population genetic forces, are sensitive to different demographic factors and are affected by background selection to different degrees. As a circumvention to this issue, DH test is reported to be effective in detecting selection while being insensitive to other perturbations. The significance in DH test suggest that the region is under the recovery phase after the loss of genetic diversity, which includes the post-selective sweep phase (Zeng et al., 2016).

Using human genetic data from the 1000 Genomes Project, we calculated the DH value for the *NOVA1* gene locus and assessed its significance among the genes on the same chromosome (chr14). *NOVA1* gene locus has value of 0.42642643, reaching the 5% significance level, whereas adjacent genes did not reach significance (FOXG1: 0.90390, STXBP6: 0.77477).

gene	position (GRCh37/hg19)	pi	thetaW	DHtest	percent.rank	significance
NOVA1	chr14:26,912,296-27,067,239	0.00063633	0.00294276	0.42642643	0.046546547	TRUE
FOXG1	chr14:29,235,993-29,239,483	0.00017486	0.00126671	0.9039039	0.806306306	FALSE
STXBP6	chr14:25,278,660-25,519,353	0.00103931	0.00310043	0.77477477	0.551051051	FALSE

In response to the reviewer's suggestion regarding the extended haploid length, we refer to the analysis and results of Trujillo et al. (2021). Their target variant is the same as ours (rs762662114, chr14:26448894/ GRCh38.p14). Trujillo et al. measured the length of the human-specific haplotype around the *61 fixed human-specific sites — the largest genomic region where no humans carry substitutions observed in the archaic genomes. (*61 human-specific positions were identified through a genomic comparison between humans, Neanderthals and Denisovans.) They found that among these 61 nonsynonymous variants, the human-specific *NOVA1* substitution is located on the third-largest human-specific haplotype. Trujillo et al. also analyzed genetic variation within the human-specific *NOVA1* haplotypes and identified two high-frequency haplotypes. The most common haplotype carries only the fixed nonsynonymous substitution (corresponding to I197V), while the second most common carries a single derived allele in addition to the fixed nonsynonymous substitution. Their analysis suggests that these haplotypes emerged relatively recently and then spread to fixation. They also measured Tajima's D statistic for these human-specific haplotypes and reported that the Tajima's D' for the *NOVA1* region in humans is the fourth-lowest among the haplotypes surrounding nonsynonymous substitutions, consistent with their haplotype analysis.

Fig. 1.

Human versus Neanderthal genetic variation and the NOVA1 haplotype. Figure from Trujillo et al., 2021.

(A) Physical haplotype lengths (bp) around human-specific fixed derived alleles in the 1000 Genomes Project dataset. They defined a haplotype as the distance upstream and downstream of a human-specific fixed derived allele for which no human in the 1000 Genomes Project dataset shares a derived allele with an archaic hominin. Lengths of haplotypes around both synonymous and nonsynonymous substitutions are shown. (B) Haplotypes around the human-specific fixed derived allele in NOVA1. Rows are individual haplotypes; columns are variable sites. Yellow boxes have a derived allele (different from the 1000 Genomes Project ancestral sequence). Human haplotypes are labeled by the number of (phased, haploid) human genomes that carry them. Only biallelic SNPs with reference alleles are shown, and the region is bounded by sites at which modern humans share derived alleles with archaic hominins. (C) Normalized Tajima's D of haplotypes around human-specific fixed derived alleles.

To further evaluate the hypothesis that the NOVA1 197V variant have been part of a selective sweep, we performed an analysis of the ancestral recombination graph (ARG) for modern and archaic hominins in the surrounding region, leveraging ARGs previously inferred using *ARGweaver-D* (Hubisz et al., 2020). These ARGs explicitly describe gene trees and accompanying recombination events throughout the region. They were sampled from an approximate posterior distribution using Markov chain Monte Carlo methods to capture uncertainty in the ARG, given the sequence data and evolutionary parameters. The sampled ARGs included two Yoruba (HGDP00927, SS6004475), two Mbuti (SS6004471, HGDP0456), and two San (HGDP01029, SS6004473) individuals, all sequenced to high coverage, as well as the Altai Neanderthal and Denisovan sequences and a chimpanzee outgroup (panTro4). Because the NOVA1 197V variant is nearly fixed in modern humans and likely predates the separation of major continental population groups, we do not anticipate significant additional insights from including more modern human samples, as most would coalesce well after the allele reached high frequency.

Focusing on the NOVA1 197V variant, we analyzed the sampled ARGs using *CLUES2* (Vaughn et al., 2024), a method that estimates a selection coefficient to best explain observed changes in allele frequency over time, based on indirect information from the ARG. *CLUES2* employs a full likelihood approach, reweighting a collection of ARG samples by importance sampling to allow for selection. It provides an estimated selection coefficient for the specified derived allele, along with a likelihood ratio and significance level. Our analysis revealed that selection at the NOVA1 SNP is relatively strong and statistically significant, with an estimated selection coefficient of $s = 0.00082$ ($p=0.019$ for the null hypothesis of no selection). While this estimate is an order of magnitude less than those observed at the strongest sweeps in the human genome (such as *LCT*, which has $s \cong 0.01$ [Hejase et al., 2022]), it is still substantial, corresponding to a population-scaled coefficient of $S = 2N_e s \cong 19$, indicating strong selection relative to nearly neutral evolution (where $|S| \leq 1$).

Recent burst of coalescence of a sweep for NOVA1 197V. The left tree shows the NOVA1 SNP, with the shortened branches and burst of relatively recent coalescence events leading to the modern humans (reflecting a rise in frequency of the derived allele). The right tree shows FRMD8 for contrast, with longer branches and delayed coalescence: more typical of what would be expected in the absence of a selective sweep. The sampled ARGs included two Yoruba (HGDP00927, SS6004475), two Mbuti (SS6004471, HGDP0456), and two San (HGDP01029, SS6004473) individuals, as well as the Altai Neanderthal and Denisovan sequences and a chimpanzee outgroup (panTro4). The numbers following the samples represent the separate haplotypes from each individual. Red lines indicate derived allele.

For comparison, we applied the same *CLUES2* analysis to 38 other SNPs for which we could obtain informative ARGs, from previously identified as potentially selected in the human genome (Trujillo et al., 2021). The results showed that the selection observed at the NOVA1 SNP is relatively strong when compared to these other genes, with 33 showing either non-significant results or smaller selection coefficients.

Gene name	Ensembl Gene ID	chr (hg19)	position (hg19)	logLR	-log10(p-value)	Epoch1_start	Epoch1_end	SelectionMLE1
HERC5	ENSG00000138646	4	89410317	5.5077	3.04	0	200000	0.00135
AHR	ENSG00000106546	7	17375392	6.2143	3.37	0	200000	0.00127
LAG3	ENSG00000089692	12	6883790	3.9207	2.29	0	200000	0.00097
C3	ENSG00000125730	19	6685111	3.8964	2.28	0	200000	0.00089
ZNF2	ENSG00000163067	2	95831534	3.8143	2.24	0	200000	0.00087
NOVA1	ENSG00000139910	14	26918100	2.7454	1.72	0	200000	0.00082
SSH2	ENSG00000141298	17	27959034	2.739	1.72	0	200000	0.00057
SSH2_2	ENSG00000141298	17	27959258	2.7372	1.71	0	200000	0.00057
IFI44L	ENSG00000137959	1	79106805	2.5711	1.63	0	200000	0.00054
KIAA1199	ENSG00000103888	15	81173308	2.0899	1.39	0	200000	0.00052
FAM166A	ENSG00000188163	9	140139881	2.4931	1.59	0	200000	0.00051
KIF18A	ENSG00000121621	11	28119295	2.3184	1.5	0	200000	0.00051
ZNF106	ENSG00000103994	15	42742312	2.2816	1.49	0	200000	0.00051
CDH16	ENSG00000166589	16	66947064	2.1992	1.44	0	200000	0.00051
KIF26B	ENSG00000162849	1	245582905	1.6157	1.14	0	200000	0.00049
ITGB4	ENSG00000132470	17	73753035	1.7434	1.21	0	200000	0.00048
PROM10	ENSG00000170325	11	129772293	1.8097	1.24	0	200000	0.00038
NLRP2	ENSG00000022556	19	55489189	1.8316	1.25	0	200000	0.00037
ADSL	ENSG00000239900	22	40760978	1.6985	1.18	0	200000	0.00036
ADAM18	ENSG00000168619	8	39537618	1.6707	1.17	0	200000	0.00035
VCAN	ENSG00000038427	5	82837946	1.6271	1.15	0	200000	0.00034
SCAP	ENSG00000114650	3	47469149	1.5695	1.12	0	200000	0.00033
OR1K1	ENSG00000165204	9	125563200	1.5636	1.11	0	200000	0.00033
DNAJC11	ENSG00000007923	1	6694660	1.52	1.09	0	200000	0.00032
ADAM18_2	ENSG00000168619	8	39564352	1.2405	0.94	0	200000	0.00031
DCHS1	ENSG00000166341	11	6654769	1.3738	1.01	0	200000	0.0003
PIG2	ENSG00000119227	3	196674495	1.0181	0.81	0	200000	0.00024
ZNHIT2	ENSG00000174276	11	64884957	1.0153	0.81	0	200000	0.00021
MRPL49	ENSG00000149792	11	64893151	1.0113	0.81	0	200000	0.00021
PIEZO1	ENSG00000103335	16	88804443	0.9977	0.8	0	200000	0.00021
AC074212.3	ENSG00000237452	19	46265288	0.9664	0.78	0	200000	0.00021
CASC5_2	ENSG00000137812	15	40915640	0.9432	0.77	0	200000	0.00021
CASC5	ENSG00000137812	15	40912860	0.943	0.77	0	200000	0.00021
NOTO	ENSG00000214513	2	73438011	0.9422	0.77	0	200000	0.00021
NCOA6	ENSG00000198646	20	33337529	0.9277	0.76	0	200000	0.0002
ZNF726	ENSG00000213967	19	24116551	0.732	0.65	0	200000	0.00018
CCT5	ENSG00000150753	5	10250094	0.7118	0.63	0	200000	0.00013
TEX2	ENSG00000136478	17	62290457	0.6899	0.62	0	200000	0.00012
FRMD8	ENSG00000126391	11	65154602	0.5303	0.52	0	200000	0.00011

Table presents the results of selection analysis for various SNPs using *CLUES2*. The columns display the log-likelihood ratio (logLR), the negative log10-transformed p-value (-log10(p-value)), and the selection coefficient (SelectionMLE1) for each SNP across a single epoch (0 to 200,000). The table is sorted by p-value, with SNPs highlighted in yellow, including NOVA1, showing stronger selection signals.

The observation that this allele in NOVA1 is shared and nearly fixed across human population groups suggests that it occurred and increased to high frequency prior to the divergence of these groups. The possibility that independent sweeps occurred in all populations cannot be strictly ruled out, but given its universal high frequency, it seems highly unlikely. In contrast, most of the strongest sweeps identified in humans are clearly population-specific and appear to be quite younger (LCT estimates of around 10 kya). One thing to note is that the best-known sweeps, such as LCT, may simply be the ones we have the best power to detect. A population-specific sweep leaves a very clear signature in the pattern of genetic variation, whereas an old sweep shared across populations leave a more obscure pattern and may require novel methods to detect.

Collectively, these findings support the notion that the NOVA1 197V variant was part of a selective sweep in modern humans, predating many other identified sweeps in the human genome. These additional statistical analyses and descriptions are now included in Supplementary Figure 3, Supplementary Table 2, and in the main text.

6: Differential expression analysis has been done in mouse embryo at day 18.5. CLIP and AS have been done on 3-week-old mice. Why use different mouse ages for different assays? This is unfortunate, as it is not possible to say if the genes that come up in CLIP and AS would come up as differentially expressed genes. If the authors would like to perform these analyses at the same time, it would be valuable (but not required in our opinion).

Following the reviewer's suggestion, gene expression analyses of the midbrain, cortex, and cerebellum of P21 were added in addition to the E18.5 midbrain dataset in the revised version (Figure 2c, Supplemental Figure 5a-d). These comprehensive analyses reveals that the transcript levels, including of *Nova1* itself, were nearly identical between genotypes throughout development (Fig. 2c), as well as in different brain regions (Supplementary Fig 5a-d). These data suggest that transcript levels are unlikely to have contributed to the change in NOVA1 binding to transcripts or changes in alternative splicing. These revised analyses are now included in the main text.

Minor comments

Line 316: “with two substitutions enriched in humans” should be “two substitutions specific for humans”.

Thank you for the point. We have corrected the sentence.

Line 376: a comma is not needed.

Thank you for the point. We have corrected the sentence.

line 387: “which they term I200V accounting for an AS variant”. Here, a transcript ID should be given if there is one.

We checked all currently registered human and mouse NOVA1 isoforms (based on current NCBI database, NOVA1 gene ID 4857 (human), gene ID 664883 (mouse)) and found no isoforms whose target site corresponds to the 200th amino acid. The longest (major) NOVA1 amino acid at position 200 is glutamine (Q) in both human and mouse. Our assumption was that they were using their own dataset or are referring to an unregistered isoform. Although their target site is the same as ours (GRCh37/hg19, chr14:26,918,100), since we could not obtain confirmation that it was a splice variant, we corrected this sentence and limit to describing “I197V variant (term I200V in one study¹⁴)”.

Line 474: “8 animals harbored the designed allele (one missense substitution and two nonsense substitution)” is unclear. Did one allele have three substitutions? Probably not.

Yes, this is how we designed the I197V substitution. Please refer to the Supplementary Figure 4a-b. One nucleotide substitution (A to G) is responsible for an amino acid substitution (missense: I197V); the other surrounding two nucleotide substitutions (T to A and A to G, both nonsense changes) were introduced to create a restriction enzyme (*BtsI* α) recognition site that can be used for genotyping. To make this point clearer, the text was changed in the revised manuscript as follows.

Methods section:

Generation of *Nova1*^{hu/hu} mice

“8 animals harbored the designed allele (with three nucleotide substitutions: one causing I197V amino acid substitution, two for restriction enzyme recognition site for genotyping (not causing amino acid changes))”

Reviewer comments and Response (Response are in blue)

Reviewer #2 (Remarks to the Author)

This is an exciting manuscript describing the “humanization” of the NOVA1 gene in mice. NOVA1 is an RNA binding factor with a critical role during neurodevelopment. Interestingly, NOVA1 is one of the genes that contain genetic variants unique to modern humans (not even present in Neanderthals, our closest relatives). The group used genome editing technologies to change a single amino acid (I197V) in mice and observed changes in gene expression and alternative splicing. Remarkably, the humanized animal showed significant alterations in vocalization, both in pups and adults. The work supports the idea that this genetic variant was positively selected in Homo sapiens, likely due to language development. The observation that a humanized splice factor can affect vocalization in mice is quite an amazing finding. Overall, I find the work quite impressive, both in terms of its technical aspects and its message and contributions to human evolution. While I am enthusiastic and supportive of the publication of these findings, several aspects need clarification. I have listed below a series of suggestions to improve the manuscript, from the choice of NOVA1 as a candidate gene for these experiments to some more technical questions.

We sincerely appreciate the reviewer’s thoughtful reading of our manuscript and are delighted that they found our findings exciting. We have carefully considered the reviewer’s insightful comments and hope that our responses, along with the revisions in the manuscript, address their concerns and questions.

-The introduction has a detailed summary of human-specific genes, which I appreciated. However, the reason why they chose NOVA1 is not immediately clear; this information will emerge later. Below, I have some suggestions on how this could appear early in the manuscript.

-They use reference #2 to illustrate how humans are the only primates capable of vocal learning. However, this reference seems related to mouse vocalization and does not support that idea.

We thank the reviewer for the comment. With respect to human vocal communication, we have revised the text in terms of its anatomical characteristics and neural circuitry, and cited appropriate references as follows.

Introduction

“Humans differ significantly from their closest living relatives, the great apes, particularly in their ability to communicate through complex learned vocal communication, a necessary component of spoken language³(Jarvis et al., 2019). This complexity is driven by some anatomical adaptations of the vocal tract and intricate neural networks linking various brain regions³⁻⁷(Fitch et al., 2016, Boë et al., 2017, Nishimura et al., 2022, Jürgens et al., 2002).”

-When discussing FOXP2, previously called a “human language gene”, the authors should point out that reference #6 has actually found alterations in mouse vocalization. Later, reference #8 goes against it. The difference between these two works should be highlighted as this work also uses mice vocalization as a readout, and readers need to understand why the previous work has observed a significant difference that gives rise to the misconception that FOXP2 was uniquely human.

Regarding the vocal analysis of humanized FOXP2 mice, we have highlighted the two previous works (Enard et al., 2009, Hammerschmidt et al., 2015) pointed out by the reviewer, and also added a more recent work from von Merten et al. (von Merten et al., 2021) in which they reported qualitative changes in humanized *Foxp2* adult mice under a more natural vocalization paradigm. The seemingly contradictory results between these reports in mouse vocalization experiments may be due to the different experimental paradigms (age of mouse, vocalization induction method, sex). We performed detailed comparisons between these studies. This is explained later in the response letter (page 26-28).

We have described the results from these studies in the Introduction as follows.

Introduction

“The transcription factor forkhead box P2 (FOXP2) is of particular interest as a potential driver of human language function, as it harbors two amino acid substitutions present in human but not in chimpanzee and many other mammal genomes.

...Studies on mice with the two amino acids substituted to the human version have reported vocal changes both in the neonatal²⁰ and adult stages^{21,22}. While Hammerschmidt et al. observed minimal vocal changes, von Merten et al. reported qualitative changes under a more natural vocalization paradigm^{21,22}, suggesting the involvement of these two amino acids in vocalization.”

-Similarly, reference #2 (Enard et al., 2002) has found evidence for positive selection of FOXP2, but that was later debunked by reference #12. The reasons these two groups have divergent conclusions about FOXP2 should also be stated in the introduction.

-More recently, another gene (TKTL1) was shown to have human-specific consequences, but again, that was also debunked, as the variant is also present in modern human populations. The sentence "this substitution is present in a small percentage (0.03-0.2%) of modern humans" refers to Pinson et al (16), but the correct reference is from Herai et al., 2023 ("Comment on "Human TKTL1 implies greater neurogenesis in frontal neocortex of modern humans than Neanderthals". Science. 2023").

We appreciate the reviewer for the accurate and precise remarks.

The report of Enard et al. 2002 and the revision by Atkinson et al. 2018 on the positive selection of *Foxp2* are now cited in the revised manuscript. We also correctly cited the report of Pinson et al. 2022 and the comments on TKTL1 from Herai et al. 2023.

-Both FOXP2 and TKTL1 have failed to be validated as modern human-specific genes, which seems to be because a uniquely European-biased dataset was used as representative of all modern humans (on references #12 and #30). This also needs to be stated in the introduction to avoid further misinterpretations.

We appreciate the reviewer's important remarks on comparative genomic studies of modern humans. We have revised and emphasized the importance of using genomic data sets from a larger and diverse sample as follows.

Introduction

“The closest evolutionary relatives of modern humans are two extinct lineages: Neanderthals and Denisovans. Genome sequencing from fossilized remains of these archaic humans has identified distinct genetic differences between them and modern humans, which may be relevant to recent human evolution⁹⁻¹². Additionally, the availability of extensive human genome data over the past few decades, initially focused on European populations, has significantly expanded the scope of evolutionary studies¹³⁻¹⁵(Atkinson et al., 2018, Trujillo et al., 2021, Herai et al., 2023).

... The transcription factor forkhead box P2 (FOXP2) is of particular interest as a potential driver of human language function, as it harbors two amino acid substitutions present in human but not in chimpanzee and many other mammal genomes.

... However, these substitutions are also present in archaic humans, and comprehensive analyses using diverse human genome datasets have found no evidence of recent selection. This suggests that the FOXP2 substitutions occurred earlier than initially thought^{13,23}(Atkinson et al., 2018, Krause et al., 2007). Similarly, the TKTL1 gene contains a human-specific amino acid thought to influence greater neurogenesis in human than Neanderthal frontal cortex, though this finding is based on European ancestry genome datasets²⁴(Pinson et al., 2022). Broader analyses of modern human genomes reveal that 0.03-0.2% of individuals possess the ‘putative Neanderthal variant’, indicating its presence in a significant portion of the population¹⁵(Herai et al., 2023). These findings underscore the importance of incorporating diverse human samples to identify and validate the genetic background of modern human traits through genomic comparisons.”

-Thus, the paragraph: “Comparison of Neanderthal and Denisovan genomes to modern humans identified 260 human-specific single nucleotide changes (SNCs) that result in amino acid substitutions¹⁰. Of the 23 changes that were most conserved, eight were involved in brain function and neural development. One of the eight was human-specific isoleucine to valine substitution at position 197 (I197V) in the neuron-specific RNA binding protein neuro-oncological ventral antigen1 (NOVA1)”, needs to be updated. The more recent work on reference #30 actually uses several modern human datasets to narrow the number of human-specific genes to 61.

As the reviewer points out, we now have clearly noted that our study subject (NOVA1 I197V) was on the list of modern human-specific variants in a recent comparative analysis using a broader population data set, performed in Trujillo et al., 2021.

-The NOVA1 variant is, indeed, one of the 61 genes uniquely present in modern humans. Moreover, reference #30 showed that the archaic variant alters neurodevelopment in humans. Thus, this information should appear up front as a rationale for why the authors focused on this gene.

We appreciate the reviewer's suggestion. We agree that updating our description on more recent work: ref #30 (Trujillo et al., 2021), a study that highlights the strong selection pressure on the NOVA1 variant in modern humans.

At the same time, we are also aware that there is controversy regarding the effects of this single amino acid substitution in human iPSC-derived organoids (also pointed out by Reviewer#1 on Maricic et al., 2021 and Riesenbergr et al., 2023) and hence, would prefer not to stress the biological effect of NOVA1 variant in humans here. We anticipate that these controversies likely reflect differences in various experimental conditions (culture, clone, genetic background etc.) indicating difficulties in verifying the effects of single amino acid substitutions *in vitro*.

In light of the considerations above, the main text has been updated as follows.

Introduction

“Genomic comparisons between archaic humans, ape genomes, and the broader human population have identified 61 human-specific nonsynonymous coding variants that are fixed or nearly fixed in modern humans¹⁴. Notably, one of the genes includes an isoleucine to valine substitution at position 197 (I197V) in the RNA binding protein neuro-oncological ventral antigen1 (NOVA1).

... Studies have explored the NOVA1 I197V variant by reverting the ancestral isoleucine 197 variant back into human iPSC-derived organoids, revealing morphological and electrophysiological changes *in vitro*¹⁴. However, these effects were not observed in another study that reintroduced the same substitution in different iPSCs⁴⁰. This discrepancy underscores challenges of obtaining consistent results with varying experimental methods and materials *in vitro*^{41,42}.”

-The sentence “An expanded genomic analysis of the dbSNP database revealed that this substitution was present in all but six of 650,058 human sequences, five of which were from individuals of Asian descent” is related to homozygote or heterozygote individuals? What are these frequencies?

Because individual information was de-identified, zygosity information could not be obtained. For the alternate allele frequency by each project, please see the below:

=====

rs762662114

Homo sapiens, chr14:26448894 (GRCh38.p14),
Alleles: C>T, Type: SNV Single Nucleotide Variation

Allele Frequency (alternate allele / total allele number, project name):

T=0.000004 (1/264690, TOPMED)
T=0.000012 (3/250246, GnomAD_exome)
T=0.000017 (2/120984, ExAC)
T=0.000000 (0/14050, ALFA)
T=0.00 (0/88, Ancient Sardinia)

[dbSNP, Build 156, Released September 21, 2022]

=====

We have added this information in the revised manuscript (Supplementary Figure 1a).

-The group has generated NOVA1wt/wt and NOVA1hu/hu. NOVA1ko/ko is lethal, but what is the behavior of NOVA1wt/hu or NOVA1ko/hu? Is a single human allele enough to drive the phenotypes described here?

We appreciate the important questions from the reviewer.

We have not yet crossed humanized NOVA1 allele with knockout lines, and therefore cannot describe *Nova1^{hu/ko}* phenotype at the moment. Based on the observation of a patient with NOVA1 haploinsufficiency with several neurological abnormalities (Tajima et al., 2023), the *Nova1^{hu/ko}* mouse may be an interesting model that more closely resemble the allele situation of a human NOVA1 haploinsufficiency. We hope to examine this mouse model in future experiments.

We have collected vocalization data on *Nova1^{hu/wt}* mice simultaneously with *Nova1^{wt/wt}* and *Nova1^{hu/hu}* in vocalization studies. In the vocalization test, *Nova1^{hu/wt}* mice (having a single human allele) show intermediate values between *Nova1^{wt/wt}* and *Nova1^{hu/hu}* mice in the parameters that shows difference in *Nova1^{hu/hu}* mice (Figure 4, Supplementary Figure 15a).

For example, in pup USVs, *Nova1^{hu/hu}* mice have lower percentage of jump syllables than *Nova1^{wt/wt}*, and increased ratio in their higher peak frequency regions. *Nova1^{hu/wt}* mice had values intermediate between the two on all these parameters, although due to the large variability of values in the experiments, the *Nova1^{hu/wt}* values did not pass statistical significance for many case (*Nova1^{hu/wt}* vs. *Nova1^{wt/wt}* or *Nova1^{hu/wt}* vs. *Nova1^{hu/hu}*).

Pup isolation induced vocalization (Excerpt)

Example for the vocal phenotypes in NOVA1^{hu/wt} mice in pup isolation induced vocalization test. (left) Syllable composition (right) Ratio of each syllable in high or low peak frequency. Data are represented as boxplots. Each open circle indicates data from a single pup. p-values were calculated by Wilcoxon rank sum test and corrected with Bonferroni method. *p value <0.05, **p value <0.01. *Nova1^{hu/hu}* N=41, *Nova1^{hu/wt}* N=23, *Nova1^{wt/wt}* N=40 pups.

In Adult USV, decrease in peak frequency was observed for *Nova1^{hu/hu}* mice in long duration 's' syllable, and increase in frequency variances in jump syllables in the higher frequency region. For these parameters, *Nova1^{hu/wt}* mice show intermediate values between *Nova1^{wt/wt}* and *Nova1^{hu/hu}*.

Adult courtship vocalization (Excerpt)

Example for the vocal phenotypes in *Nova1^{hu/wt}* mice in adult courtship vocalization test. (left) Peak frequency parameters (start, minimum, mean, maximum and end) in long duration s. (right) Frequency variance (Fq variance) in high Fq_{max} USVs. Data are represented as boxplots. Each open circle indicates data from one adult male mouse; the average of the three recordings (see method section for details). *p*-values were calculated by Wilcoxon rank sum test and corrected with Bonferroni method. **p* value < 0.05, ***p* value < 0.01. *Nova1^{hu/hu}* N=13, *Nova1^{wt/wt}* N=14, *Nova1^{hu/wt}* N=13.

Overall, *Nova1^{hu/wt}* mice tend to show intermediate values between *Nova1^{wt/wt}* and *Nova1^{hu/hu}* in both pup and adult USVs, suggesting that the effect of the I197V substitution in NOVA1 protein is dosage dependent.

We have now included all USV data of heterozygous *Nova1^{hu/wt}* mice in the revised manuscript. We also have revised/ updated USV analyses, which are explained in the next response.

-The vocalization phenotypes are pretty amazing. When discussing the data, the authors should revisit the differences observed in references #6 vs. #8 and situate their own methods. This seems quite important to avoid another scientific fiasco, as others might want to replicate the author's mice observations.

We thank the reviewer for this important suggestion. Following the reviewer's suggestion, we have compared our methods and results with those of previous reports that performed vocal analyses on humanized *Foxp2* mice (Enard et al., 2009, Hammerschmidt et al., 2015, von Merten et al., 2021). Before mentioning them, we would like to explain the revision and improvements we made in vocal analyses and findings (the comparison between studies is two pages later).

In this revision, we have conducted a comprehensive update of our vocalization analyses, by enhancing the statistical methods and also conducted additional analyses to compare each syllable type property in more detail. Specifically, we noticed that the analysis in the first manuscript was insufficient in some parameters, which shows bimodal distributions; mice change their vocal characteristics depending on their stage of development, social context, and stimuli (Grimsley et al., 2011, Chabout et al., 2015, Lefebvre et al., 2020, Hammerschmidt et al., 2015). For example, in terms of peak frequency (Fq), neonatal mice show a clear bimodal distribution between P5 to P9 (i.e., high and low Fq), which later converges to a single peak (Grimsley et al.,

2011). Similarly, syllable duration can be divided into two distributions depending on its length, and each characteristic is influenced by genotype, vocal induction stimuli, or experience (Castellucci et al., 2016, Castellucci et al., 2018, Enard et al., 2009, Hammerschmidt et al., 2015).

(left) Bimodality in USV peak frequency during mouse development. Figure from Grimsley et al., 2011. There are two clear peaks in the distribution. Pup isolation induced calls in mice show a bimodal distribution in their frequency (F_q), especially at 5-9 days of age, and to consolidate to one peak as they grow (Grimsley et al., 2011).

REDACTED

REDACTED

(right) Example spectrogram showing short and long duration USVs in adult male ultrasonic courtship vocalizations. Figure from Castellucci et al., 2018.

In the initial manuscript, we had simply averaged the values in the individual mouse syllables without taking into account their properties. However, if the data are compared by taking the overall average for parameters that are indeed bimodal, the results strongly reflect the more weighted distribution, and will mask the real characteristics in each distribution. To analyze these parameters more accurately, we assessed each bimodality, measuring the degree of separation between two distributions, and determined mean, weight, and cutoff value of each distribution to classify syllables accordingly.

Example of analysis for bimodal distribution. (left) Density plot of the maximum frequency ($F_{q_{max}}$) of syllable "d" observed in pup USVs. (right) The same data with left. D (Ashman's D) score indicates the degree of separation between two distributions. $D > 2$ is an indicator of marked separation. To separate distributions, two Gaussians (blue and orange lines) fitted are overlaid. The Gaussian centers (blue and orange circles) and weights, the intercept of the two Gaussian distributions (black triangle) used as the cutoff between two distributions are labelled.

This method allowed for a more careful breakdown of mouse vocal characteristics and rigorous comparisons between genotypes. The major findings from these revisions are listed below.

*Major points in the revised USV analyses.

[Pup USV]

- *Nova1^{hu/hu}* produces significantly lower amplitude for 's' syllable.
- *Nova1^{hu/hu}* produces less syllables containing pitch jumps.
- *Nova1^{hu/hu}* had increased proportion of complex syllables with high maximum frequency ($F_{q_{max}}$).

[Adult USV]

- *Nova1^{hu/hu}* had significantly lower start, mean and end peak frequencies (F_q) in long 's' syllable.

- *Nova1^{hu/hu}* had increased frequency variance (Fq_{variance}) in complex syllables with high Fq_{max} .
- *Nova1^{hu/hu}* mice emitted a comparable number of vocalizations, having qualitative changes in some specific parameters both in pups and adults.
- Vocalization changes are specific to each developmental stage (or specific to social context).
- Heterozygous (*Nova1^{hu/wt}*) mice show intermediate values between *Nova1^{wt/wt}* and *Nova1^{hu/hu}* in both pup and adult USVs, suggesting that the effect of the I197V substitution in NOVA1 protein is dosage dependent.

These data are included in the revised manuscript (see Fig. 4, Supplementary Fig. 15 and main text for detailed description). To provide the reviewers and all readers access to the full vocalization data, we included all raw data and associated tables in supplementary materials (Supplementary Table 6-16).

Following the reviewer's suggestion, we next compared our methods to those of previous humanized *Foxp2* mice studies: Enard et al., 2009, Hammerschmidts et al., 2015. We also included one more study, von Merten et al., 2021 ("A humanized version of *Foxp2* affects ultrasonic vocalization in adult female and male mice") as this report also provided important findings /insights to vocalization analysis in humanized mouse models.

Together with the suggestion from Reviewer #1, we have prepared a table comparing the experimental methods, analytical methods, and findings of these studies.

[Comparison table for vocalization tests between humanized mouse models]

Comparison on USV tests between this study and three studies using humanized *Foxp2* mice (Enard et al., 2009, Hammerschmidts et al., 2015, von Merten et al., 2021). The table contains experimental conditions (methods and analysis) and findings in each study. To avoid changing the nuance of the words in each report, the terms in each paper were quoted verbatim in the table (e.g., calls, elements, vocalizations).

Study	Title (DOI)			
Enard et al., 2009	A Humanized Version of Foxp2 Affects Cortico-Basal Ganglia Circuits in Mice (https://doi.org/10.1016/j.cell.2009.03.041)			
Experiment				
Mouse	modification	Age	sex	
humanized FOXP2	2 amino acid substitution in mouse FOXP2	pup (P4, P7, P10, P13)	male and female	
Method	USV induction	recording time		
isolation-induced	isolate pup from their mother and nest	2 min (3 min for P13)		
Recording setting				
microphone	preamplifier, recording software, whistle tracking algorithm		Call count / structure analysis	extraction for acoustic parameters
UltraSoundGate CM16	UltraSoundGate 116, Avisoft SASLab Pro v4.33c		AVISOFT Recorder 2.97	custom software program LMA 2005
Representative parameter				
Number of calls, peak frequency (mean, start, end, max and min), location of the max frequency, greatest difference [Hz] in peak frequency, Slope of the call from start to the end peak frequency, slope of a linear trend through the peak frequencies of a call, modulation of calls (global/local)				
Syllable classification				
call type1: all calls that contained no or only minor frequency jumps , short call (<50ms); call type2: all calls that contained no or only minor frequency jumps, long call (>50ms); call type3: all calls with frequency jumps; call type4: remaining sounds (not analyzed)				
Representative findings				
No significant differences in the number of calls Foxp2hum/hum had a significantly lower start, mean, minimum and maximum peak frequencies in short 's' For short 's', the slope of the calls declined less in frequency in Foxp2hum/hum Calls with frequency jumps lasted longer, had longer gaps, and started and ended with higher peak frequencies in Foxp2hum/hum mice				

Study	Title (DOI)			
Hammerschmidt et al., 2015	A humanized version of Foxp2 does not affect ultrasonic vocalization in adult mice (10.1111/gbb.12237)			
Experiment				
Mouse	modification	Age	sex	
humanized FOXP2	2 amino acid substitution in mouse FOXP2	adult (9-16 wks)	male (with female)	
Method	USV induction	recording time		
courtship	place male mouse with female in the same box	120sec for habituation, 30sec pre-recording, 2min recording		
Recording setting				
microphone	preamplifier, recording software, whistle tracking algorithm		Call count / structure analysis	extraction for acoustic parameters
UltraSoundGate CM16	UltraSoundGate 116, Avisoft SASLab Pro v4.33/5.2		Avisoft Recorder 4.2	custom software program lma 2012
Representative parameter				
number of call elements, inter-element interval (ICI), element duration, amplitude gap, PF start, P _{fmax} , P _{fmax} location, P _{fjump} , element slope				
Syllable classification				
category1: elements<60ms; category2: elements >60ms; category3: elements >=200ms. Also classified 19 element type for finer grained level of analysis, using a visual classification.				
Representative findings				
No significant differences in the number of elements Males homozygous for humanized Foxp2 produced call elements with slightly more pronounced frequency jumps and a slightly earlier frequency maximum				

Study	Title (DOI)			
von Merten et al., 2021	A humanized version of Foxp2 affects ultrasonic vocalization in adult female and male mice (10.1111/gbb.12764)			
Experiment				
Mouse	modification	Age	sex	
humanized FOXP2	2 amino acid substitution in mouse FOXP2	adult (10-16wks)	male and female	
Method	USV induction	recording time		
courtship (natural condition)	Natural vocalizations of males and females without physical contact with each other (but with visual, olfactory, and acoustic contact)	12h, during night (19pm-7am)		
Recording setting				
microphone	preamplifier, recording software, whistle tracking algorithm		Call count / structure analysis	extraction for acoustic parameters
CM16/CMPA	Avisoft UltraSoundGate 416Hm, Avisoft USGH recorder		Selena (University of Tübingen)	custom-written MATLAB script (SVM, SH)
Representative parameter				
number of songs: number of syllables, duration, syllables per second syllable: average start, and minimum frequency, frequency bandwidth, duration, frequency slope, relative amplitude				
Syllable classification				
Three syllable groups: simple syllables (S), turn syllables (T) and jump syllables (J). Also classified 12 syllable types; depending on their frequency slope and existence of turning points and jumps				
Representative findings				
No significant differences in the number of calls Foxp2hum/hum mice showed a significantly larger variance in average number of songs Foxp2hum/hum mice emitted USV with significantly higher average and start frequencies, and larger frequency bandwidths (a consequence of the larger number of jump syllables) Foxp2hum/hum mice used more complex syllables containing frequency jumps and less simple or turn syllables Foxp2hum/hum mice used many transitions containing jumps (Female) Females and males contributed almost equally on number of songs and syllables during recording (Female) Females emitted USV syllables with smaller frequency bandwidths than males (Female) Females used less jump syllables than males, which fitting to the smaller average frequency bandwidth of female calls.				

Study	Title			
This study	A humanized NOVA1 splicing factor alters mouse vocal communications			
Experiment				
Mouse	modification	Age	sex	
humanized NOVA1 (Nova1hu/hu)	1 amino acid substitution in mouse NOVA1	pup (P7)	male and female	
humanized NOVA1 (Nova1hu/hu)	1 amino acid substitution in mouse NOVA1	adult (8-12wks)	male (with female)	
Method	USV induction	recording time		
(pup) isolation-induced	isolate pup from their mother and nest	5min		
(adult) courtship	place male mouse with female in the same box	15min for habituation, 5min recording		
Recording setting				
microphone	preamplifier, recording software, whistle tracking algorithm		Call count / structure analysis	extraction for acoustic parameters
UltraSoundGateCM16/CMPA	Ultrasound Gate USGH amplifier, AvisoftRecorderUSG		Avisoft Bioacoustics	Mouse Song Analyzer 2' (MSA2)
Representative parameter				
number of syllable, each duration, bandwidth, amplitude, purity, frequency variable, start- minimum- maximun end-frequency				
Syllable classification				
Four syllable type: simple syllable (s), one downward pitch jump (d), one upward pitch jump (u), multiple pitch jumps (M). Also classified with each duration or peak frequency (parameter with bimodal distribution)				
Representative findings				
pup	No significant differences in the number of syllables Nova1hu/hu produced lower amplitude for 's' syllable Nova1hu/hu had less syllables containing pitch jumps NOVA1hu/hu had increased proportion of complex syllables with high maximum frequency (Fq _{max})			
adult	No significant differences in the number of syllables Nova1hu/hu had lower start, mean and end peak frequencies (Fq) in long 's' syllable Nova1hu/hu had increased frequency variance (Fq _{variance}) in complex syllables with high Fq _{max} .			
Nova1hu/hu mice had comparable number of vocalizations, having qualitative changes in some specific parameters both in pups and adults. Vocal changes are specific to each developmental stage (or specific to social context). Heterozygous (Nova1hu/wt) mice show intermediate values between Nova1wt/wt and Nova1hu/hu in both pup and adult USVs.				

According to previous studies in the field, the following points are known to be affect the result of the analysis for mouse vocalizations, and hence the comparison of the four studies have been emphasized on this information.

- Age of the mouse (especially the neonatal period)
- Sex of the mouse (especially in the adult, this requires specific setup)
- Syllable classification

Pup-USV analysis (in comparison with the experiments of Enard et al. 2009)

Our group and Enard et al. both used isolation-induced USVs for pup-USV analysis. Our analysis used only 7-day-old pups, whereas Enard et al. used 4 to 13-day-old pups (analysis/ data by age is not presented). Neonatal mouse vocalizations are known to change daily (Grimsley et al., 2011), and analysis of combined vocalizations from different postnatal days may obscure transient changes at certain stages. Regarding the classification of syllables, Enard et al. divided them into three main categories according to the duration and the presence of jumps. We classified syllables according to the presence of jumps, their direction and number, and additionally, the distribution of duration and peak frequency (Arriaga et al., 2012). Another minor point is the length of the recording time (5 minutes in this study, 2 minutes (3 minutes for P13) in Enard et al.).

Adult-USV analysis (in comparison with the experiments of Hammerschmidts et al. 2015 and von Merten et al., 2021)

With respect to adult USV analysis, our method is based on Arriaga et al 2012 and Hammerschmidts et al 2015. A live female is placed in the same box to induce courtship vocalizations in the male mouse. Minor differences include the length of the recording time (5 minutes in Arriaga et al and this study, 2 minutes in Hammerschmidts et al.) and the syllable classification. Hammerschmidts et al. classified syllables into three main categories according to the duration, and further into 19 different types according to visual classification.

In contrast, von Merten et al.'s method differs significantly from ours. They recorded 12 hours of vocalizations in an environment that excluded physical contact (but did have visual, olfactory, auditory contact) in order to record more natural vocalizations for each sex. Their method can obtain vocalization data from each sex, although the number of vocalizations per time is far less than that of Hammerschmidts et al. and our method. von Merten et al. classified syllables into three main categories and further divided them into 12 types depending on frequency slope, turning points and jumps. Although male vocalizations are considered to be the main component under our experimental conditions, we cannot exclude the vocalizations of female in the same box, and thus we cannot rule out the possibility that the mixed vocalizations from female will obscure some vocal characteristics in male. It is also important to note that our study did not address changes in vocal characteristics in female mice.

In our study, we classified the syllables according to the presence and number of jumps and their directions, as well as subdividing them according to their characteristics. As a result, we observed precise differences in each parameter between genotypes. The classification method of syllables/ parameters varies among research teams and studies. We feel that the lack of classification buried characteristics, while over-classification makes it difficult to analyze. As described above, vocal characteristics are affected by age, sex, social context, and stimuli. Differences in the syllable/ parameter classification among the research teams may be attributed to the lack of clarification of the contextual significance of the mouse vocalizations. In the future, multi-dimensional analysis of each syllable as well as a unified interpretation of each vocalization linked to behavioral observation will be required in the mouse vocalization research, which will make the study more relevant for vocal communication research in other organisms, including humans.

We have added the comparison table as Supplementary Table 17 and included the discussion from these observation in humanized mouse models.

-Some information is missing from the methods, including:

*detailed information on the bioinformatics analysis, including software parameters for all multi-omics analysis;

*accession code for all sequenced multi-omics datasets;

* version of ExAC database and from all others used in the manuscript.

We apologize for the lack of detailed information and our mistake for the lack of accessibility to the genomic datasets. In the revised version, we have updated the Methods section, including description of the bioinformatics analysis, accession code for all sequence dataset (GEO numbers), as well as the version of the data set in the study.

-Figure 2B shows related gene expressions indicating they are similar. However, what is the statistical significance variation, and is there a detailed calculation on which reference gene was used to perform the related expression (this information is unclear)? Moreover, according to the figure, *NeuroD1*, an important neuronal marker, looks to present a significant alteration between *Nova1(hu/hu)* and *Nova1(wt/wt)*, which is contrary to what is claimed by the manuscript.

As the reviewers pointed out, the reason for the selection of these reference genes were not made clear. In the Figure 2B of the first manuscript, well-known cell type markers demonstrated no major differences in transcript expression levels in cell-type compositions between genotypes. For example, there was no significant difference between genotypes for *Neurod1* as shown below (average TPM: *Nova1^{wt/wt}* 11.591, *Nova1^{hu/hu}* 12.389, *p*-value=0.3091).

Gene_name	cell.type	exon_len	Nova1 (wt/wt) (tpm_average)	Nova1 (hu/hu) (tpm_average)	log2FC	P-Value	FDR
Sox1	Neural Peogonitor Cells	4037	8.604	8.817	0.038	0.71233139	1
Pax6	Neural Peogonitor Cells	3410	9.327	8.956	-0.053	0.62403181	1
Nes	Neural Peogonitor Cells	6401	5.518	5.231	-0.075	0.59155925	1
Smarca4	Neural Peogonitor Cells	6376	133.415	130.608	-0.027	0.59837066	1
Dcx	Immature Neurons	12460	8.451	8.871	0.073	0.2311984	1
Neurod1	Immature Neurons	2494	11.591	12.389	0.099	0.3091407	1
Tbr1	Immature Neurons	3976	62.355	62.649	0.010	0.86352399	1
Rbfox3	Mature Neuron	4707	245.216	238.816	-0.034	0.52829368	1
Mapt	Mature Neuron	5603	267.637	267.131	0.001	0.98625805	1
Dlg4	Mature Neuron	3357	1391.650	1358.836	-0.031	0.55017737	1
Gad1	Mature Neuron	7135	154.613	152.591	-0.016	0.7853679	1
Gad2	Mature Neuron	6872	84.267	83.999	-0.001	0.9881287	1
Slc17a6	Mature Neuron	4320	18.143	17.283	-0.067	0.43630165	1
Slc17a7	Mature Neuron	3199	1803.030	1730.526	-0.055	0.3063122	1
Th	Mature Neuron	1757	14.917	13.004	-0.192	0.31960173	1
Chat	Mature Neuron	4623	3.076	3.191	0.057	0.68348733	1
Gap43	Mature Neuron	1301	443.531	436.771	-0.019	0.73440878	1
Nrp1	Mature Neuron	10066	16.870	17.797	0.080	0.29494894	1
Gfap	Astrocytes	4072	144.802	153.931	0.092	0.19419476	1
S100b	Astrocytes	1661	150.066	160.306	0.099	0.10421263	1
Aldh11	Astrocytes	3084	120.726	125.149	0.056	0.28441995	1
Olig1	Oligodendrocytes	2171	129.355	141.414	0.132	0.14319623	1
Olig2	Oligodendrocytes	2437	26.208	28.231	0.111	0.18981804	1
Mbp	Oligodendrocytes	6562	1582.106	1632.931	0.049	0.44854193	1
Mog	Oligodendrocytes	1707	290.155	303.311	0.067	0.28706339	1
Cx3cr1	Microglia	3729	37.414	39.995	0.100	0.10316336	1
Itgam	Microglia	6257	19.128	20.228	0.084	0.2290415	1

Comparison of the RNA expression of representative cell-type marker genes (listed in Figure 2B in the original manuscript) between *NOVA1^{hu/hu}* and *Nova1^{wt/wt}* of P21 cortex. The list includes the average TPM (transcripts per million) value for each genotype. Statistics are performed using edgeR. *Nova1^{wt/wt}* N=4, *Nova1^{hu/hu}* N=4.

We had presented these genes as ‘representative’ markers for each cell type, but following to the reviewers’ comments, we decided to present a more global gene expression comparison in *Nova1^{hu/hu}* mice. We have now provided a comparison of global gene expression: correlation plots between genotypes (Figure 2c, Supplementary Figure 5d) instead of showing a subset of marker genes. These comprehensive analyses reveals that the transcript levels, including above genes in the list and *Nova1* itself, were nearly identical between genotypes throughout development (Fig. 2c), as well as in different brain regions (Supplementary Fig 5a-d).

-according to the authors, Figure 2C indicates that *Nova1* is most highly expressed in the mouse midbrain. However, additional protein quantification is required to confirm this observation.

In response to the reviewer's comments, we conducted the following additional analysis regarding NOVA1 protein expression in mouse brain tissue.

- Western blotting with anti-NOVA1 and anti-NOVA antibodies in extracts of dissected adult mouse brain regions. Signals were quantified over ACTB.
- Immunofluorescence images of 3-week-old and postnatal day 0 mouse brains stained with anti-NOVA1 antibody.

NOVA1 expression in adult mouse brain (western blot)

(top) Western blotting for NOVA proteins in dissected brain regions of adult mouse brain. Proteins and isoforms corresponding to each band are listed on the right. The highly expressed NOVA1 in the hypothalamus, substantia nigra and periaqueductal gray is exon4 minus isoform (Dredge et al., 2005). (bottom) Quantification of NOVA1 bands was done using ImageJ and normalized by signal to ACTB. Data represent the mean \pm SD of three replicates. The cortex value set as 1. CTX: cortex, HIP: hippocampus, CP: Caudate putamen, TH: thalamus, HY: hypothalamus, SN: substantia nigra, PAG: periaqueductal gray, CB: cerebellum.

NOVA1 expression in mouse brain (immunostaining)

Postnatal day 21, Sagittal section

scale bar: 500µm

Postnatal day 21, Coronal section

Postnatal day 0, Sagittal section

Postnatal day 0, Coronal section

scale bar: 500µm

Immunostaining for NOVA1 (green) and DAPI (blue) in P21 and P0 mouse brain (sagittal section and coronal section). The scale bar indicates 500µm. The corresponding brain regions are indicated in the orange characters. CTX: cortex, HIP: hippocampus, CP: Caudate putamen, TH: thalamus, HY: hypothalamus, MB: midbrain, CB: cerebellum.

These data shows that the highest levels of NOVA1 is in the mouse midbrain, with a particularly high level in the periaqueductal gray (PAG), a region well known to contain neurons that control vocal behavior. We have included these data in Figure 3a-b, Supplementary Figure 10-11, and updated the main text.

-Figure 2d shows the expression variation between *Nova1*(ko/ko) and *Nova1*(wt/wt). A table listing all genes, including expression, fold change, and statistical significance would be desirable. Moreover, please include a clear description of the x/y axis in the figure's legend.

Following the reviewer's suggestion, a list of variable transcripts (gene name, each TPM value, fold change, statistical significance) between *Nova1*^{ko/ko} and *Nova1*^{wt/wt} is included in Supplemental Table 18. The X/Y axis (each average TPM (log₂)) has also been clarified in all plots as follows.

This Figure 2D in the first manuscript is now moved to Supplemental Figure 5f in the revised manuscript.

-non-significant (p-value<0.05) gene expression alterations could be removed from Figure 2, and it should not be mentioned as altered in the manuscript since there's no statistical support.

We agree with the reviewer's suggestions, and removed those plots from the Figure.

In this revision, we have revised gene expression analysis with using edgeR for statistical analysis and applied a more stringent thresholds considering the read counts and biological replicates. Although the initial manuscript indicated that a handful of genes may be differentially expressed in humanized mice, most of these genes are expressed at low levels (low read counts), and thus were excluded by the stringent thresholds.

We have now presented the correlation plots showing all genes expressed in the target tissue/region to allow for comprehensive gene expression comparisons (Figure 2c, Supplementary Figure 5a-d). From these gene expression analyses in *NOVA1^{hu/hu}* mice showed that most genes maintain expression levels comparable to those of wild-type mice. And hence we conclude that 'Humanized NOVA1 mice are comparable to wild-type mice in gene expression in the brain'. We have updated the main text in the revised manuscript as below.

Result

"Comprehensive gene expression analysis of the midbrain at embryonic day 18.5 (E18.5) and at postnatal day 21 (P21), when fundamental neural circuits and behaviors have been established, revealed that the transcript levels, including of *Nova1* itself, were nearly identical between genotypes throughout development (Fig. 2c), as well as in different brain regions (Supplementary Fig 5a-d)."

-Figure 3C is explained as "Among the transcripts with significant AS changes, 47 of them had significantly different NOVA1 binding peaks on their transcripts between NOVA(hu/hu) and Nova1(wt/wt) (Figure 3C)." However, RNA-seq and CLIP data show similar and comparable read coverage and peaks, respectively. The only observed variation is between the number of reads, but the values do not look to be normalized for a correct comparison between them. I suggest to clarify this description.

We thank the reviewer for the suggestion. The RNA sequencing and CLIP data are separate datasets, and as the reviewer states, no correction was made between these different datasets. We have updated the description in the revised manuscript to clarify the points as follows.

Results

"AS analysis in the P21 midbrain revealed that several AS events were specifically altered in *Nova1^{hu/hu}* mice compared to *Nova1^{wt/wt}* controls (Fig. 3c). Specifically, 720 events showed significant changes (delta PSI (*dI*) more than 5% ($|dI| > 0.05$, $p < 0.05$)) (Supplementary Table 4).

... We further investigated the NOVA1 I197V variant's influence on AS by cross-referencing the CLIP and gene annotation datasets and assessing the statistical significance of these results using random resampling methods, as detailed in the section below.

To assess the potential direct effect of the I197V substitution, we examined NOVA1 binding on these transcripts using the CLIP dataset. Among the 720 differential AS events, 258 (41%) had NOVA1 binding peaks on their transcripts (Figure 3c)."

-The gene ontology analysis did not correctly describe the origin of the genes considered in each pathway (Figure 3D). Did the authors use a background dataset from a normal brain, or was the analysis based on the background provided by the software?

We apologize for the lack of clarity in the description; Figure 3D in the first manuscript is not a gene ontology analysis using the background dataset, but shows the percentage of genes with differential alternative splicing (AS) events that belong to each behavioral category. Please refer to Supplemental Table 5 for the attribution and classification of these genes in the analysis.

Driven by the reviewer's comment, we conducted gene ontology analysis (not limited to behavioral categories) using a background dataset of genes expressed in the midbrain of P21 mice to see the overall effects on AS in *Nova1^{hu/hu}* mice. The result shows that the 630 transcripts with

differential AS events are enriched in processes involved in cell projection organization or chromatin remodeling, and those transcripts with NOVA1 binding peaks are enriched in processes related to cell projection, morphogenesis and synaptic function. Such enrichments have been found in prior studies for genes with specialized expression in speech brain regions (Pfenning et al 2014; Gedman et al 2022).

The result is included in Figure 3e and the description is added in the revised manuscript as below.

Result

“Gene annotation analysis revealed that the 630 transcripts with differential AS events are enriched in processes related to cell projection organization or chromatin remodeling, and the transcripts with NOVA1 binding peaks are further enriched in processes involving cell projection, morphogenesis, and synaptic function (Fig. 3e).”

-How did the authors confirm that the CRISPR experiment did not generate any relevant off-target effects? How many founders were used in this work?

Humanized NOVA1 mice lines were generated using CRISPR/Cas9, resulting in several lines carrying the designed allele (ref. Methods section-- Generation of *Nova1^{hu/hu}* mice). After characterizing the physical features (weight, growth, and reproduction) for the lines with the desired knock-in allele, all of which showed nearly identical parameters, we did an in-depth analysis of line #01. We performed multiple (more than 8 generations) backcrosses to C57BL/6J strain before collecting the data, to make sure that potential off targets were segregated.

To confirm this, we performed genome sequencing on the possible off-target sites of the gRNA used (predicted by CRISPR direct: <https://crispr.dbcls.jp/>) to compare sequences of control (wild type) and humanized NOVA1 mouse. There were 10 potential off target loci with mismatches outside of the PAM+12mer core sequences. None of them were on chromosome 12 (carrying NOVA1), the target site of this study and could be segregated even if they received off-target edits.

gRNA for generation of humanized NOVA1 knock-in mice
 # [CRISPRdirect]
 Genome
 Mouse (Mus musculus) genome, GRCm39/mm39 (Jun, 2020)

gRNA sequence	pam sequence	hit_20mer	hit_12mer
TGCTACTGTGAAGGCTATAATGG	NGG	1	11

Target Site: 20mer+PAM

chr.	strand	start	end	sbjct	match	Name	Gene	region
chr12	-	46747681	46747703	CCATTATAGCCTTCA	23	Target Site	NOVA1	CDS

Potential Off Target Sites: 12mer+PAM

chr.	strand	start	end	sbjct	match	Name	Gene	region
chr2	+	141303564	141303578	TGAAGGCTATAACGG	15	OT1	MacroD2	intron
chr5	+	71203878	71203892	TGAAGGCTATAATGG	15	OT2	Gabra2	intron
chr6	+	14018751	14018765	TGAAGGCTATAAAGG	15	OT3	AK143417	intron
chr7	+	127986246	127986260	TGAAGGCTATAAAGG	15	OT4	Rgs10	intron
chr8	+	6037054	6037068	TGAAGGCTATAAAGG	15	OT5	upstream of Gm44843	deep intergenic
chr9	+	34818427	34818441	TGAAGGCTATAAAGG	15	OT6	Kirrel3	intron
chr17	+	7424358	7424372	TGAAGGCTATAAAGG	15	OT7	Rps6ka2	intron
chr1	-	58770592	58770606	CCTTTATAGCCTTCA	15	OT8	Cflar	intron
chr3	-	143183496	143183510	CCTTTATAGCCTTCA	15	OT9	downstream of 1700001N15Rik	deep intergenic
chr12	-	46747681	46747695	CCATTATAGCCTTCA	15	Target Site	NOVA1	CDS
chr18	-	40219960	40219974	CCTTTATAGCCTTCA	15	OT11	Gm50395	intron

The gRNA sequence and predicted off-target site information predicted by CRISPR direct (<https://crispr.dbcls.jp/>). There are 10 potential off target loci with mismatches outside of the PAM+12mer core sequences.

Potential Off Target (POT) Genomic Sequences
 10 locations: POT1 - 10 (+ target NOVA1 site)
 Subject +/- 50bp (115bp)

Alignment

- ref: reference genome (Mouse GRCm39/mm39)
- wt: NOVA1^{wt/wt} mouse
- hu: NOVA1^{hu/hu} mouse

* identical

- subject (12mer+PAM)
- target site (20mer+PAM)

The genomic sequencing of the potential off target (POT) loci. Alignment of each genotype and reference genome for the genomic sequence of 100 bases around the POTs are shown. Asterisks indicate identical nucleotides. All POT sites were identical between genotypes and the reference genome, with the target site (responsible for I197V substitution) being the only detectable edits.

The genomic PCR/ sequencing results showed that for all potential off-target sites, the sequence in the humanized NOVA1 mice were identical to control mice and the reference genome. These results indicate that off-target effects in humanized NOVA1 mice are undetectable (at least for the potential off-target sites).

This analysis is included in the Supplementary Figure 4d-e, and the description in the Methods 'Generation of *Nova*^{hu/hu} mice' section is updated.

Reviewer comments and Response (Response are in blue)

Reviewer #3 (Remarks to the Author)

We deeply appreciate this reviewer and their colleague for taking time to review our manuscript carefully and for suggestions and comments. We have revised our manuscript and made updates according to the suggestions. We hope that the reviewers' concerns and questions have been addressed in this revision.

Reviewer comments and Response (Response are in blue)

Reviewer #4 (Remarks to the Author)

In this manuscript, Tajima and co-workers look to address a fascinating question about the molecular and genetic underpinnings that accompanied human evolution, in particular surrounding vocal learning. The work described in this manuscript builds on the observation of a highly conserved, 'human-specific' mutation within the splicing factor NOVA1. To better understand the role of this mutation, authors generate a mouse containing a 'humanized' version of NOVA1 and examined the impact of this mutation on several different molecular and phenotypic readouts. The bulk of the experimental data included in the manuscript are derived from RNAseq analysis, CHIPseq analysis, and analysis of vocal patterning. While I find the problem and the approach quite interesting and compelling, I am not convinced that the data presented support a strong conclusion about any role for this particular mutation in the pathway of interest. Certainly there are hints about the potential for a role, but for reasons expanded upon below it doesn't strike me that there is strong evidence in support and so I don't see this as a good candidate for publication in a journal with broad readership like Nature Communications.

We appreciate the reviewer's careful review of our initial manuscript and for the suggestions and comments. The comments were instructive for us and helped improve the quality of our manuscript, and highlighted future challenges for our research.

Regarding the RNAseq dataset, I have several questions/concerns about methodology surrounding data analysis, and more broadly about the conclusions drawn from this. From a methodological perspective, it seems that data shown in figure 2 panels B, D, E, F, and G are all derived from analysis of RNAseq done on mice with three different genotypes (wt, ko, and 'humanized' NOVA1). Although the manuscript notes that the data from these experiments are available via GEO, no accession number is provided and no such dataset appears available that is linked to the two corresponding authors (specific to this study).

We deeply apologize for our mistake about the lack of access to the transcriptome data. The GEO access numbers and a token associated with this manuscript are as follows:

GEO access number: GSE253298 (SuperSeries)

comprising subseries GSE253296 and GSE253297

Secure token: ypanqaaufnufjyn

Please note that the NOVA1-KO data is from an earlier report (Saito et al., 2016) and has been re-analyzed in this manuscript. For the KO data, please refer to the following paper and GEO number as follows:

GEO access number: GSE69711

Reference: Saito et al., 2016, NOVA2-mediated RNA regulation is required for axonal pathfinding during development <https://www.ncbi.nlm.nih.gov/pmc/articles/PMC4930328/>

This information is included in the Data Availability and Methods section.

Although it isn't entirely clear from the text, it appears that figure 2B is derived from one RNAseq study (P21 cerebral cortex), while figures 2D, E, F, and G are derived from a separate RNAseq study (e18.5 midbrain). For the former, authors have selected (how?) a handful of cell-type specific markers and have represented the data from the sequencing data using box plots and (apparently?) statistical analysis using

a Wilcoxin test to compare their differences (based upon the figure legend). From a statistical perspective, I would argue that it is inappropriate to extract RNAseq data in this way and use individual tests on the underlying counts. Authors describe using edgeR for analysis of differential expression, but it isn't clear how or why they changed the approach in this figure. Certainly there is wide agreement in the field that the statistical approaches used by a program like edgeR are superior (in spite of their limitations) to the approach that authors appear to have taken for these extracted genes.

We apologize for the unclear description of the dataset in the text. We directly address the reviewer's concern by undertaking an unbiased global analysis with edgeR.

The reviewer is correct that the original figure 2B was RNA sequencing data from P21 cortex and figures 2D, E, F, and G are from E18.5 midbrain in the original manuscript. For cell type markers (figure 2B in the first manuscript), we chose them based on well-known cell-specific expression patterns with an intent to show that the cell-type compositions are unaffected.

However, we agree with the reviewer that it is more convincing to describe the global view of gene expression patterns to demonstrate that there is little or no changes in specific cell-type compositions. Following to the reviewer's suggestions, instead of showing limited number of genes with box plot graphs, we now provide a global view of transcriptome in the midbrain at both E18.5 and P21.

Gene expression correlations between $Nova1^{wt/wt}$ and $Nova1^{hu/hu}$ in midbrain at E18.5 and P21. Scatterplots of gene expressions measured in average TPMs are shown. The axes are shown in \log_2 scales. The red dot indicates differentially expressed gene between genotype (p -value <0.05 , FDR <0.1). The yellow dot indicates *Nova1* gene. Pearson correlation is reported on top of the plots. E18.5 midbrain samples $Nova1^{wt/wt}$ N=6, $Nova1^{hu/hu}$ N=6, P21 midbrain samples $Nova1^{wt/wt}$ N=4, $Nova1^{hu/hu}$ N=4.

We also included the data from the cortex and cerebellum at P21 to compare global gene expression in $Nova1^{hu/hu}$ mice. The data is analyzed as correlation plots between genotypes in corresponding brain regions and ages, as well as global matrix and PCA plots between samples analyzed.

Comprehensive gene expression analysis in the brain. **(a)** Global correlation matrix of gene expression levels between brain samples: midbrain at e18.5, cortex, midbrain, cerebellum at P21 in *Nova1^{hu/hu}* and *Nova1^{wt/wt}* mice. Heatmap showing correlation coefficients for log₂ (TPM+1), color intensity and the size of the circle are proportional to the correlation coefficients. **(b)** Principal component analysis of gene expression levels between samples. The X axis is the first principal component, and the Y axis is the second principal component, with a percentage of variances explained by each component approximately 53% and 29%, respectively. The ellipses indicate confidence ellipses around group mean points (large dot). **(c)** Principal component analysis of gene expression levels in each corresponding sample (age and brain region). **(d)** Gene expression correlations between *Nova1^{wt/wt}* and *Nova1^{hu/hu}* in corresponding brain regions and age. Scatterplots of gene expressions measured in average TPMs are shown. The axes are shown in log₂ scales. The red dot indicates a differentially expressed gene between genotypes (FDR<0.05). The yellow dot indicates *Nova1* gene. Pearson correlation is reported on top of the plots. The upper two plots (midbrain) are identical with the plots in the previous page. The midbrain sample of e18.5 *Nova1^{hu/hu}* N=6, *Nova1^{wt/wt}* N=6. The cortex, midbrain and cerebellum samples of P21, *Nova1^{hu/hu}* N=4, *Nova1^{wt/wt}* N=4, respectively.

These re-analyses are included in Figure 2c and Supplementary Figure 5a-d. These changes help readers see the overall picture of gene expression. We appreciate the input from the reviewer.

This technical point aside, it is surprising/disappointing that authors have provided no further insights into any of the other information derived from this initial RNAseq experiment. Are there other changes observed? What do those changes suggest? How do they compare with the other RNAseq experiments described in the figure (from a different developmental time point)? Currently, it reads as if authors are only presenting the data that support a single conclusion and but they don't provide access for the readers to contemplate alternative explanations/interpretations.

We appreciate the concerns raised by the reviewer. As mentioned above, we have made changes in the gene expression analysis so that the entire picture can be seen, not just a limited subset of genes. The developmental time points and each brain region (E18.5 and P21, and each brain region for P21) are additionally presented (Figure 2c, Supplementary Figure 5a-d). From these results, the following insights are obtained.

In the midbrain, where NOVA1 is most highly expressed (Figure 3a-b, Supplementary Figure 10-11), there is little overall gene expression variation between genotypes (*Nova1*^{wt/wt} vs. *Nova1*^{hu/hu}) (Figure 2c). This is true across developmental stages (both in E18.5 and P21). The only gene that showed a notable difference in P21 midbrain was *Gkn3*, a secreted protein involved in endothelial cell proliferation. The NOVA1 protein is also expressed in the cortex and cerebellum at P21. In these regions, the overall gene expression is maintained between genotypes (Supplementary Figure 5c-d). Our results indicate that the I197V substitution in the NOVA1 protein does not significantly affect overall transcript expression levels, at least based on RNA sequencing analysis in the dissected brain regions. From these analyses and results, the text is revised as follows.

Result

“Comprehensive gene expression analysis of the midbrain at embryonic day 18.5 (E18.5) and at postnatal day 21 (P21), when fundamental neural circuits and behaviors have been established, revealed that the transcript levels, including of *Nova1* itself, were nearly identical between genotypes throughout development (Fig. 2c), as well as in different brain regions (Supplementary Fig 5a-d).

... The only gene that showed a significant steady-state difference in *Nova1*^{hu/hu} mice was *Gkn3* at P21, a secreted protein involved in endothelial cell proliferation (Figure 2c; (p -value<0.05, FDR<0.1)). *Gkn3* is thought to be involved in adaptive gene loss during recent human evolution⁵⁴, and showed down regulation in the P21 midbrain of *Nova1*^{hu/hu} mice (average TPM 18.6 in *Nova1*^{wt/wt}, 11.3 in *Nova1*^{hu/hu}, $\log_2FC = -0.71$, p -value = 2.3×10^{-6} , FDR = 0.037).”

Regarding panels D, E, F and G of figure 2, the approach to analysis here seems flawed intellectually rather than statistically. If I understand authors' approach, they have used edgeR to identify a subset of genes with differential expression in the ko strain (vs. wt), and then asked about the behavior of that subset of transcripts in the background of the humanized strain. But it seems clear from the lower panel of 2D that many of these transcripts are not affected by the V197I variant and so their inclusion in this analysis seems irrelevant (certainly one could imagine other alleles of NOVA which altered its activity in ways that would be totally unrelated to the V197I variant, why should the ko be considered differently?). Much more important would be to see a robust statistical analysis of all of the transcripts whose expression is differentially impacted in a comparison of the V197I variant to the wt strain. Perhaps a reciprocal analysis could be interesting (that is, highlight in 2D the edgeR identified transcripts from the hu vs. wt experiment and then show those in the context of the ko), but the current approach does not make sense for trying to understand the role of the V197I variant.

We agree with the reviewer's point that we should highlight the comparison between hu vs. wt, and have revised the gene expression analyses as described above. We have included these data in Figures and accordingly moved the comparison with KO data to Supplementary Figure. On the other hand, we believe it make sense to show that the effect of this substitution is distinct

from the loss of function of NOVA1 for readers. Below we describe the nature of the 197th amino acid in the NOVA1 protein and its relevance to the previously known loss of function of RNA-binding proteins.

As shown in Figure 1a, the protein sequence of NOVA1 is highly conserved across species, especially in the KH domains, which are responsible for RNA-binding capability. The KH domain found in many RNA-binding proteins contains common motifs (invariant Gly-X-X-Gly motif, hydrophobic core, variable loop), and these properties are widely shared with many other RNA binding proteins. The 197th amino acid of NOVA1 is one of the amino acids that constitute the hydrophobic core. There are several examples where one amino acid substitution constituting the hydrophobic core in the KH domain cause loss of function of the protein. One is an amino acid substitution (I304N) in the hydrophobic core of the KH2 domain of the FMR1 protein. That patient presented with severe fragile X mental retardation (Boulle et al., 1993). *In vitro* substitution of this corresponding amino acid leads to diminished RNA binding of both FMR1 and hnRNPk (Siomi et al., 1994). One amino acid substitution (Ala->Thr) in the hydrophobic core comprising the KH domain of GLD-1 in *C. elegans* shows a recessive phenotype (arrested germ cells differentiation) presumably due to structural changes in the KH domain (Jones et al., 1995). Also, in NOVA1, *in vitro* binding assays have shown that single amino acid substitutions (Ile->Thr or Leu -> Asn) that form the hydrophobic core of the KH domain cause loss of RNA binding ability (Buckanovich et al., 1996, Buckanovich et al., 1997).

REDACTED

KH domain sequence alignment. Figure adapted from Lewis et al., 2000 with modifications. Each RNA binding protein and comprising KH domain number are listed on the left. Secondary structural elements were based on the X-ray structure. Color coding scheme: yellow, invariant GXXG motif; purple, hydrophobic core (aliphatic α/β platform). Functional classifications: A, aliphatic stacking interaction; S, side chain-base hydrogen bond, including water-mediated contacts; M, protein backbone-base hydrogen bond; *, van der Waals contact. Amino acids on the red background are those for which loss of protein function was reported due to the substitution. Amino acids on the green background indicate the 197th valine of NOVA1, which is unique to modern humans.

In light of these studies that showed substitutions in amino acids within the hydrophobic core could lead to loss of function by reducing the RNA-binding capacity of the protein, one could imagine that 197th amino acid substitution would have negative effect related to loss of function of NOVA1. Our results ruled out this possibility, demonstrating that the I197V substitution in NOVA1 does not simply cause loss of function in the NOVA1 protein. We have included this description from previous studies on RBPs in the main text, to provide the readers understanding of the nature and specificity of this substitution.

Regarding the CLIPseq, I would again note that it was difficult to easily evaluate the raw data from authors' experiments for a number of reasons including trivial ones (the supplemental tables are mis-numbered between the text and the supplemental tables) to meaningful ones (only author-selected data are available).

We apologize again for the confusion and for the inability to access the data. We have taken great care to prevent mistakes in the revised version. As mentioned above, we provide GEO accession numbers/ token for reviewer accessibility and revised the data so that the readers can look at the effects on a global range of transcripts.

Nevertheless, my main concern with these data are that the apparent changes in CLIP between wt and V197I are modest in impact, modest in statistical significance, and of unclear relationship to any of the changes previously described from the RNAseq experiment. In the text, authors describe 215 differential peaks between wt and hu, but then say that 78 were higher in human and 67 were higher in V197I ... what does that mean about the other 70 peaks? In looking at Supplemental Table 2 (not 3 as described in the text), it appears that authors have 'trimmed' the list of peaks included on the "ST2.1_diff.peaks" tab to include only those with a log₂FC value above a certain threshold (apparently 1.47?) on the subsequent tabs, presumably resulting in the 78 and 67 peaks noted above. But it is unclear how this threshold was selected.

The description regarding the threshold was documented in the original version of the manuscript: each table legend. For each genotype, the peaks that are marked as 'greater' are due to the additional threshold value for a sufficiently low peak height (peak height less than 10) for the other genotype, and not the log₂FC threshold. This was done to extract peaks that were lost or gained due to I197V substitution.

Following concerns from the reviewer, we also revised our CLIP analysis. Please refer the explanation in the next response.

Moreover, if authors don't have confidence in those with lower FC then why were they called significant in the first place? Likewise, if there is a threshold for FC in considering significance, shouldn't there also be for count depth? I note that many of the called peaks are covered by an exceedingly small number of reads (again, it is difficult to compare these to the remaining ~29,000 peaks because authors have only provided a small amount of the underlying data.).

We agree with the reviewer's point. Most of the peaks we presented in our initial manuscript as differential peaks have low counts. We originally focused on these changes so that we can capture any differences by the single amino acid substitution, but was somewhat misleading to the readers that these are critical differences.

Following the reviewer's concerns and suggestion, we revised our CLIP analysis as follows.

- Consider the expression levels of each transcript (low expression transcripts were removed by referring RNA seq data).
- Consider biological replicates (CLIP tag per peak should be detected in all replicates).
- Consider count depth (tags per peak greater than 10).
- Use edgeR to compare read counts per peak between genotypes.

The revised analysis shows that CLIP peaks are remarkably similar between *Nova1^{wt/wt}* and *Nova1^{hu/hu}*. Differences in these stringently called peaks remain minimal, with only low-count changes detected (a total of 250 peaks: average CLIP tags/ peak 16.3 in *Nova1^{hu/hu}*, 13.1 in *Nova1^{wt/wt}*, p -value <0.01 , $|\log_2FC|>1$) (Figure 2g, Supplementary Table 3). While we initially focused on and described the minor differences in CLIP peaks, the comparable number of tags in the majority of peaks between genotypes supports the findings from other analyses, including enriched binding motifs (Figure 2f), genomic distribution (Figure 2e), and biochemical *in vitro* assays (Figure 2i-j). These results emphasize that the RNA-binding capacity and sequence specificity of NOVA1 are not affected by the I197V substitution.

NOVA1-CLIP analysis and *in vitro* binding assay. (e) Distribution of NOVA1 CLIP peaks on the genome. (f) The most enriched binding sequence from NOVA1 CLIP peak (upper part) and frequency of that sequence (UCAU) present around the binding site (lower part). (g) Scatterplot of CLIP tag number per peak between *Nova1^{wt/wt}* and *Nova1^{hu/hu}*. The axes are shown in log₂ scales. R square value is shown. The P21 midbrain samples, *Nova1^{hu/hu}* N=3, *Nova1^{wt/wt}* N=3, respectively. (i) Representative data from gel shift assay using each purified NOVA1 protein and ³²P-labeled UCAU RNA oligo probe. Leftmost lane is probe only, no purified protein. The bottom band represents the free probe, and the top shifted band represents the purified protein bound to the RNA probe. (j) The bands were quantified from gel images of the gel shift assay and the amount of binding per protein concentration were plotted. The dissociation constants (K_d values) for each purified NOVA1 protein are shown in the graph. N=5 for each point, respectively.

In this revision, we also included the CLIP data from P21 cortex and cerebellum samples, showing that these conclusions are common to all three brain regions (Supplemental Figure 8). From these results, we concluded that “Modern human specific amino acid substitution does not affect sequence specific RNA-binding capacity of NOVA1”. We have updated the main text with these results.

On the other hand, this CLIP results does not negate the entire effects of the I197V substitution, since post-transcriptional regulation of mRNAs by RBPs (more specifically NOVA1) often requires not only their own binding itself, but also the influence of other factors acting in cis (e.g. miRNAs

or cofactor) or trans (e.g. signals or stimuli/ modifications) and is spatiotemporally regulated on a specific template (e.g., developmental stage, tissue, cell type, stimulus) (Saito et al., 2016, Gill et al., 2017, Saito et al., 2019, Tajima et al., 2023). Indeed, we detected a solid phenotype in mouse vocalizations. Since the phenotype of the vocalizations was observed to be context/ condition-dependent (see explanation in the last response), this needs to be considered in detail in the future with respect to the construction of a new experimental setup for the underlying molecular mechanisms. Nonetheless, the NOVA1-CLIP in this study provides information of physical properties of NOVA1 in each brain region, for elucidating the potential molecular mechanisms that NOVA1 is involved in (Figure 2h, Supplementary Fig. 8e, j, Fig. 3e).

Importantly, there appear to be several inconsistencies between the data shown in figure 3 and the data included in the supplemental tables. Figure 3C includes three transcripts which authors describe as having statistically significant changes in AS and having differential CLIP peaks. But two of these transcripts, Gria2 and Ube2q2, don't appear in the supplemental data files that authors include. What are the data that are included for this figure representing? For me, this is the weakest portion of the paper. To really make the argument that the V197I variant is important there needs to be some molecular defect that can be plausibly traced to the mutation.

We apologize for the inconsistencies between the Figure and the Supplemental table. We found out that this was due to the different threshold applied when creating the Figure and the table for Differential CLIP Peaks ($p < 0.05$ was adopted when creating the Fig, while $p < 0.01$ was adopted when creating the Table). As explained above, we have now revised/ updated our CLIP analysis with the revised threshold and replaced Figure 3C and Table in the original manuscript.

For the molecular effect by the I197V substitution, we have found in several splicing changes (Figure 3c, d), and unique enrichment on cell projection/ morphogenesis and behavior/ vocalization term (Figure 3e, f). Based on the suggestion from Reviewer #1, we additionally conducted random resampling tests to verify significance of these observations. Specifically, we conducted 1,000 times random resampling of 630 transcripts (number of transcripts with differential AS event) detected in our RNAseq dataset to determine how many are annotated under the "behavior" category in gene ontology.

The resampling analysis. The 650 random resampling were repeated 1000 times to calculate the number of transcripts those annotated in the Behavior category in Gene Ontology. The histogram shows the density of the number of transcripts detected for each resampling, the boxplots at the top show the distribution features (median and quartiles (box) and maximum minimum (whiskers) and outliers (dots)). The green triangle indicates mean value of the resampling. The red triangle indicates 27: number transcripts detected in this study.

The results showed that the number of 27 transcripts (detected number in our study) was a probability of 9.9%, higher than the average number of transcripts detected by chance.

We conducted the same analysis for vocalization related transcripts. In our study, four of 27 transcripts are associated with the vocalization behavior category. We performed 1,000 times

random resampling of 27 transcripts from the 843 transcripts annotated under the behavior category in the gene ontology database and calculated how many were annotated in the vocalization category.

Number of Resamplings	Mean	Variance	SD	Median	First quartile	Third quartile
1000	0.7	0.6	0.8	1.0	0.0	1.0

Observed		count	rank
Minimum	Maximum		
0.0	2.0	4.0	< 4 %

The resampling analysis. The 27 random resampling were repeated 1000 times to calculate the number of transcripts those annotated in the vocalization category. The data presentation is the same as above figure. The green triangle indicates mean value of the resampling. The red triangle indicates 4: number transcripts detected in this study.

The probability of detecting the four vocalization-related transcripts is less than 4%, indicating the number of vocalization-related transcripts detected in this study is significantly higher than what would be expected by chance.

We further updated the analyses on NOVA1 expression (Figure 3a-b, Supplementary Figure 10-11), and the specific enrichments of NOVA1-binding targets in the brain (Figure 2h, Supplementary Figure 8e, 8j). The CLIP and detailed NOVA1 expression analysis in the brain suggest a link between NOVA1 and vocal behavior. These are noteworthy as possible molecular mechanisms, as they underlie a clear output of actual vocal changes. On the other hand, as the reviewer points out, our study provides the association, but has yet to prove the exact molecular mechanism. Considering the fact that the function of RNA binding proteins (RBPs) is influenced by regulatory factors beyond itself, and that the phenotype of vocalization depends on environmental factors (age, stimuli, social context), further insights of vocal pathway and experimental systems will be essential to elucidate the molecular mechanisms linking the observation.

Figure 3B is quite uninteresting to me: it shows many apparent AS events which seem to be unrelated to any detectable change in NOVA binding. Are these simply noise in the AS calling? If not, how to understand them if NOVA binding (CLIP) isn't impacted? If the V1971 is altering something other than binding (CLIP), then are the CLIP differences all noise? Critically, I am again worried about cherry-picking of data given the apparent inconsistency between what is shown in the figure and what is presented in the supplemental table.

Figure 3B in the original manuscript showed examples of differential AS events and nearby NOVA1-CLIP binding. The effects on AS events are not necessarily explained by changes in the number of bindings of the RNA binding protein itself. We are statistically and objectively detecting AS events, and these are not cherry picking. This entire dataset was used to analyze their significance through gene ontology and enrichment analysis. The results showed specific biologic significance, indicating this observation are not noise. As for the inconsistency between the figure and table, we have recognized the error and corrected them in line with the revised analysis. Regarding to the question 'how to understand them if NOVA binding (CLIP) isn't impacted', we discuss this in light of findings from previous RBP studies below.

The function of RNA binding proteins on mRNA processing is often influenced by various factors beyond its own RNA binding ability; several studies indicate that competing or cooperating

proteins, or other molecules (such as metals or ATP), determines the downstream effects on target mRNAs (Kashima et al., 2006, Fiorini et al., 2015, Jackson et al., 2010, Ping et al., 2014, Liu et al., 2014, Linder et al., 2011). The function of NOVA proteins on specific mRNA targets (e.g. splicing or stabilization) also depend on association with other proteins or specific experience/stimulation (Saito et al., 2016, Gill et al., 2017, Saito et al., 2019, Tajima et al., 2023). For example, NOVA2, a family molecule of NOVA1, shares a high degree of amino acid homology, expression pattern, binding motif (UCAU) and binding peaks on their target transcripts with NOVA1 (Saito et al., 2016). Nevertheless, their effects on alternative splicing are quite different, indicating the effects of additional factors. We noted that the I197V substitution does not result in loss of function of NOVA1 despite changes within the hydrophobic core of the KH domain (Figure 2c, Supplementary Figure 5). However, a comparison of the KH domain of NOVA1 with human-specific amino acids and ancient NOVA1 in the predicted 3D structure model suggests that the substitution from Isoleucine to Valine changes the interacting amino acid relationships within the KH domain by shortening one methyl group, affecting the interactions of the secondary structural elements that constitute KH domain (the second helix and the first and third helices) (Supplementary Figure 9a-b). Given these structural changes within the KH domain, as well as the predicted involvement of immediately adjacent amino acid residues in protein-protein interactions (Supplementary Figure 9d, Teplova et al., 2011), it is possible that human-specific amino acid substitutions affect NOVA1 function apart from its RNA-binding capacity. This explanation is now included in the revised results and discussion.

Regarding the AS analysis, I again note the challenge of critically analyzing authors' underlying data. Authors note that ~40% of the detected AS events also had NOVA peaks on them "implicating direct effects of changes in NOVA1 amino acid substitution" but it is entirely unclear how the former connects to the later. As noted above, there are many apparent AS events for which there is no apparent change in NOVA binding. And the counter to authors' note is that ~60% of the detected AS events contain no NOVA binding. Together, it is entirely unclear if/how the V197I variant can be mechanistically linked to a critical role in the observed changes in AS (or gene expression, or CLIP signal).

Regarding the reviewer's question about the absence of an obvious NOVA1 CLIP-peak on transcripts with AS, we think there are several possibilities underlying this observation as listed below.

(1) AS events on transcripts that do not have NOVA1 CLIP may be an indirect consequence of the action of RBPs (other than NOVA1) that are somehow affected by the I197V substitution. This will include the changes in binding pattern of the RBPs via interactions with NOVA1^{hu} or alternatively the RBP function itself was altered by AS (there are several splicing-related factors with differential AS; Supplementary Table 4).

(2) Another possibility is the nature of the UV-CLIP methodology. UV light at a wavelength of 254 nm used in CLIP experiments induces covalent bonds between RNA bases and proteins at the moment of irradiation (Ule et al., 2003, Licatalosi et al., 2008). UV irradiation can be applied to intact tissues (e.g., brain), does not cross-link proteins to proteins, therefore it shows high selectivity, only identifies direct protein-RNA interactions. This method utilizes the natural photo-reactivity of the RNA bases, especially pyrimidines, and specific amino acids, such as Cys, Lys, Phe, Trp, and Tyr at 254 nm irradiation (Hockensmith et al., 1986, Ule et al., 2005). In addition to this reaction preference, the UV cross-linking efficiency is relatively low, and the double-stranded structure is not easily cross-linked because the base pairs are often shielded from protein contact. RNA and protein with favorable interacting configuration are amenable to UV, while interactions with low frequencies or weak bindings are potentially difficult to be captured. Although various

variations of CLIP have been developed to optimize the UV cross-linking, the inherent issue of UV cross-linking remains to date.

Nonetheless, CLIP is a powerful method for identifying stable definitive RBP-mRNA interaction maps to study the functions and properties of RBPs of interest. This approach has been widely used to study dozens of RBPs, and the direct binding on the transcript is broadly accepted as strong evidence of a role in downstream regulations (Ule et al., 2003, Ule et al., 2005, Licatalosi et al., 2008, Sibley et al., 2016). This conclusion is supported by numerous genetic, biochemical binding, and mutation studies (Buckanovich et al., 1996, Buckanovich et al., 1997, Jensen et al., 2000, Dredge et al., 2003, Saito et al., 2016, Saito et al., 2019). Taken together, the CLIP provides direct points of action of the RBP in the transcripts, which is crucial in understanding the nature and biological role of the RBP *in vivo*. Similarly, in this study, NOVA1-CLIP allows comparison of the physical properties of the NOVA1 proteins *in vivo*, and provides an important basis for elucidating the physiological functions involved in NOVA1 in the mouse brain.

Regarding the vocalization data included in the manuscript, I would note that I am not expert in this field and so cannot fully weigh in on these data. Nevertheless, from a statistical perspective, it strikes me that these are modest changes at best (in some cases relying on differences not in means but in variance between samples). Perhaps this is considered acceptable within this field, but given my concerns with authors' treatment of statistical data from the molecular aspects of this work described above, I am concerned about the significance of these results as well.

We take the reviewer's concerns seriously. In response to the reviewers' suggestions and criticisms, we reviewed all statistical treatments in the vocalization data, and conducted additional analyses. These improvements have allowed for more detail and accurate comparisons of the nature of mouse vocalizations and have led to the discovery of the robust differences between genotypes.

Specifically, for each vocal property analysis, we noticed that the analysis in the first manuscript was insufficient in some parameters, which shows bimodal distributions; mice change their vocal characteristics depending on their stage of development, social context, and stimuli (Grimsley et al., 2011, Chabout et al., 2015, Lefebvre et al., 2020, Hammerschmidt et al., 2015). For example, in terms of peak frequency (Fq), neonatal mice show a clear bimodal distribution between P5 to P9 (i.e., high and low Fq), which later converges to a single peak (Grimsley et al., 2011). Similarly, syllable duration can be divided into two distributions depending on its length, and each characteristic is influenced by genotype, vocal induction stimuli, or experience (Castellucci et al., 2016, Castellucci et al., 2018, Enard et al., 2009, Hammerschmidt et al., 2015).

REDACTED

(left) Bimodality in USV peak frequency during mouse development. Figure from Grimsley et al., 2011. There are two clear peaks in the distribution. Pup isolation induced calls in mice show a bimodal distribution in their frequency (Fq), especially at 5-9 days of age, and to consolidate to one peak as they grow (Grimsley et al., 2011).

REDACTED

(right) Example spectrogram showing short and long duration USVs in adult male ultrasonic courtship vocalizations. Figure from Castellucci et al., 2018.

In the initial manuscript, we had simply averaged the values in the individual mouse syllables without taking into account their variable properties depending on context. However, if the data are compared by taking the overall average for parameters that are indeed bimodal, the results strongly reflect the more weighted distribution, and will mask the real characteristics in each distribution. To analyze these parameters more accurately, we assessed each bimodality, measuring the degree of separation between two distributions, and determined mean, weight, and cutoff value of each distribution to classify syllables accordingly.

Example of analysis for bimodal distribution. (left) Density plot of the maximum frequency ($F_{q_{max}}$) of syllable “d” observed in pup USVs. (right) The same data with left. D (Ashman’s D) score indicates the degree of separation between two distributions. $D > 2$ is an indicator of marked separation. To separate distributions, two Gaussians (blue and orange lines) fitted are overlaid. The Gaussian centers (blue and orange circles) and weights, the intercept of the two Gaussian distributions (black triangle) used as the cutoff between two distributions are labelled.

This method allowed for a more careful breakdown of mouse vocal characteristics and rigorous comparisons between genotypes. The major findings from these revisions are listed below.

*Major points in the revised USV analyses.

[Pup USV]

- *Nova1^{hu/hu}* produces significantly lower amplitude for ‘s’ syllable.
- *Nova1^{hu/hu}* produces less syllables containing pitch jumps.
- *Nova1^{hu/hu}* had increased proportion of complex syllables with high maximum frequency ($F_{q_{max}}$).

[Adult USV]

- *Nova1^{hu/hu}* had significantly lower start, mean and end peak frequencies (F_q) in long ‘s’ syllable.
- *Nova1^{hu/hu}* had increased frequency variance ($F_{q_{variance}}$) in complex syllables with high $F_{q_{max}}$.
- *Nova1^{hu/hu}* mice emitted a comparable number of vocalizations, having qualitative changes in some specific parameters both in pups and adults.
- Vocalization changes are specific to each developmental stage (or specific to social context).
- Heterozygous (*Nova1^{hu/wt}*) mice show intermediate values between *Nova1^{wt/wt}* and *Nova1^{hu/hu}* in both pup and adult USVs, suggesting that the effect of the I197V substitution in NOVA1 protein is dosage dependent.

Example for the vocal phenotypes in *Nova1^{hu/hu}* mice. (left) Frequency variance (Fq_{variance}) in high Fq_{max} in adult USVs. (right) Peak frequency (Fq) parameters (start, minimum, mean, maximum and end) in long duration "s" in adult USVs. Data is represented as boxplots. Each open circle indicates data from one adult male mouse; the average of the three recordings (see method section for details). p -values were calculated by Wilcoxon rank sum test and corrected with Bonferroni method. * p value < 0.05, ** p value < 0.01. *Nova1^{hu/hu}* N=13, *Nova1^{wt/wt}* N=14, *Nova1^{hu/wt}* N=13.

In summary, we have carefully reexamined the individual observations in vocalizations for a more appropriate and detailed analysis. These improvements have highlighted the nature of mouse vocalizations more precisely and enabled comparisons of vocal characteristics between genotypes. These data are included in the revised manuscript (Fig. 4, Supplementary Fig. 15 and main text for detailed description). To provide the reviewers and all readers access to the full vocalization data, we include all raw data and associated tables in supplementary materials (Supplementary Table 6-16).

Reviewer comments and Response (Response are in blue)

Reviewer #5 (Remarks to the Author)

The exploration of the genetic basis for human language and elaborate vocal communication is both fascinating and critical. This topic has been previously addressed but often without providing satisfactory answers. The study in question investigates the role of the NOVA1 gene, particularly a human-specific mutation, I197V, and its implications for vocal behavior. The findings present a compelling narrative, though some aspects warrant further scrutiny.

As a non-expert in genetics, my ability to evaluate the detailed genetic analysis is limited. However, the study's approach appears thorough and convincingly novel, with deep evidence supporting its conclusions. The introduction of the human-specific I197V variant in NOVA1 into mice (Nova1hu/hu) and the subsequent analysis of molecular and behavioral changes provide a significant contribution to our understanding of human evolution.

The relationship between the described NOVA1 mutation and vocalization is particularly intriguing. The study suggests that the I197V variant may have played a role in the evolution of human spoken language. The analysis indicates that this variant affects RNA regulation and alternative splicing in the brain, which in turn influences vocal behavior. However, while the evidence for changes in brain gene expression are compelling, the link to vocalization behavior appears somewhat limited and inconclusive. While it provides evidence suggesting a connection, it does not definitively prove it. The evidence mainly supports the hypothesis but lacks conclusive power.

In order to strengthen this link the authors generate a mouse mutant where the humanized mutation is introduced and compared well characterized ultrasonic vocalizations (USVs) behavior in mice, both in pups and adults.

a. It is unfortunate that raw vocalization data, particularly for adult courtship, is not presented in the main text, limiting the ability to fully assess the findings.

We appreciate the reviewer's comment and suggestion. To provide the reviewers and all readers access to the full data, we present the following files as Supplementary Tables.

- All syllable information observed in USV studies of pups and adult mice (Supplementary Table 6 and 11)
- Tables summarizing the call structure for each individual mouse (Supplementary Table 7 and 12)
- Tables summarizing the call structure for each genotype (Supplementary Table 8 and 13)

Also, we added the following tables accompanying the analyses (see explanation below)

- Bimodal distribution parameters on maximum frequency ($F_{q_{max}}$) in pup USVs (Supplementary Table 9)
- Proportion of high/ low $F_{q_{max}}$ in pup USVs in each genotype (Supplementary Table 10)
- Tables summarizing the call structure in short/long duration "s" in adult USVs (Supplementary Table 14)
- Proportion of syllable composition in high/ low $F_{q_{max}}$ in adult USVs (Supplementary Table 15)

- Tables summarizing the call structure in high/ low $F_{q_{max}}$ in adult USVs (Supplementary Table 16)

b. The performed analysis is reasonable but not highly sophisticated, potentially impacting the robustness of the conclusions.

We appreciate the reviewer's comments. The experimental setup for vocalization tests (USV induced method, recording and syllable detection system) used in this study are well established (Arriaga et al., 2012, Chabout et al., 2015) and the obtained USVs are sufficiently reliable. However, as the reviewer points out, the original submitted manuscript simply listed the observed data as it is with little explanation and discussions. Moreover, the statistical analyses performed were not sufficient. Based on reviewers' input, we applied a more appropriate statistical approach to the vocalization analysis. From these observation in pup- and adult- vocalization, the possible effects of the NOVA1 I197V substitution on vocal behavior are now discussed and included in the revised text. Please refer the revised analyses from next response.

c. The statistical results are weak, with all the significant tests showing significance levels around 0.05 except for Fig 4E and 4G. There is no mention of corrections for multiple comparisons (at least I couldn't find it), raising concerns about the reliability of the significant findings. If these tests were not corrected it is highly likely that they would not pass a significance test and will make the conclusions obsolete.

We appreciate the reviewer's criticism. In response to reviewers' remarks, all statistic tests are now corrected for multiple comparisons (with Bonferroni method) between genotypes. However, correction was not applied for the parameters in call element/ structure. This is because individual properties are assumed to be related to each other (e.g., each syllable compositions, variable of the frequency peak [start, minimum, maximum, end, mean], frequency variance and bandwidth, duration, and number of calls, etc.), which increases type 2 error caused by overcorrection. This is now stated in the revised manuscript as below to allow the reader to consider these statistic methods.

Methods (USV analysis section)

“Statistical analysis was performed by pairwise Wilcoxon rank sum tests, with correction (Bonferroni) for multiple comparisons between genotypes. Correction was not applied for the parameters in call structure. This is because individual properties are assumed to be related to each other, which increases type 2 error caused by overcorrection.”

The revision of the statistics revealed that several parameters that were considered significant in the initial manuscript did not meet the statistical significance level due to their large variability. Therefore, we have revised all the USV analyses both in pups and adults. In conjunction with the reviewer's suggestions below, we have improved them to capture the effects of the I197V substitution on vocalization more accurately.

It is also suspected that the high levels of significance in Fig 4E,G are due to the huge number of samples and may not represent a real change. For example the huge green bars on the histograms on fig4g at about 120KHz are clearly and artifact and also are likely beyond the hearing range of the mice. It is surprising that a research team with a strong genetic analysis would apply less stringent criteria to non-genetic data.

We thank the reviewer for pointing this out. In the original manuscript, Figures 4E and G showed a plot of all syllables observed in each genotype of pup, and Mann-Whitney statistical analysis was performed to test for differences. However, as the reviewer pointed out, this method is strongly affected by individual differences in mice, and furthermore, as is evident from the shape of the histograms, the data exhibit bimodal distributions, which means that the method originally used (accumulate all data and compare its mean or median) was not appropriate.

Thus, we adopted the more appropriate analytical methods for those bimodal parameters as explained below. Also, we revised all statistics with correction (Bonferroni) for multiple comparisons as described above.

Bimodal distribution
(from original manuscript Fig4E)

Example of bimodal distribution. Density plot of the maximum frequency ($F_{q_{max}}$) of syllable *d* observed in pup USVs.

Bimodality in USV peak frequency during mouse development

Figure from Grimsley et al., 2011

Distributions of the frequency of mouse USVs. There are two clear peaks in the distribution. The frequency of the higher peak gradually reduces with age.

Pup isolation induced calls in mice have been reported to show a bimodal distribution in their frequency (F_q), especially at 5-9 days of age, and to consolidate to one peak as they grow (Grimsley et al., 2011). In our study, the pups are 7 days of age, and regardless of the genotypes, the USVs showed clear bimodal distribution.

REDACTED

To analyze these parameters more accurately, we assessed the bimodality by Ashman's D score (Ashman et al., 1994). Ashman's D measures the degree of separation between two distributions. $D > 2$ is an indicator of marked separation. We found that the scores indicate clear separations in two peaks in "s", "d" and "m" syllables ($D > 2$), except "u" syllable ($D = 1.603$) in pup USVs. To separate these two peaks, we fitted two Gaussians (Benaglia et al., 2009) for each syllable type showing bimodal distribution (Supplementary Table 9).

*The following data was presented using the parameters for the maximum frequency ($F_{q_{max}}$), which showed the pronounced bimodality, and we confirmed a similar trend in the minimum frequency ($F_{q_{min}}$) in pup USVs.

Density plots of frequency maximum ($F_{q_{max}}$) in each syllable (s, d, u, m) in pup isolation induced USV test. Two Gaussians (blue and orange lines) fitted are overlaid. The Gaussian centers (blue and orange circles) and weights, the intercept of the two Gaussian distributions (black triangle) used as the cutoff between high and low $F_{q_{max}}$ USVs are labelled. The dot plot at the bottom of the histogram shows the mean (black dots) and standard deviation (whiskers) by peak for each genotype. There are no significant differences between genotypes.

Performing these fits allowed us to calculate the average low and high frequency peaks, cutoff values between the two peaks (intersection point of each distribution), and proportion of each USV type. Individual syllables were classified either into those with 'low' $F_{q_{max}}$ or those with 'high' $F_{q_{max}}$ according to the cutoff value of each syllable type, and their ratios were determined to compare between genotypes.

We found that for the syllable "m", which contains multiple jumps, the proportion of low $F_{q_{max}}$ decreased whereas the proportion of high $F_{q_{max}}$ increased in *Nova1^{hu/hu}* pups. Although not statistically significant, similar trend was observed for syllable "d" (Supplementary Table 10). The changes observed in isolation induced pup USVs were not observed in courtship vocalizations in adults, suggesting that these changes are transient or context-dependent observations.

Ratio of each syllable in high or low Fq_{max} USVs. Syllables that showed a bimodal distribution (Ashman's $D > 2$) in the previous panel were analyzed (*s*, *d* and *m*). The ratio of syllables that belong to each distribution (high or low) are calculated for the total number of each syllable type. Data is represented as boxplots. Each circle indicates data from a single pup. p -values were calculated by Wilcoxon rank sum test and corrected with Bonferroni method. * p value < 0.05 , ** p value < 0.01 . $Nova1^{hu/hu}$ $N=41$, $Nova1^{hu/wt}$ $N=23$, $Nova1^{wt/wt}$ $N=40$ pups.

We also verified adult USVs, whether the green peaks in the high-frequency region of Fig. 4G (in the initial manuscript, which are overlapping green bar: $Nova1^{wt/wt}$ and yellow bar: $Nova1^{hu/hu}$) were artifacts. These signals are only observed in the plot of maximum frequency (Fq_{max}) and not observed in minimum frequency (Fq_{min}).

Density plot of the Fq_{max} of syllable *d*, *u* and *m* observed in adult USVs. A peak signal in the high frequency region (arrow) is clearly discernible in the maximum frequency (Fq_{max}) plot.

We first visually check each spectrogram of USVs. Observation of the spectrogram showed that these high Fq_{max} signals are indeed USVs with linear characteristics.

Example of USVs containing syllable with high maximum frequency (Fq_{max}) spectrogram from our study

Examples of USVs spanning high frequency region. The x-axis indicates time (sec) and y-axis indicates peak frequency (kHz). A yellow dotted line is drawn at 100 kHz. USVs with signals above 100 kHz are indicated by orange arrows.

Several studies have shown that mouse vocalizations range from 30-120 kHz (Grimsley et al., 2011, Vogel et al., 2019, Yao et al., 2023), and the high-frequency signals in Fig. 4G are within the range. These high $F_{q_{max}}$ signals are reported in harmonic syllables, syllables containing pitch jumps, and some simple syllables.

REDACTED

Examples of mouse USVs spanning the high-frequency regions from previous literature (Vogel et al., 2019. And Grimsley et al., 2011). USVs with signals above 100 kHz are indicated by orange arrows.

Of note, as the reviewer pointed out, these high-frequency signals are clearly separated from the main population, rather than a continuous distribution from the main population (the tip of a gentle normal distribution) as seen in the above density plots. Ashman's D score indicates clear separation ($D=2.281$) for this high frequency population.

Density plot of frequency maximum ($F_{q_{max}}$) in syllables in adult courtship USV test. Two Gaussians (blue and orange lines) fitted are overlaid. The Gaussian centers (blue and orange circles) and weights are labeled. The black star indicates 100kHz cutoff of High $F_{q_{max}}$ USV and Low $F_{q_{max}}$ USVs. The dot plot at the bottom of the histogram shows the mean (black dots) and standard deviation (whiskers) in each USVs for genotype. There are no significant differences between genotypes.

To analyze the nature of these USVs containing high-frequency signals (high $F_{q_{max}}$), we fitted two Gaussians to syllables observed in adult courtship USV tests.

The high peak on $F_{q_{max}}$ was found to be 107.1 ± 2.1 kHz, whereas main peak was 80.8 ± 11.5 kHz (mean \pm standard deviation) in average. The high $F_{q_{max}}$ peak occupied roughly 8.5 % in total USVs. We set the cutoff as 100 kHz, at which the high $F_{q_{max}}$ peak begins to

appear. USVs with Fq_{max} above 100 kHz were operationally defined as the high Fq_{max} USVs and distinguished from the main population below 100 kHz (low Fq_{max} USVs). The high Fq_{max} USVs contains more syllables with pitch jump compared to the main population: in low Fq_{max} USVs, the composition of syllable “s”, “d”, “u” and “m” were roughly 77%, 10%, 6%, and 4%, whereas in high Fq_{max} USVs, they are roughly 41%, 23%, 14%, and 11%. There was no difference in composition among genotypes (Supplementary Table 15).

We then compared the USV parameters in each population (both in high Fq_{max} and low Fq_{max}) between genotypes. While most of the properties were comparable between genotypes, frequency variances ($Fq_{variance}$) in high Fq_{max} USVs were significantly greater in all syllables containing pitch jump (“d”, “u” and “m”) in *Nova1^{hu/hu}* (Supplementary Table 16).

We also detected notable differences in syllable duration between genotypes in adult USVs in the revised analyses. Adult male B6 mice produce two distinct classes of courtship USVs: short-duration and long-duration USVs (Castellucci et al., 2016, Castellucci et al., 2018, Enard et al., 2009).

REDACTED

Example spectrogram showing short and long duration USVs in adult male ultrasonic courtship vocalizations. Figure from Castellucci et al., 2018.

The “s” syllables detected in our study also showed a bimodal distribution in the duration (Ashman’s $D=5.087$). Two Gaussian were fitted to this distribution, and the average and cutoff value for each peak were calculated. The duration of short- and long-“s” were found to be 22.6 ± 9.1 ms and 69.0 ± 40.1 ms (mean \pm standard deviation) in average, respectively. Adult mice produced roughly 71% of short and 29% for long “s”. The cutoff duration between short and long USVs was determined at 44 ms.

We compared the USV parameters in each population (both in short- and long- “s”) between genotypes. *Nova1^{hu/hu}* mice had a significantly lower start peak frequency, lower end and lower mean peak frequency in long-duration “s” (Supplementary Table 14), indicating modulation in Fq for certain type of syllables in *Nova1^{hu/hu}* adult mice.

(left) Density plot of duration in syllable “s” in adult courtship USVs. Two Gaussians (blue and pink lines) fitted are overlaid. The Gaussian centers (blue and pink circles) and weights, the intercept of the two Gaussian distributions (black triangle) used as the cutoff between long and short duration s are labelled. The dot plot at the bottom of the histogram shows the mean (black dots) and standard deviation (whiskers) by peak for each genotype. There are no significant differences between genotypes. (right) Peak frequency (Fq) parameters (start, minimum, mean, maximum and end) in long duration “s”. Data is represented as boxplots. Each open circle indicates data from one adult male mouse; the average of the three recordings (see method section for details). *p*-values were calculated by Wilcoxon rank sum test and corrected with Bonferroni method. * *p* value <0.05, ** *p* value <0.01. *Nova1^{hu/hu}* N=13, *Nova1^{wt/wt}* N=14, *Nova1^{wt/wt}* N=13.

From these revised and additional analyses, we detected more robust changes in vocalizations both in pups and adult humanized NOVA1 mice. High frequency USVs in adult male mouse have been reported to be emitted more during social interaction tasks (Lefebvre et al., 2020). Also, female mice are reported to be attracted to male mice that emit more complex USVs (Chabout et al., 2015). Increasing the proportion of higher frequency USVs (in pup) or increasing their complexity (in adult) in *Nova1^{hu/hu}* mice may have some potential advantages in social interactions in mice. However, as auditory frequency resolution in mice has been reported to be limited (Portfors et al., 2011), it is unclear to what extent mice recognize vocalization changes in humanized NOVA1 mice. In fact, we conducted a preference experiment in mother mice using humanized NOVA1 pup USVs and did not observe preferences in the mother’s choice (Supplemental Figure 16). This may suggest a limitation of discriminative resolution in mouse hearing.

In summary, the USV analysis methods were revised and improved to extract vocal characteristics more accurately, and to compare them between genotypes. These refinement leads to several new findings on the changes in vocalization in *Nova1^{hu/hu}* mice. These data are included in the revised manuscript (Fig. 4, Supplementary Figure 15). The values for all other

parameters not shown in the figures were also disclosed as supplementary material (Supplementary Table 6-16).

In conclusion, the topic of genetic mutations contributing to human language is of extreme interest and potential. The study makes a significant effort to link the NOVA1 mutation to changes in vocalization behavior, but the connection remains tentative. The evidence for a real change in vocal behavior due to the genetic mutation needs to be stronger to make definitive conclusions. I would expect more rigorous analysis and if required additional data to establish the link. The authors have laid important groundwork, but further efforts are necessary to solidify the relationship between the genetic findings and the phenotypic expressions in vocalization.

We are grateful to the reviewer for the interest and careful review of this study. As mentioned above, we have carefully reexamined our statistics and the individual observations for a more appropriate/detailed analysis. These improvements have highlighted the nature of mouse vocalizations precisely and enabled to compare the vocal characteristics between genotypes.

We also provided a further update on our analysis of NOVA1 expression in the brain, showing that its expression is extremely high in the PAG region, which plays a crucial function in mammalian vocalizations (Fig. 3a-b, Supplementary Fig. 10-11). Furthermore, we showed that the binding target transcripts of NOVA1 are strongly enriched in behavior-related terms (Fig. 2h, Supplementary Fig. 8e, 8j) and that many vocalization-related genes are among its targets (Supplementary Fig. 14). With regard to the enrichment in vocalization behavior term of differential splicing changes in *Nova1^{hu/hu}* mice brain (Fig. 3f), we additionally examine its significance by random resampling tests. The results indicate that enrichment to the vocal behavior term is significantly higher than events observed by chance (Supplementary Fig. 13). These additional data provide further support for the association between NOVA1 and vocalization, and the effects of the I197V substitution. We compared in detail our findings on vocalization changes with those from several previous studies and discussed the biological significance of vocal changes and remaining issues in the study (please refer Supplementary Table 17 and Discussion section).

As the reviewer points out, although we have worked to elucidate the molecular basis of humanized NOVA1 on vocalization in mice, we have yet to demonstrate a clear/ exact molecular mechanism to explain the phenotype. We argue that an initial discovery of a candidate gene and specific amino acid substitutions is just as critical, and necessary on the path to extracting the exact mechanism. As a result, we believe that this study will have an important impact on the field of genetics and evolution of vocal learning and spoken language. We think more detailed investigation in the mechanism is beyond the scope of a single research study.

Taken together, in response to suggestions and criticisms from reviewers, we have performed more rigorous analyses and additional analyses in the population genetics, transcriptomic, and vocalization analyses, and presented these data transparently. These improvements have allowed us to suggest a strong link between the specificity of the one amino acid substitution unique to modern humans and vocal behavior. We believe that these data provide one interesting insight into the molecular basis of modern human evolution.

References for the response letter

Luque et al., 1991, Anti-Ri: an antibody associated with paraneoplastic opsoclonus and breast cancer.
<https://pubmed.ncbi.nlm.nih.gov/2042940/>

Buckanovich et al., 1993, Nova, the paraneoplastic Ri antigen, is homologous to an RNA-binding protein and is specifically expressed in the developing motor system
<https://pubmed.ncbi.nlm.nih.gov/8398153/>

Simard et al., 2020, Clinical spectrum and diagnostic pitfalls of neurologic syndromes with Ri antibodies
<https://pubmed.ncbi.nlm.nih.gov/32170042/>

Velazquez et al., 2014, Anti-Ri Antibody Paraneoplastic Syndrome Without Opsoclonus-Myoclonus
<https://www.neurores.org/index.php/neurores/article/view/261/259#:~:text=Anti%2DRi%20antibody%20pa,aneoplastic%20neurologic.course%20of%20heterogeneous%20neurological%20manifestations.>

Musunuru et al., 2008, Paraneoplastic opsoclonus-myoclonus ataxia associated with non-small-cell lung carcinoma
<https://www.ncbi.nlm.nih.gov/pmc/articles/PMC2652648/>

Fadare et al., 2004, Anti-Ri antibodies associated with short-term memory deficits and a mature cystic teratoma of the ovary
<https://www.ncbi.nlm.nih.gov/pmc/articles/PMC534089/>

Yang et al., 2023, NOVA1 prevents overactivation of the unfolded protein response and facilitates chromatin access during human white adipogenesis
<https://academic.oup.com/nar/article/51/13/6981/7184163>

Villate et al., 2014, Nova1 is a master regulator of alternative splicing in pancreatic beta cells
<https://pubmed.ncbi.nlm.nih.gov/25249621/>

Eizirik et al., 2012, The human pancreatic islet transcriptome: expression of candidate genes for type 1 diabetes and the impact of pro-inflammatory cytokines
<https://pubmed.ncbi.nlm.nih.gov/22412385/>

Tajima et al., 2023, NOVA1 acts on Impact to regulate hypothalamic function and translation in inhibitory neurons
<https://pubmed.ncbi.nlm.nih.gov/36716149/>

Whitney et al., 1973, Ultrasonic emissions: Do they facilitate courtship of mice?
<https://psycnet.apa.org/fulltext/1974-04797-001.pdf>

Neunuebel et al., 2015, Female mice ultrasonically interact with males during courtship displays
<https://www.ncbi.nlm.nih.gov/pmc/articles/PMC4447045/#:~:text=Unexpectedly%2C%20this%20approach%20revealed%20that,as%20coming%20from%20males%20instead>

Moles et al., 2007, Ultrasonic vocalizations emitted during dyadic interactions in female mice: A possible index of sociability?
<https://www.sciencedirect.com/science/article/pii/S0166432807000587>

Trujillo et al. in 2021, Reintroduction of the archaic variant of NOVA1 in cortical organoids alters neurodevelopment
<https://www.science.org/doi/10.1126/science.aax2537>

Atkinson et al., 2019, No Evidence for Recent Selection at FOXP2 among Diverse Human Populations

<https://pubmed.ncbi.nlm.nih.gov/30078708/>

Zeng et al., 2016, Statistical Tests for Detecting Positive Selection by Utilizing High-Frequency Variants
<https://www.ncbi.nlm.nih.gov/pmc/articles/PMC1667063/>

Dredge et al., 2005, Nova autoregulation reveals dual functions in neuronal splicing
<https://www.ncbi.nlm.nih.gov/pmc/articles/PMC1142566/>

Saito et al., 2016, NOVA2-mediated RNA regulation is required for axonal pathfinding during development
<https://www.ncbi.nlm.nih.gov/pmc/articles/PMC4930328/>

Boulle et al., 1993, A point mutation in the FMR-1 gene associated with fragile X mental retardation
<https://www.nature.com/articles/ng0193-31>

Siomi et al., 1994, Essential role for KH domains in RNA binding: Impaired RNA binding by a mutation in the KH domain of FMR1 that causes fragile X syndrome
<https://www.sciencedirect.com/science/article/pii/0092867494902321?via%3Dihub>

Jones et al., 1995, Mutations in *gld-1*, a female germ cell-specific tumor suppressor gene in *Caenorhabditis elegans*, affect a conserved domain also found in Src-associated protein Sam68
<https://genesdev.cshlp.org/content/9/12/1491.long>

Buckanovich et al., 1996, The Onconeural Antigen Nova-I Is a Neuron-Specific RNA-Binding Protein, the Activity of which Is Inhibited by Paraneoplastic Antibodies
<https://www.jneurosci.org/content/jneuro/16/3/1114.full.pdf>

Buckanovich et al., 1997, The Neuronal RNA Binding Protein Nova-1 Recognizes Specific RNA Targets In Vitro and In Vivo
<https://www.ncbi.nlm.nih.gov/pmc/articles/PMC232172/pdf/173194.pdf>

Kashima et al., 2006, Binding of a novel SMG-1-Upf1-eRF1-eRF3 complex (SURF) to the exon junction complex triggers Upf1 phosphorylation and nonsense-mediated mRNA decay
<https://www.ncbi.nlm.nih.gov/pmc/articles/PMC1361706/>

Fiorini et al., 2015, Human Upf1 is a highly processive RNA helicase and translocase with RNP remodelling activities
<https://www.nature.com/articles/ncomms8581>

Jackson et al., 2010, The mechanism of eukaryotic translation initiation and principles of its regulation
<https://www.nature.com/articles/nrm2838>

Ping et al., 2014, Mammalian WTAP is a regulatory subunit of the RNA N6-methyladenosine methyltransferase
<https://www.nature.com/articles/cr20143>

Liu et al., 2014, A METTL3–METTL14 complex mediates mammalian nuclear RNA N6-adenosine methylation
<https://www.nature.com/articles/nchembio.1432>

Linder et al., 2011, From unwinding to clamping - the DEAD box RNA helicase family
<https://www.nature.com/articles/nrm3154>

Saito et al., 2019, Differential NOVA2-mediated splicing in excitatory and inhibitory neurons regulates cortical development and cerebellar function
<https://www.ncbi.nlm.nih.gov/pmc/articles/PMC6649687/>

Esposito et al., 1999, Complete mutism after midbrain periaqueductal gray lesion
https://journals.lww.com/neuroreport/fulltext/1999/03170/complete_mutism_after_midbrain_periaqueductal_gray.4.aspx

Jürgens et al., 1994, The role of the periaqueductal grey in vocal behaviour
<https://www.sciencedirect.com/science/article/pii/0166432894900175>

Jürgens et al., 2002, Neural pathways underlying vocal control
<https://pubmed.ncbi.nlm.nih.gov/11856561/>

Scally et al., 2012, Revising the human mutation rate: implications for understanding human evolution
<https://www.nature.com/articles/nrg3295>

Mounier et al., 2019, Deciphering African late middle Pleistocene hominin diversity and the origin of our species
<https://www.nature.com/articles/s41467-019-11213-w>

Fitch et al., 2016, Monkey vocal tracts are speech-ready
<https://www.science.org/doi/10.1126/sciadv.1600723>

Boë et al., 2017, Evidence of a Vocalic Proto-System in the Baboon (*Papio papio*) Suggests Pre-Hominin Speech Precursors
<https://www.ncbi.nlm.nih.gov/pmc/articles/PMC5226677/>

Nishimura et al., 2022, Evolutionary loss of complexity in human vocal anatomy as an adaptation for speech
<https://www.science.org/doi/10.1126/science.abm1574>

Faull et al., 2019, The midbrain periaqueductal gray as an integrative and interoceptive neural structure for breathing
https://www.sciencedirect.com/science/article/pii/S0149763418306067?ref=pdf_download&fr=RR-2&rr=8af82bc4cbde4349

Kittelberger et al., 2006, Midbrain Periaqueductal Gray and Vocal Patterning in a Teleost Fish
https://journals.physiology.org/doi/full/10.1152/jn.00067.2006?rfr_dat=cr_pub++0pubmed&url_ver=Z39.88-2003&rfr_id=ori%3Arid%3Aacrossref.org

Arriaga et al., 2012, Of Mice, Birds, and Men: The Mouse Ultrasonic Song System Has Some Features Similar to Humans and Song-Learning Birds
<https://www.ncbi.nlm.nih.gov/pmc/articles/PMC3468587/>

Chabout et al., 2015, Male mice song syntax depends on social contexts and influences female preferences
<https://www.ncbi.nlm.nih.gov/pmc/articles/PMC4383150/>

Grimsley et al., 2011, Development of Social Vocalizations in Mice
<https://journals.plos.org/plosone/article?id=10.1371/journal.pone.0017460>

Vogel et al., 2019, Quantifying ultrasonic mouse vocalizations using acoustic analysis in a supervised statistical machine learning framework
<https://www.nature.com/articles/s41598-019-44221-3>

Yao et al., 2023, A review of ultrasonic vocalizations in mice and how they relate to human speech
<https://pubs.aip.org/asa/jasa/article/154/2/650/2905641/A-review-of-ultrasonic-vocalizations-in-mice-and>

Ashman et al., 1994, Detecting bimodality in astronomical datasets
<https://articles.adsabs.harvard.edu//full/1994AJ....108.2348A/0002348.000.html>

Benaglia et al., 2009, mixtools: An R Package for Analyzing Mixture Models
<https://cran.r-project.org/web/packages/mixtools/citation.html>

Castellucci et al., 2016, Knockout of Foxp2 disrupts vocal development in mice.
<https://www.nature.com/articles/srep23305>

Castellucci et al., 2018, The temporal organization of mouse ultrasonic vocalizations
<https://journals.plos.org/plosone/article?id=10.1371/journal.pone.0199929>

Enard et al., 2009, A humanized version of Foxp2 affects cortico-basal ganglia circuits in mice
<https://pubmed.ncbi.nlm.nih.gov/19490899/>

Hammerschmidt et al., 2015, A humanized version of Foxp2 does not affect ultrasonic vocalization in adult mice
<https://onlinelibrary.wiley.com/doi/full/10.1111/gbb.12237>

von Merten et al., 2021, A humanized version of Foxp2 affects ultrasonic vocalization in adult female and male mice
<https://pubmed.ncbi.nlm.nih.gov/34342113/>

Lefebvre et al., 2020, Social context increases ultrasonic vocalizations during restraint in adult mice
<https://link.springer.com/article/10.1007/s10071-019-01338-2>

Chabout et al., 2015, Male mice song syntax depends on social contexts and influences female preferences, Ultrasonic courtship vocalizations of male house mice contain distinct individual signatures
<https://www.frontiersin.org/journals/behavioral-neuroscience/articles/10.3389/fnbeh.2015.00076/full>

Portfors et al., 2011, Spatial organization of receptive fields in the auditory midbrain of awake mice
<https://pubmed.ncbi.nlm.nih.gov/21807069/>

Holy, T. E. & Guo, Z. 2005, Ultrasonic Songs of Male Mice
<https://journals.plos.org/plosbiology/article/file?id=10.1371/journal.pbio.0030386&type=printable>

Burkett et al., 2015, A semi-automated pipeline for standardizing vocal analysis across models
<https://www.ncbi.nlm.nih.gov/pmc/articles/PMC4446892/>

Maricic et al., 2021, Comment on "Reintroduction of the archaic variant of NOVA1 in cortical organoids alters neurodevelopment"
<https://pubmed.ncbi.nlm.nih.gov/34648345/>

Herai et al., 2021, Response to Comment on "Reintroduction of the archaic variant of NOVA1 in cortical organoids alters neurodevelopment"
<https://pubmed.ncbi.nlm.nih.gov/34648331/>

Riesenberg et al., 2023, Efficient high-precision homology-directed repair-dependent genome editing by HDRobust
<https://www.nature.com/articles/s41592-023-01949-1>

Gill et al., 2017, Regulated Intron Removal Integrates Motivational State and Experience
<https://pubmed.ncbi.nlm.nih.gov/28525754/>

Ule et al., 2003, CLIP Identifies Nova-Regulated RNA Networks in the Brain
<https://pubmed.ncbi.nlm.nih.gov/14615540/>

Ule et al., 2005, CLIP: a method for identifying protein-RNA interaction sites in living cells
<https://pubmed.ncbi.nlm.nih.gov/16314267/>

Hockensmith et al., 1986, Laser cross-linking of nucleic acids to proteins. Methodology and first applications to the phage T4 DNA replication system
<https://pubmed.ncbi.nlm.nih.gov/3949776/>

Licatalosi et al., 2008, HITS-CLIP yields genome-wide insights into brain alternative RNA processing
<https://www.nature.com/articles/nature07488>

Sibley et al., 2016, Lessons from non-canonical splicing
<https://www.nature.com/articles/nrg.2016.46>

Jensen et al., 2000, Nova-1 regulates neuron-specific alternative splicing and is essential for neuronal viability
<https://pubmed.ncbi.nlm.nih.gov/10719891/>

Dredge et al., 2003, Nova regulates GABA(A) receptor gamma2 alternative splicing via a distal downstream UCAU-rich intronic splicing enhancer
<https://pubmed.ncbi.nlm.nih.gov/12808107/>

Hubisz et al., 2020, Mapping gene flow between ancient hominins through demography-aware inference of the ancestral recombination graph
<https://pubmed.ncbi.nlm.nih.gov/32760067/>

Vaughn et al., 2024, Fast and Accurate Estimation of Selection Coefficients and Allele Histories from Ancient and Modern DNA
<https://pubmed.ncbi.nlm.nih.gov/39078618/>

Hejase et al., 2022, A Deep-Learning Approach for Inference of Selective Sweeps from the Ancestral Recombination Graph
<https://pubmed.ncbi.nlm.nih.gov/34888675/>

Yan et al., 2015, Systematic discovery of regulated and conserved alternative exons in the mammalian brain reveals NMD modulating chromatin regulators
<https://www.pnas.org/doi/full/10.1073/pnas.1502849112>

Waidmann et al 2024, bioRxiv, Mountable miniature microphones to identify and assign mouse ultrasonic vocalizations
<https://www.biorxiv.org/content/10.1101/2024.02.05.579003v1>

Pfenning et al 2014, Convergent transcriptional specializations in the brains of humans and song-learning birds
<https://www.science.org/doi/10.1126/science.1256846>

Gedman et al 2022, Convergent gene expression highlights shared vocal motor microcircuitry in songbirds and humans
<https://www.biorxiv.org/content/10.1101/2022.07.01.498177v1>

Jarvis et al., 2019, Evolution of vocal learning and spoken language
<https://pubmed.ncbi.nlm.nih.gov/31604300/>

Response to the reviewers' comments

Reviewer #1 (Remarks to the Author):

The authors have have done an impressive amount of work to address the concerns of the reviewers. All our concerns are adequately addressed.

We are very grateful to the reviewers for their guidance and suggestions that have improved the quality of our research.

Reviewer #2 (Remarks to the Author):

In this revised version, the authors comprehensively answered many of the reviewers' questions, including demonstrating the specificity of NOVA1 to the CNS, excluding motor alterations in their mouse model, improving their vocalization data and statistics with comparative analyses with other publications, confirming the strong selection pressure at the NOVA1 197V, include more details about their bioinformatic pipelines, performed a GO analysis and found genes related with speech in certain brain regions, included off-target analyses, showed that the 197V substitution is not causing loss of function, among several other clarifications.

The manuscript is a much-improved version, and several novel data back up their findings. The observation that the archaic NOVA1 variant is linked to language is fascinating and corroborates previous observations made in human cells. Nonetheless, certain aspects of the introduction need to be scientifically accurate.

1) Maricic et al.'s technical criticism has been previously responded to by Herai et al. Science 2021. It nicely demonstrated that even though one alteration occurred in one clone of one of the cell lines, this alteration changed the phenotype from Arc/Arc to Arc/KO. Thus, the specific clone in question has only the Arc version of NOVA1, confirming the original results from Trujillo et al. 2021. Therefore, the authors should not use Maricic et al. 2021 to highlight a technical inconsistency in Trujillo's original publication.

2) I was surprised to learn about the work of Riesenberg et al., 2023 and decided to check it out more carefully. In fact, the authors claimed somewhere hidden in the manuscript that they could not reproduce the organoid morphological changes upon replacing the NOVA1 version with the archaic variant in a single cell line. However, upon close inspection, the work does not challenge the original Trujillo et al. 2021 publication. Technically, the experiment was done with a single allele being substituted, using a single cell line and a completely different brain organoid protocol to analyze organoid morphology. Moreover, the authors only analyze organoid morphology as a phenotype. Organoid morphology is variable and dependent on the genetic background of the cell line (Trujillo et al. used different iPSC lines, not just one). Based on this, one cannot conclude that the experiments done by Riesenberg fall short of trying to reproduce the effects of NOVA1 archaic version in the human model. Thus, as of now, no publication experimentally challenged the data of Trujillo et al. 2021 in a rigorous fashion. This sentence in the introduction needs to be altered for scientific accuracy and to avoid propagating wrong information:

“Studies have explored the NOVA1 I197V variant by reverting the ancestral isoleucine 197 variant back into human iPSC-derived organoids, revealing morphological and electrophysiological changes *in vitro*¹³. However, these effects were not observed in another study that reintroduced the same substitution in different iPSCs³⁹. This discrepancy underscores challenges of obtaining consistent results with varying experimental methods and materials *in vitro*^{40,41}. “

Other than these adjustments, the manuscript compiles a series of experimental evidence that links, for the first time, a modern-human-specific point mutation to vocalization. This observation opens novel perspectives to study this alteration from evolutionary, linguistic, and neuroscience perspectives.

We appreciate the reviewer's rigorous guidance and suggestions for correction. We are recognizing the difficulties in analysis using cell lines, including iPS cells; different protocols and

culture conditions can produce different results, even for the same cell line. Trujillo et al. have indeed carefully examined their experimental system in their rebuttal experiments, and their results cannot be negated by experiments performed with different cell lines and different protocols/ analysis. In our previous version of the manuscript, we intended to describe differing results as an issue of experimental systems using cell lines. Our current goal should not be to adjudicate differing results in a system that does not bear on our current paper. While appreciating this reviewer's points regarding Riesenberg et al., 2023, these issues should be considered by those involved, not in the brief mention of the literature here. In recognition of the reviewer's points, we have changed the text as follows (in green) so that this information is more accurately conveyed.

Before:

Studies have explored the NOVA1 I197V variant by reverting the ancestral isoleucine 197 variant back into human iPSC-derived organoids, revealing morphological and electrophysiological changes *in vitro*¹³. However, these effects were not observed in another study that reintroduced the same substitution in different iPSCs³⁹. This discrepancy underscores challenges of obtaining consistent results with varying experimental methods and materials *in vitro*^{40,41}.

After:

Studies have explored the role of the NOVA1 I197V variant by reverting the ancestral isoleucine 197 variant back into human iPSC-derived organoids reveal morphological and electrophysiological changes *in vitro* (Refs: Trujillo et al., 2021, Herai et al., 2021). While these effects were not observed in a study that reintroduced the same substitution in different iPSCs (Ref: Riesenberg et al., 2023), technical concerns continue to make definitive conclusions about the nature of the NOVA1 I197V variant in brain challenging. Therefore, we generated humanized mice harboring this variant to study its consequences for RNA regulation and behavior *in vivo*.

Reviewer #4 (Remarks to the Author):

In this revised manuscript, Tajima and colleagues have made extensive changes in response to feedback from original reviewers. I commend authors on their approach to addressing the concerns raised by this (and other) reviewers, it is a remarkable amount of work that has been clearly dedicated to this revised manuscript. Nevertheless, and in spite of my continued general interest in the work presented here, for reasons elaborated upon below I admit to being underwhelmed by the experimental support that is presented for the claimed conclusions, and so can't provide a strong argument in favor of its publication.

One important change in the revised manuscript is a change in the conclusions about the RNAseq and CLIPseq data generated by authors. Whereas the original manuscripts had sections with the following titles: "Humanized NOVA1 mice show normal development, but have specific gene expression changes in brain" and "Human and ancestral NOVA1 have the same RNA-binding activity but distinct RNA targets", the revised approach to data analysis in the current manuscript led authors to remove any such claims, so that the equivalent sections are now titled: "Humanized NOVA1 mice are comparable to wild-type mice in development and gene expression in the brain" and "Modern human specific amino acid substitution does not affect sequence specific RNA-binding capacity of NOVA1". In my view, this is certainly a positive improvement in the quality of the science, and it relieves some of my prior criticism regarding the disconnect between binding targets, AS targets, and overall changes in gene expression. On the other hand, it does reduce the experimental support for a biologically meaningful impact of the I197V variant. Indeed, at a molecular level authors have now reduced the molecular argument to the idea that there are changes in AS in the humanized variant. Certainly this is a plausible outcome of an allele of a protein like NOVA1, but the results shown here are again modest in impact, and modest in statistical significance.

When considering the changes in AS associated with the I197V variant as described in the current manuscript, I would argue that by themselves they aren't sufficient to make a strong claim. Certainly if this were a manuscript exclusively focusing on the molecular changes associated with this variant, one would expect considerably more work in support of these claims. Authors correctly note that interactions of RBPs with spliced RNAs can be complex, and indeed complicated in how they impact AS, but alleles with single amino acid substitutions are likely to be much less pleiotropic in their impact – here we are left with no hints about how this variant is driving the changes in AS. Again, AS events are about equally likely to have a NOVA1 binding signature as not. Based on homology, authors (plausibly) suggest that the I197V allele could disrupt the protein-RNA interface, but how would this work for those events without CLIP signals? Again, authors argue that the absence of a peak may reflect a technical 'artifact' (that detection requires the interaction to satisfy the chemical requirements for UV-induced crosslinking), but at ~60% of the detected AS events lacking a CLIP signal, this is a tough argument to swallow. Alternatively, authors suggest that those AS events without a CLIP signal might reflect 'secondary' effects of changes in other RBP whose splicing was directly impacted by I197V – again, if this manuscript were strictly focused on the molecular defects associated with I197V then one would expect some additional analysis of the 'true' CLIP-positive transcripts to determine the cis-features that enabled their differential regulation in the I197V relative to wildtype. Perhaps the differential AS events described here by authors are indeed real, but I'm not sufficiently convinced that they are anything more than noise that might be seen in a complex experiment such as this.

Regarding the vocalization studies that authors include, I would once again note that I am not expert in this field and so cannot strongly weigh in on the conclusions drawn. I would, however, reiterate my

hesitance given the statistical significance associated with many of these studies – I am concerned that these merely reflect noise. Perhaps equally importantly, I don't consider these studies to be a substitute for a more considered analysis of the molecular defects of the I197V variant.

First, we are very grateful for all of the reviewer's remarks; they helped us improve the quality of our work. We think that the revision led to more accurate interpretation of the significance of a single amino acid substitution of NOVA1 in modern humans. Our analysis indicates that a single amino acid substitution in NOVA1 has been clearly evolutionarily selected for in modern humans, and that this substitution can have a subtle but unique molecular influence in the mouse brain leading to the vocal phenotype.

As the reviewer states, the changes in transcriptome, AS, and CLIP events are small under the conditions tested. To examine the possibility that these were simply random noise, we performed additional analyses; we tested the enrichment of previously annotated biological terms of the affected transcripts and also examined their statistical significance by random resampling. The results indicate that these differences are indeed enriched for behavior and vocalization and were statistically significant.

In our manuscript, we describe high confident (although quantitatively small) NOVA1^{I197V}-dependent AS events by analyzing the transcriptome. 40% of transcripts with differential AS events contained NOVA1-CLIP peaks, whereas the rest did not. The presence of a NOVA1 CLIP-peak strongly suggests that the AS events are directly affected in a NOVA1^{I197V} manner, although as discussed below, more mechanistic studies will help clarify these actions. In contrast, there could be multiple reasons for the absence of NOVA1-CLIP peaks on the other 60% of transcripts with AS changes. The possibilities include, (1) potential mechanisms of NOVA1-tethering to target mRNAs (direct RNA binding vs indirect binding), (2) indirect effects through other RBPs that are regulated by NOVA1^{I197V} (true secondary effects), and (3) depth of analysis done in current CLIP data. Detailed explanation are as follows.

1. Indirect actions: if NOVA1 acts on the target transcript via protein-protein interactions (e.g. dimerization with NOVA2), it may not be possible to detect by CLIP methods that detect direct protein contacts with RNA.
2. Other RBPs: mRNA processing factors including several splicing-related factors (e.g. Dyrk1a, Hnrnpk, Upf1, Tra2b, Acin1, Mbnl1, Srrm4, Ern1, Rbm39, Taf6l, Ythdc1, Sfsmap, Hnrnp3, Ildr2, etc.) have been detected in the transcripts with differential AS (Supplemental Data 4), suggesting that these factors may mediate the NOVA1-dependent (but indirect) AS events. See for example Saito et al., Neuron 2019, discussed and cited in the text as reference 63.
3. Depth of analysis: CLIP studies were done in large areas of brain (cortex, midbrain and cerebellum). We have published data indicating that different RNA regulation occurs in different individual cell types, using cTag technology (Refs: Saito et al., Neuron 2019; Hwang et al., Neuron, 2017; Hale et al., eLife, 2021). In this work, RNA and CLIP studies were exploratory, looking for differences in large brain areas in humanized mice. They were not intended to study the biology of vocalization which is likely done by small populations of neurons, perhaps within individual nuclei (consider for example the detailed cellular analyses of cellular populations within the hypothalamus responsible for feeding behavior and jaw opening; see Tan et al., Nature 636:198–205 (2024) and Kosse et al., Nature 636:151-161 (2024)).

We would like to stress that we also do not consider the behavioral analysis as a substitute for the molecular analysis. Rather we think that the behavioral results now provided the basis for further molecular analysis (see also the exchanges with reviewer#5). We consider our molecular analyses, which show reproducible and significant changes in the brains of hNOVA1 mice, assessed by analysis of transcriptome, AS, and CLIP events as supportive of moving forward with behavioral tests. We examined the significance of these small but significant changes in hNOVA1-dependent RNA regulation by gene ontology analysis. This linked the RNA changes to a specific biological phenomenon—vocalization changes—which were confirmed by vocal behavior tests.

We think that our data lay the basis for exploring which neuronal populations are responsible for the phenotype will feed back to understanding the molecular mechanisms of the vocalization differences. Such fine-tuned experiments are now feasible to consider in the context of the hNOVA1 phenotype, and, we believe, will improve the resolution and the robustness of the sequencing analysis (RNA and CLIP) that this Reviewer, like us and hopefully many others, is especially interested in.

Again, we thank this reviewer for all the constructive criticisms and suggestions.

Reviewer #5 (Remarks to the Author):

I appreciate the efforts made by the authors to address my concerns, particularly with regard to the reanalysis of the vocalization data. The new analyses demonstrate a more careful approach in addressing sample size issues and corrections for multiple comparisons, which I had previously flagged as a potential concern. Indeed, some of the previously reported findings have failed to pass these criteria, but some findings were also found to be robust with stricter rules. The mixed findings presented in the revised manuscript provide a more nuanced view of the NOVA1 mutation's impact on vocalization.

The updated data of ultrasonic vocalizations (USVs) show important differences across developmental stages. The observation that the humanized genotype alters some features of pup USVs but that those changes do not persist into adulthood, while other differences emerge in adult USVs, adds depth to the analysis and conclusions.

At this stage, I think that it is mainly a judgment call. I don't have strong suggestions for further analysis and I think that overall the findings are of high interest and importance. Nevertheless, I still find that the link between these vocalization changes and the NOVA1 mutation remains suggestive rather than conclusive. On the other hand, the yardstick here may be previous studies on humanized mutation in the FOXP2 genes that have seen similar effects on vocalizations (even similar in nature as the authors now highlight in the text of the new revision). These FOXP2 findings have also been published in high-profile journals, and thus, I personally feel that this manuscript has passed the threshold.

I recommend that it be published in Nature Communications.

We deeply appreciate the reviewer for the comments on our revised manuscript. The reviewer's remarks have improved the accuracy and rigor of our vocalization analysis and have provided interesting insights. We do agree with the reviewer's sentiment that this study will launch further investigations into our findings linking the NOVA1 I197V substitution, molecular consequences, and changes in vocalization behavior.